# Probabilistic Linear Solvers for Machine Learning

**Jonathan Wenger**     **Philipp Hennig**
University of Tübingen
Max Planck Institute for Intelligent Systems
Tübingen, Germany
{jonathan.wenger, philipp.hennig}@uni-tuebingen.de

## Abstract

Linear systems are the bedrock of virtually all numerical computation. Machine learning poses specific challenges for the solution of such systems due to their scale, characteristic structure, stochasticity and the central role of uncertainty in the field. Unifying earlier work we propose a class of probabilistic linear solvers which jointly infer the matrix, its inverse and the solution from matrix-vector product observations. This class emerges from a fundamental set of desiderata which constrains the space of possible algorithms and recovers the method of conjugate gradients under certain conditions. We demonstrate how to incorporate prior spectral information in order to calibrate uncertainty and experimentally showcase the potential of such solvers for machine learning.

## 1   Introduction

Arguably one of the most fundamental problems in machine learning, statistics and scientific computation at large is the solution of linear systems of the form $Ax_* = b$, where $A \in \mathbb{R}^{n \times n}_{\mathrm{sym}}$ is a symmetric positive definite matrix [1–3]. Such matrices usually arise in the context of second-order or quadratic optimization problems and as Gram matrices. Some of the numerous application areas in machine learning and related fields are least-squares regression [4], kernel methods [5], Kalman filtering [6], Gaussian (process) inference [7], spectral graph theory [8], (linear) differential equations [9] and (stochastic) second-order methods [10].

Linear systems in machine learning are typically large-scale, have characteristic structure arising from generative processes, and are subject to noise. These distinctive features call for linear solvers that can explicitly make use of such structural information. While classic solvers are highly optimized for general problems, they lack key functionality for machine learning. In particular, they do not consider generative prior information about the matrix.

An important example are kernel Gram matrices, which exhibit specific sparsity structure and spectral properties, depending on the kernel choice and the generative process of the data. Exploiting such prior information is a prime application for probabilistic linear solvers, which aim to quantify numerical uncertainty arising from limited computational resources. Another key challenge, which we will not yet address here, are noisy matrix evaluations arising from data subsampling. Ultimately, linear algebra for machine learning should integrate all sources of uncertainty in a computational pipeline – aleatoric, epistemic and numerical – into one coherent probabilistic framework.

**Contribution**   This paper sets forth desiderata for probabilistic linear solvers which establish first principles for such methods. From these, we derive an algorithm incorporating prior information on the matrix $A$ or its inverse $A^{-1}$, which jointly estimates both via repeated application of $A$. This results in posterior beliefs over the two operators and the solution which quantify numerical uncertainty. Our approach unifies and extends earlier formulations and constitutes a new way of

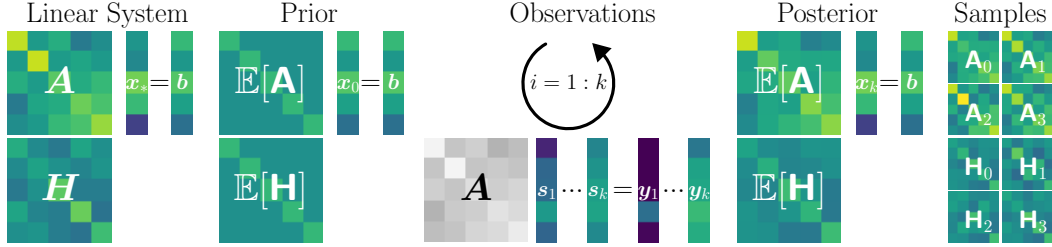

Figure 1: *Illustration of a probabilistic linear solver.* Given a prior for **A** or **H** modelling the linear operator $\boldsymbol{A}$ and its inverse $\boldsymbol{A}^{-1}$, posterior beliefs are inferred via observations $\boldsymbol{y}_i = \boldsymbol{A}\boldsymbol{s}_i$. This induces a distribution on the solution $\boldsymbol{x}_*$, quantifying numerical uncertainty arising from finite computation. The plot shows $k = 3$ iterations of Algorithm 1 on a toy problem of dimension $n = 5$.

interpreting linear solvers. Further, we propose a prior covariance class which recovers the method of conjugate gradients as its posterior mean and uses prior spectral information for uncertainty calibration, one of the primary shortcomings of probabilistic linear solvers. We conclude by presenting simplified examples of promising applications of such solvers within machine learning.

## 2    Probabilistic Linear Solvers

Let $\boldsymbol{A}\boldsymbol{x}_* = \boldsymbol{b}$ be a linear system with $\boldsymbol{A} \in \mathbb{R}^{n \times n}_{\text{sym}}$ positive definite and $\boldsymbol{b} \in \mathbb{R}^n$. *Probabilistic linear solvers* (PLS) [11–13] iteratively build a model for the linear operator $\boldsymbol{A}$, its inverse $\boldsymbol{H} = \boldsymbol{A}^{-1}$ or the solution $\boldsymbol{x}_*$, represented by random variables **A**, **H** or **x**. In the framework of probabilistic numerics [14, 15] such solvers can be seen as Bayesian agents performing *inference* via linear *observations* $\boldsymbol{Y} = [\boldsymbol{y}_1, \dots, \boldsymbol{y}_k] \in \mathbb{R}^{n \times k}$ resulting from *actions* $\boldsymbol{S} = [\boldsymbol{s}_1, \dots, \boldsymbol{s}_k] \in \mathbb{R}^{n \times k}$ given by an internal *policy* $\pi(\boldsymbol{s} \mid \textbf{A}, \textbf{H}, \textbf{x}, \boldsymbol{A}, \boldsymbol{b})$. For a matrix-variate prior $p(\textbf{A})$ or $p(\textbf{H})$ encoding prior (generative) information, our solver computes posterior beliefs over the matrix, its inverse and the solution of the linear system. An illustration of a probabilistic linear solver is given in Figure 1.

**Desiderata**    We begin by stipulating a fundamental set of desiderata for probabilistic linear solvers. To our knowledge such a list has not been collated before. Connecting previously disjoint threads, the following presents a roadmap for the development of these methods. Probabilistic linear solvers modelling $\boldsymbol{A}$ and $\boldsymbol{A}^{-1}$ must assume matrix-variate distributions which are expressive enough to capture structure and generative prior information either for $\boldsymbol{A}$ or its inverse. The distribution choice must also allow computationally efficient sampling and density evaluation. It should encode symmetry and positive definiteness and must be closed under positive linear combinations. Further, the two models for the system matrix or its inverse should be translatable into and consistent with each other. Actions $\boldsymbol{s}_i$ of a PLS should be model-based and induce a tractable distribution on linear observations $\boldsymbol{y}_i = \boldsymbol{A}\boldsymbol{s}_i$. Since probabilistic linear solvers are low-level procedures, their inference procedure must be computationally lightweight. Given (noise-corrupted) observations this requires tractable posteriors over **A**, **H** and **x**, which are calibrated in the sense that at convergence the true solution $\boldsymbol{x}_*$ represents a draw from the posterior $p(\textbf{x} \mid \boldsymbol{Y}, \boldsymbol{S})$. Finally, such solvers need to allow preconditioning of the problem and ideally should return beliefs over non-linear properties of the system matrix extending the functionality of classic methods. These desiderata are summarized concisely in Table 1.

### 2.1    Bayesian Inference Framework

Guided by these desiderata, we will now outline the inference framework for **A**, **H** and **x** forming the base of the algorithm. The choice of a matrix-variate prior distribution is severely limited by the desideratum that conditioning on linear observations $\boldsymbol{y}_i = \boldsymbol{A}\boldsymbol{s}_i$ must be tractable. This reduces the choice to stable distributions [16] and thus excludes candidates such as the Wishart, which has measure zero outside the cone of symmetric positive semi-definite matrices. For symmetric matrices, this essentially forces use of the symmetric matrix-variate normal distribution, introduced in this context by Hennig [11]. Given $\boldsymbol{A}_0, \boldsymbol{W}_0^{\textbf{A}} \in \mathbb{R}^{n \times n}_{\text{sym}}$, assume a prior distribution

$$p(\textbf{A}) = \mathcal{N}(\textbf{A}; \boldsymbol{A}_0, \boldsymbol{W}_0^{\textbf{A}} \otimes \boldsymbol{W}_0^{\textbf{A}}),$$

Table 1: *Desired properties of probabilistic linear solvers.* Symbols (✗, ∼, ✓) indicate which properties are encoded in our proposed solver (see Algorithm 1) and to what degree.

| No. | Property | Formulation | |
|-----|----------|-------------|---|
| (1) | distribution over matrices | $\mathbf{A} \sim \mathcal{D},\ p_{\mathcal{D}}(\mathbf{A})$ | ✓ |
| (2) | symmetry | $\mathbf{A} = \mathbf{A}^{\mathsf{T}}$ a.s. | ✓ |
| (3) | positive definiteness | $\forall \boldsymbol{v} \neq 0:\ \boldsymbol{v}^{\mathsf{T}}\mathbf{A}\boldsymbol{v} > 0$ a.s. | ∼ |
| (4) | positive linear combination in same distribution family | $\forall \alpha_j > 0:\ \sum_j \alpha_j \mathbf{A}_j \sim \mathcal{D}$ | ✓ |
| (5) | corresponding priors on the matrix and its inverse | $p(\mathbf{A}) \longleftrightarrow p(\mathbf{H})$ | ✓ |
| (6) | model-based policy | $\boldsymbol{s}_i \sim \pi(\boldsymbol{s} \mid \boldsymbol{A}, \boldsymbol{b}, \mathbf{A}, \mathbf{H}, \mathbf{x})$ | ✓ |
| (7) | matrix-vector product in tractable distribution family | $\mathbf{A}\boldsymbol{s} \sim \mathcal{D}'$ | ✓ |
| (8) | noisy observations | $p(\boldsymbol{Y} \mid \mathbf{A}, \boldsymbol{S}) = \mathcal{N}(\boldsymbol{Y}; \boldsymbol{A}\boldsymbol{S}, \boldsymbol{\Lambda})$ | ✗ |
| (9) | tractable posterior | $p(\mathbf{A} \mid \boldsymbol{Y}, \boldsymbol{S})$ or $p(\mathbf{H} \mid \boldsymbol{Y}, \boldsymbol{S})$ | ✓ |
| (10) | calibrated uncertainty | $\boldsymbol{x}_* \sim \mathcal{N}(\mathbb{E}[\mathbf{x}], \operatorname{Cov}[\mathbf{x}])$ | ∼ |
| (11) | preconditioning | $(\boldsymbol{P}^{-\mathsf{T}}\boldsymbol{A}\boldsymbol{P}^{-1})\boldsymbol{P}\boldsymbol{x}_* = \boldsymbol{P}^{-\mathsf{T}}\boldsymbol{b}$ | ✓ |
| (12) | distributions over non-linear derived quantities of $\boldsymbol{A}$ | $\det(\mathbf{A}),\ \sigma(\mathbf{A}),\ \mathbf{A} = \mathbf{L}^{\mathsf{T}}\mathbf{L}, \dots$ | ✗ |

where $\otimes$ denotes the symmetric Kronecker product [17].[1] The symmetric matrix-variate Gaussian induces a Gaussian distribution on linear observations. While it has non-zero measure only for symmetric matrices, its support is not the positive definite cone. However, positive definiteness can still be enforced post-hoc (see Proposition 1). We assume noise-free linear observations of the form $\boldsymbol{y}_i = \boldsymbol{A}\boldsymbol{s}_i$, leading to a Dirac likelihood

$$p(\boldsymbol{Y} \mid \mathbf{A}, \boldsymbol{S}) = \lim_{\varepsilon \downarrow 0} \mathcal{N}(\boldsymbol{Y}; \boldsymbol{A}\boldsymbol{S}, \varepsilon^2 \boldsymbol{I} \otimes \boldsymbol{I}) = \delta(\boldsymbol{Y} - \boldsymbol{A}\boldsymbol{S}).$$

The posterior distribution follows from the properties of Gaussians [4] and has been investigated in detail in previous work [18, 11, 13]. It is given by $p(\mathbf{A} \mid \boldsymbol{S}, \boldsymbol{Y}) = \mathcal{N}(\mathbf{A}; \boldsymbol{A}_k, \boldsymbol{\Sigma}_k)$ with

$$\boldsymbol{A}_k = \boldsymbol{A}_0 + \boldsymbol{\Delta}_0^{\mathbf{A}}\boldsymbol{U}^{\mathsf{T}} + \boldsymbol{U}(\boldsymbol{\Delta}_0^{\mathbf{A}})^{\mathsf{T}} - \boldsymbol{U}\boldsymbol{S}^{\mathsf{T}}\boldsymbol{\Delta}_0^{\mathbf{A}}\boldsymbol{U}^{\mathsf{T}}$$

$$\boldsymbol{\Sigma}_k = \boldsymbol{W}_0^{\mathbf{A}}(\boldsymbol{I}_n - \boldsymbol{S}\boldsymbol{U}^{\mathsf{T}}) \otimes \boldsymbol{W}_0^{\mathbf{A}}(\boldsymbol{I}_n - \boldsymbol{S}\boldsymbol{U}^{\mathsf{T}})$$

where $\boldsymbol{\Delta}_0^{\mathbf{A}} = \boldsymbol{Y} - \boldsymbol{A}_0\boldsymbol{S}$ and $\boldsymbol{U} = \boldsymbol{W}_0^{\mathbf{A}}\boldsymbol{S}(\boldsymbol{S}^{\mathsf{T}}\boldsymbol{W}_0^{\mathbf{A}}\boldsymbol{S})^{-1}$. We aim to construct a probabilistic model $\mathbf{H}$ for the inverse $\boldsymbol{H} = \boldsymbol{A}^{-1}$ consistent with the model $\mathbf{A}$ as well. However, not even in the scalar case does the inverse of a Gaussian have finite mean. We ask instead what Gaussian model for $\mathbf{H}$ is as consistent as possible with our observational model for $\mathbf{A}$. For a prior of the form $p(\mathbf{H}) = \mathcal{N}(\mathbf{H}; \boldsymbol{H}_0, \boldsymbol{W}_0^{\mathbf{H}} \otimes \boldsymbol{W}_0^{\mathbf{H}})$ and likelihood $p(\boldsymbol{S} \mid \mathbf{H}, \boldsymbol{Y}) = \delta(\boldsymbol{S} - \boldsymbol{H}\boldsymbol{Y})$, we analogously to the $\mathbf{A}$-model obtain a posterior distribution $p(\mathbf{H} \mid \boldsymbol{S}, \boldsymbol{Y}) = \mathcal{N}(\mathbf{H}; \boldsymbol{H}_k, \boldsymbol{\Sigma}_k^{\mathbf{H}})$ with

$$\boldsymbol{H}_k = \boldsymbol{H}_0 + \boldsymbol{\Delta}_0^{\mathbf{H}}(\boldsymbol{U}^{\mathbf{H}})^{\mathsf{T}} + \boldsymbol{U}^{\mathbf{H}}(\boldsymbol{\Delta}_0^{\mathbf{H}})^{\mathsf{T}} - \boldsymbol{U}^{\mathbf{H}}\boldsymbol{Y}^{\mathsf{T}}\boldsymbol{\Delta}_0^{\mathbf{H}}(\boldsymbol{U}^{\mathbf{H}})^{\mathsf{T}}$$

$$\boldsymbol{\Sigma}_k^{\mathbf{H}} = \boldsymbol{W}_0^{\mathbf{H}}(\boldsymbol{I}_n - \boldsymbol{Y}(\boldsymbol{U}^{\mathbf{H}})^{\mathsf{T}}) \otimes \boldsymbol{W}_0^{\mathbf{H}}(\boldsymbol{I}_n - \boldsymbol{Y}(\boldsymbol{U}^{\mathbf{H}})^{\mathsf{T}})$$

where $\boldsymbol{\Delta}_0^{\mathbf{H}} = \boldsymbol{S} - \boldsymbol{H}_0\boldsymbol{Y}$ and $\boldsymbol{U}^{\mathbf{H}} = \boldsymbol{W}_0^{\mathbf{H}}\boldsymbol{Y}(\boldsymbol{Y}^{\mathsf{T}}\boldsymbol{W}_0^{\mathbf{H}}\boldsymbol{Y})^{-1}$. In Section 3 we will derive a covariance class, which establishes correspondence between the two Gaussian viewpoints for the linear operator and its inverse and is consistent with our desiderata.

## 2.2 Algorithm

The above inference procedure leads to Algorithm 1. The degree to which the desiderata are encoded in our formulation of a PLS can be found in Table 1. We will now go into more detail about the policy, the choice of step size, stopping criteria and the implementation.

**Policy and Step Size** In each iteration our solver collects information about the linear operator $\boldsymbol{A}$ via actions $\boldsymbol{s}_i$ determined by the policy $\pi(\boldsymbol{s} \mid \mathbf{A}, \mathbf{H}, \mathbf{x}, \boldsymbol{A}, \boldsymbol{b})$. The next action $\boldsymbol{s}_i = -\mathbb{E}[\mathbf{H}]\boldsymbol{r}_{i-1}$ is

Algorithm 1: Probabilistic Linear Solver with Uncertainty Calibration
---
1  **procedure** PROBLINSOLVE($A(\cdot), b, \mathbf{A}, \mathbf{H}$)                               # prior for $\mathbf{A}$ or $\mathbf{H}$
2      $x_0 \leftarrow \mathbb{E}[\mathbf{H}]b$                                                    # initial guess
3      $r_0 \leftarrow Ax_0 - b$
4      **while** $\min(\sqrt{\operatorname{tr}(\operatorname{Cov}[\mathbf{x}])}, \|r_i\|_2) > \max(\delta_{\mathrm{rtol}}\|b\|_2, \delta_{\mathrm{atol}})$ **do**   # stopping criteria
5          $s_i \leftarrow -\mathbb{E}[\mathbf{H}]r_{i-1}$                                          # compute action via policy
6          $y_i \leftarrow As_i$                                                                  # make observation
7          $\alpha_i \leftarrow -s_i^\mathsf{T} r_{i-1}(s_i^\mathsf{T} y_i)^{-1}$                  # optimal step size
8          $x_i \leftarrow x_{i-1} + \alpha_i s_i$                                                 # update solution estimate
9          $r_i \leftarrow r_{i-1} + \alpha_i y_i$                                                 # update residual
10         $\mathbf{A} \leftarrow \text{INFER}(\mathbf{A}, s_i, y_i)$                              # infer posterior distributions
11         $\mathbf{H} \leftarrow \text{INFER}(\mathbf{H}, s_i, y_i)$                              # (see Section 2.1)
12         $\boldsymbol{\Phi}, \boldsymbol{\Psi} \leftarrow \text{CALIBRATE}(S, Y)$                # calibrate uncertainty
13     $\mathbf{x} \leftarrow \mathcal{N}(x_k, \operatorname{Cov}[\mathbf{H}b])$                   # belief over solution
14     **return** $(\mathbf{x}, \mathbf{A}, \mathbf{H})$
---

chosen based on the current belief about the inverse. If $\mathbb{E}[\mathbf{H}] = A^{-1}$, i.e. if the solver's estimate for the inverse equals the true inverse, then Algorithm 1 converges in a single step since

$$x_{i-1} + s_i = x_{i-1} - \mathbb{E}[\mathbf{H}]r_{i-1} = x_{i-1} - A^{-1}(Ax_{i-1} - b) = A^{-1}b = x_*.$$

The step size minimizing the quadratic $q(x_i + \alpha s_i) = \frac{1}{2}(x_i + \alpha s_i)^\mathsf{T} A(x_i + \alpha s_i) - b^\mathsf{T}(x_i + \alpha s_i)$ along the action $s_i$ is given by $\alpha_i = \arg\min_\alpha q(x_i + \alpha s_i) = s_i^\mathsf{T}(b - Ax_i)(s_i^\mathsf{T} As_i)^{-1}$.

**Stopping Criteria**   Classic linear solvers typically use stopping criteria based on the current residual of the form $\|Ax_i - b\|_2 \leq \max(\delta_{\mathrm{rtol}}\|b\|_2, \delta_{\mathrm{atol}})$ for relative and absolute tolerances $\delta_{\mathrm{rtol}}$ and $\delta_{\mathrm{atol}}$. However, this residual may oscillate or even increase in all but the last step even if the error $\|x_* - x_i\|_2$ is monotonically decreasing [19, 20]. From a probabilistic point of view, we should stop if our posterior uncertainty is sufficiently small. Assuming the posterior covariance is calibrated, it holds that $(\mathbb{E}_{x_*}[\|x_* - \mathbb{E}[\mathbf{x}]\|_2])^2 \leq \mathbb{E}_{x_*}[\|x_* - \mathbb{E}[\mathbf{x}]\|_2^2] = \operatorname{tr}(\operatorname{Cov}[\mathbf{x}])$. Hence given calibration, we can bound the expected (relative) error between our estimate and the true solution by terminating when $\sqrt{\operatorname{tr}(\operatorname{Cov}[\mathbf{x}])} \leq \max(\delta_{\mathrm{rtol}}\|b\|_2, \delta_{\mathrm{atol}})$. A probabilistic criterion is also necessary for an extension to the noisy setting, where classic convergence criteria become stochastic. However, probabilistic linear solvers typically suffer from miscalibration [21], an issue we will address in Section 3.

**Implementation**   We provide an open-source implementation of Algorithm 1 as part of PROBNUM, a Python package implementing probabilistic numerical methods, in an online code repository:

$$f\left(\bigwedge\right) \quad \texttt{https://github.com/probabilistic-numerics/probnum}$$

The mean and covariance up- and downdates in Section 2.1 when performed iteratively are of low rank. In order to maintain numerical stability these updates can instead be performed for their respective Cholesky factors [22]. This also enables computationally efficient sampling or evaluation of probability density functions downstream.

## 2.3   Theoretical Properties

This section details some theoretical properties of our method such as its convergence behavior and computational complexity. In particular we demonstrate that for a specific prior choice Algorithm 1 recovers the method of conjugate gradients as its solution estimate. All proofs of results in this section and the next can be found in the supplementary material. We begin by establishing that our solver is a *conjugate directions method* and therefore converges in at most $n$ steps in exact arithmetic.

**Theorem 1** (Conjugate Directions Method)
*Given a prior $p(\mathbf{H}) = \mathcal{N}(\mathbf{H}; H_0, W_0^\mathbf{H} \otimes W_0^\mathbf{H})$ such that $H_0, W_0^\mathbf{H} \in \mathbb{R}_{\mathrm{sym}}^{n \times n}$ positive definite, then actions $s_i$ of Algorithm 1 are $A$-conjugate, i.e. for $0 \leq i, j \leq k$ with $i \neq j$ it holds that $s_i^\mathsf{T} As_j = 0$.*

We can obtain a better convergence rate by placing stronger conditions on the prior covariance class as outlined in Section 3. Given these assumptions, Algorithm 1 recovers the iterates of (preconditioned) CG and thus inherits its favorable convergence behavior (overviews in [23, 10]).

**Theorem 2** (Connection to the Conjugate Gradient Method)
*Given a scalar prior mean $\boldsymbol{A}_0 = \boldsymbol{H}_0^{-1} = \alpha \boldsymbol{I}$ with $\alpha > 0$, assume (1) and (2) hold, then the iterates $\boldsymbol{x}_i$ of Algorithm 1 are identical to the ones produced by the conjugate gradient method.*

A common phenomenon observed when implementing conjugate gradient methods is that due to cancellation in the computation of the residuals, the search directions $\boldsymbol{s}_i$ lose $\boldsymbol{A}$-conjugacy [24, 25, 3]. In fact, they can become independent up to working precision for $i$ large enough [25]. One way to combat this is to perform complete reorthogonalization of the search directions in each iteration as originally suggested by Lanczos [26]. Algorithm 1 does this *implicitly* via its choice of policy which depends on all previous search directions as opposed to just $\boldsymbol{s}_{i-1}$ for (naive) CG.

**Computational Complexity**   The solver has time complexity $\mathcal{O}(kn^2)$ for $k$ iterations without uncertainty calibration. Compared to CG, inferring the posteriors in Section 2.1 adds an overhead of four outer products and four matrix-vector products per iteration, given (1) and (2). Uncertainty calibration outlined in Section 3 adds between $\mathcal{O}(1)$ and $\mathcal{O}(k^3)$ per iteration depending on the sophistication of the scheme. Already for moderate $n$ this is dominated by the iteration cost. In practice, means and covariances do not need to be formed in memory. Instead they can be evaluated lazily as linear operators $\boldsymbol{v} \mapsto \boldsymbol{L}\boldsymbol{v}$, if $\boldsymbol{S}$ and $\boldsymbol{Y}$ are stored. This results in space complexity $\mathcal{O}(kn)$.

## 2.4   Related Work

Numerical methods for the solution of linear systems have been studied in great detail since the last century. Standard texts [1, 2, 10, 3] give an in-depth overview. The conjugate gradient method recovered by our algorithm for a specific choice of prior was introduced by Hestenes and Stiefel [19]. Recently, randomization has been exploited to develop improved algorithms for large-scale problems arising from machine learning [27, 28]. The key difference to our approach is that we do not rely on sampling to approximate large-scale matrices, but instead perform probabilistic inference. Our approach is based on the framework of probabilistic numerics [14, 15] and is a natural continuation of previous work on probabilistic linear solvers. In historical order, Hennig and Kiefel [18] provided a probabilistic interpretation of Quasi-Newton methods, which was expanded upon in [11]. This work also relied on the symmetric matrix-variate Gaussian as used in our paper. Bartels and Hennig [29] estimate numerical error in approximate least-squares solutions by using a probabilistic model. More recently, Cockayne et al. [21] proposed a Bayesian conjugate gradient method performing inference on the solution of the system. This was connected to the matrix-based view by Bartels et al. [13].

## 3   Prior Covariance Class

Having outlined the proposed algorithm, this section derives a prior covariance class which satisfies nearly all desiderata, connects the two modes of prior information and allows for calibration of uncertainty by appropriately choosing remaining degrees of freedom in the covariance. The third desideratum posited that **A** and **H** should be almost surely positive definite. This evidently does not hold for the matrix-variate Gaussian. However, we can restrict the choice of admissable $\bar{\boldsymbol{W}}_0^{\mathsf{A}}$ to act like $\boldsymbol{A}$ on $\text{span}(\boldsymbol{S})$. This in turn induces a positive definite posterior mean.

**Proposition 1** (Hereditary Positive Definiteness [30, 18])
*Let $\boldsymbol{A}_0 \in \mathbb{R}^{n \times n}_{\text{sym}}$ be positive definite. Assume the actions $\boldsymbol{S}$ are $\boldsymbol{A}$-conjugate and $\boldsymbol{W}_0^{\mathsf{A}}\boldsymbol{S} = \boldsymbol{Y}$, then for $i \in \{0, \ldots, k-1\}$ it holds that $\boldsymbol{A}_{i+1}$ is symmetric positive definite.*

Prior information about the linear system usually concerns the matrix $\boldsymbol{A}$ itself and not its inverse, but the inverse is needed to infer the solution $\boldsymbol{x}_*$ of the linear problem. So a way to translate between a Gaussian distribution on **A** and **H** is crucial. Previous works generally committed to either one view or the other, potentially discarding available information. Below, we show that the two correspond, if we allow ourselves to constrain the space of possible models. We impose the following condition.

**Definition 1**
Let $\boldsymbol{A}_i$ and $\boldsymbol{H}_i$ be the means of **A** and **H** at step $i$. We say a prior induces *posterior correspondence* if $\boldsymbol{A}_i^{-1} = \boldsymbol{H}_i$ for all $0 \le i \le k$. If only $\boldsymbol{A}_i^{-1}\boldsymbol{Y} = \boldsymbol{H}_i\boldsymbol{Y}$, *weak posterior correspondence* holds.

The following theorem establishes a sufficient condition for weak posterior correspondence. For an asymmetric prior model one can establish the stronger notion of posterior correspondence. A proof is included in the supplements.

**Theorem 3** (Weak Posterior Correspondence)
*Let $W_0^{\mathsf{H}} \in \mathbb{R}_{\mathrm{sym}}^{n \times n}$ be positive definite. Assume $H_0 = A_0^{-1}$, and that $W_0^{\mathsf{A}}, A_0, W_0^{\mathsf{H}}$ satisfy*

$$W_0^{\mathsf{A}} S = Y, \tag{1}$$

$$S^{\mathsf{T}}(W_0^{\mathsf{A}} A_0^{-1} - A W_0^{\mathsf{H}}) = 0, \tag{2}$$

*then weak posterior correspondence holds for the symmetric Kronecker covariance.*

Given the above, let $A_0$ be a symmetric positive definite prior mean and $H_0 = A_0^{-1}$. Define the orthogonal projections $P_S^A = AS(S^{\mathsf{T}}AS)^{-1}S^{\mathsf{T}}A$ and $P_Y^{H_0} = A_0^{-1}Y(Y^{\mathsf{T}}A_0^{-1}Y)^{-1}Y^{\mathsf{T}}A_0^{-1}$ with respect to the inner products induced by $A$ and $A_0^{-1}$, as well as $P_{S^\perp} = I - S(S^{\mathsf{T}}S)^{-1}S^{\mathsf{T}}$ and $P_{Y^\perp} = I - Y(Y^{\mathsf{T}}Y)^{-1}Y^{\mathsf{T}}$ projecting to the spaces $\mathrm{span}(S)^\perp$ and $\mathrm{span}(Y)^\perp$. We propose the following prior covariance class given by the prior covariance factors

$$W_0^{\mathsf{A}} = P_S^A + P_{S^\perp} \Phi P_{S^\perp} \quad \text{and} \quad W_0^{\mathsf{H}} = P_Y^{H_0} + P_{Y^\perp} \Psi P_{Y^\perp}, \tag{3}$$

where $\Phi \in \mathbb{R}^{n \times n}$ and $\Psi \in \mathbb{R}^{n \times n}$ are degrees of freedom. This choice of covariance class satisfies Theorem 1, Proposition 1, Theorem 3 and for a scalar mean also Theorem 2. Therefore, it produces symmetric realizations, has symmetric positive semi-definite means, it links the matrix and the inverse view and at any given time only needs access to $v \mapsto Av$ not $A$ itself. It is also compatible with a preconditioner by simply transforming the given linear problem.

This class can be interpreted as follows. The derived covariance factor $W_0^{\mathsf{A}}$ acts like $A$ on the space $\mathrm{span}(S)$ explored by the algorithm. On the remaining space its uncertainty is additionally determined by the degrees of freedom in $\Phi$. Likewise, our best guess for $A^{-1}$ is $A_0^{-1}$ on the space spanned by $Y$. On the orthogonal space $\mathrm{span}(Y)^\perp$ the uncertainty is also influenced by $\Psi$. Note that the prior depends on actions and observations collected during a run of Algorithm 1, hence one might call this an empirical Bayesian approach. This begs the question how the algorithm is realizable for the proposed prior (3) given its dependence on future data. Notice that the posterior mean in Section 2.1 only depends on $W_0^{\mathsf{A}} S = Y$ *not* on $W_0^{\mathsf{A}}$ alone. Using eq. (3), at iteration $i$ we have $W_0^{\mathsf{A}} S_{1:i} = Y_{1:i}$, i.e. the observations made up to this point. Similar reasoning applies for the inverse. Now, the posterior covariances do depend on $W_0^{\mathsf{A}}$, respectively $W_0^{\mathsf{H}}$ alone, but prior to convergence we only require $\mathrm{tr}(\mathrm{Cov}[\mathbf{x}])$ for the stopping criterion. We show in Section S4.3 under the assumptions of Theorem 2 how to compute this at any iteration $i$ independent of future actions and observations. Therefore prior to convergence of Algorithm 1 *the covariance factors are never explicitly formed*.

**Uncertainty Calibration**  Generally the actions of Algorithm 1 identify eigenpairs $(\lambda_i, v_i)$ in descending order of $\lambda_i v_i^{\mathsf{T}} r_0$ which is a well-known behavior of CG (see eqn. 5.29 in [10]). In part, since this dynamic of the underlying Krylov subspace method is not encoded in the prior, the solver in its current form is typically miscalibrated (see also [21]). While this non-linear information is challenging to include in the Gaussian framework, we can choose $\Phi$ and $\Psi$ in (3) to empirically calibrate uncertainty. This can be interpreted as a form of hyperparameter optimization similar to optimization of kernel parameters in GP regression.

We would like to encode prior knowledge about the way $A$ and $H$ act in the respective orthogonal spaces $\mathrm{span}(S)^\perp$ and $\mathrm{span}(Y)^\perp$. For the Rayleigh quotient $R(A, v) = (v^{\mathsf{T}}Av)(v^{\mathsf{T}}v)^{-1}$ it holds that $\lambda_{\min}(A) \leq R(A, v) \leq \lambda_{\max}(A)$. Hence for vectors $v$ lying in the respective null spaces of $S$ and $Y$ our uncertainty should be determined by the not yet explored eigenvalues $\lambda_{k+1}, \ldots, \lambda_n$ of $A$ and $H$. Without prior information about the eigenspaces, we choose $\Phi = \phi I$ and $\Psi = \psi I$. If a priori we know the respective spectra, a straightforward choice is

$$\phi = \psi^{-1} = \frac{1}{n-k} \sum_{i=k+1}^{n} \lambda_i(A).$$

In the absence of prior spectral information we can make use of already collected quantities during a run of Algorithm 1. We build a one-dimensional regression model $p(\ln R_i \mid Y, S)$ for the ln-Rayleigh quotient $\ln R(A, s_i)$ given actions $s_i$. Such a model can then encode the well studied

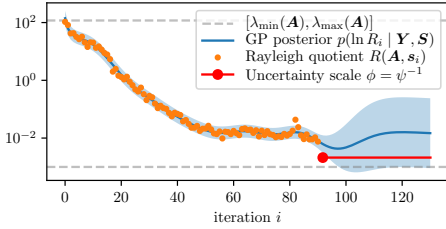

Figure 2: *Rayleigh regression.* Uncertainty calibration via GP regression on $\{\ln R(\boldsymbol{A}, \boldsymbol{s}_i)\}_{i=1}^{k}$ after $k = 91$ iterations of Algorithm 1 on an $n = 1000$ dimensional Mátern32 kernel matrix inversion problem. The degrees of freedom $\phi = \psi^{-1} > 0$ are set based on the average predicted Rayleigh quotient for the remaining $n - k = 909$ dimensions.

Table 2: *Uncertainty calibration for kernel matrices.* Monte Carlo estimate $\bar{w} \approx \mathbb{E}_{\boldsymbol{x}_*}[w(\boldsymbol{x}_*)]$ measuring calibration given $10^5/n$ sampled linear problems of the form $(\boldsymbol{K} + \varepsilon^2 \boldsymbol{I})\boldsymbol{x}_* = \boldsymbol{b}$ for each kernel and calibration method. For $\bar{w} \approx 0$ the solver is well calibrated, for $\bar{w} \gg 0$ underconfident and for $\bar{w} \ll 0$ overconfident.

| Kernel | $n$ | none | Rayleigh | $\varepsilon^2$ | $\overline{\lambda}_{k+1:n}$ |
|---|---|---|---|---|---|
| Matérn32 | $10^2$ | $-5.99$ | $-0.24$ | $0.32$ | $0.09$ |
| Matérn32 | $10^3$ | $-1.93$ | $7.53$ | $4.26$ | $4.19$ |
| Matérn32 | $10^4$ | $3.87$ | $17.16$ | $8.48$ | $8.47$ |
| Matérn52 | $10^2$ | $-7.84$ | $-1.01$ | $-0.76$ | $-0.80$ |
| Matérn52 | $10^3$ | $-4.63$ | $1.43$ | $-0.80$ | $-0.81$ |
| Matérn52 | $10^4$ | $-4.34$ | $10.81$ | $0.80$ | $0.80$ |
| RBF | $10^2$ | $-7.53$ | $-0.70$ | $-0.84$ | $-0.87$ |
| RBF | $10^3$ | $-4.94$ | $6.60$ | $0.77$ | $0.77$ |
| RBF | $10^4$ | $0.14$ | $21.32$ | $2.92$ | $2.92$ |

behaviour of CG, whose Rayleigh coefficients rapidly decay at first, followed by a slower continuous decay [10]. Figure 2 illustrates this approach using a GP regression model. At convergence, we use the prediction of the Rayleigh quotient for the remaining $n - k$ dimensions by choosing

$$\phi = \psi^{-1} = \exp\left( \frac{1}{n - k} \sum_{i=k+1}^{n} \mathbb{E}[\ln R_i \mid \mathbf{A}, \boldsymbol{S}] \right),$$

i.e. uncertainty about actions in $\text{span}(\boldsymbol{S})^{\perp}$ is calibrated to be the average Rayleigh quotient as an approximation to the spectrum. Depending on the application a simple or more complex model may be useful. For large problems, where generally $k \ll n$, more sophisticated schemes become computationally feasible. However, these do not necessarily need to be computationally demanding due to the simple nature of this one-dimensional regression problem with few data. For example, approximate [31] or even exact GP regression [32] is possible in $\mathcal{O}(k)$ using a Kalman filter.

## 4 Experiments

This section demonstrates the functionality of Algorithm 1. We choose some – deliberately simple – example problems from machine learning and scientific computation, where the solver can be used to quantify uncertainty induced by finite computation, solve multiple consecutive linear systems, and propagate information between problems.

**Gaussian Process Regression** GP regression [7] infers a latent function $f : \mathbb{R}^N \to \mathbb{R}$ from data $\boldsymbol{D} = (\boldsymbol{X}, \boldsymbol{y})$, where $\boldsymbol{X} \in \mathbb{R}^{n \times N}$ and $\boldsymbol{y} \in \mathbb{R}^n$. Given a prior $p(f) = \mathcal{GP}(f; 0, k)$ with kernel $k$ for the unknown function $f$, the posterior mean and marginal variance at $m$ new inputs $\tilde{\boldsymbol{x}} \in \mathbb{R}^{N \times m}$ are $\mathbb{E}[\tilde{\boldsymbol{f}}] = \tilde{\boldsymbol{k}}^{\intercal}(\boldsymbol{K} + \varepsilon^2 \boldsymbol{I})^{-1} \boldsymbol{y}$ and $\mathbb{V}[\tilde{\boldsymbol{f}}] = k(\tilde{\boldsymbol{x}}, \tilde{\boldsymbol{x}}) - \tilde{\boldsymbol{k}}^{\intercal}(\boldsymbol{K} + \varepsilon^2 \boldsymbol{I})^{-1} \tilde{\boldsymbol{k}}$, where $\boldsymbol{K} = k(\boldsymbol{X}, \boldsymbol{X}) \in \mathbb{R}^{n \times n}$ is the Gram matrix of the kernel and $\tilde{\boldsymbol{k}} = k(\boldsymbol{X}, \tilde{\boldsymbol{x}}) \in \mathbb{R}^{n \times m}$. The bulk of computation during prediction arises from solving the linear system $(\boldsymbol{K} + \varepsilon^2 \boldsymbol{I})\boldsymbol{z} = \boldsymbol{b}$ for some right-hand side $\boldsymbol{b} \in \mathbb{R}^n$ repeatedly. When using a probabilistic linear solver for this task, we can quantify the uncertainty arising from finite computation as well as the belief of the solver about the shape of the GP at a set of not yet computed inputs. Figure 3 illustrates this. In fact, we can estimate the marginal variance of the GP without solving the linear system again by multiplying $\tilde{\boldsymbol{k}}$ with the estimated inverse of $\boldsymbol{K} + \varepsilon^2 \boldsymbol{I}$. In large-scale applications, we can trade off computational expense for increased uncertainty arising from the numerical approximation and quantified by the probabilistic linear solver. By assessing the numerical uncertainty arising from not exploring the full space, we can judge the quality of the estimated GP mean and marginal variance.

**Kernel Gram Matrix Inversion** Consider a linear problem $\boldsymbol{K}\boldsymbol{x}_* = \boldsymbol{b}$, where $\boldsymbol{K}$ is generated by a Mercer kernel. For a $\nu$-times continuously differentiable kernel the eigenvalues $\lambda_n(\boldsymbol{K})$ decay

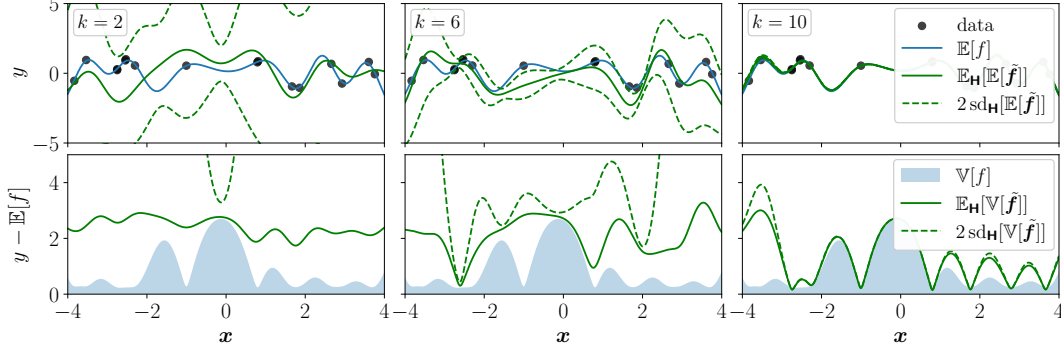

Figure 3: *Numerical uncertainty in GP inference.* Computing posterior mean and covariance of a GP regression using a PLS. *Top:* GP mean for a toy data set ($n = 16$) computed with increasing number of iterations $k$ of Algorithm 1. The numerical estimate of the GP mean approaches the true mean. Note that the numerical variance is different from the marginal variance of the GP. *Bottom:* GP variance and estimate of GP variance with numerical uncertainty. The GP variance estimate is computed using the estimated inverse from computing $\mathbb{E}[\tilde{\boldsymbol{f}}]$ *without any additional solver iterations.*

approximately as $|\lambda_n| \in \mathcal{O}(n^{-\nu-\frac{1}{2}})$ [33]. We can make use of this generative prior information by specifying a parametrized prior mean $\mu(n) = \ln(\theta_0' n^{-\theta_1}) = \theta_0 - \theta_1 \ln(n)$ for the ln-Rayleigh quotient model. Typically, such Gram matrices are ill-conditioned and therefore $\boldsymbol{K}' = \boldsymbol{K} + \varepsilon^2 \boldsymbol{I}$ is used instead, implying $\lambda(\boldsymbol{K}')_i \geq \varepsilon^2$. In order to assess calibration we apply various differentiable kernels to the airline delay dataset from January 2020 [34]. We compute the ln-ratio statistic $w(\boldsymbol{x}_*) = \frac{1}{2} \ln(\text{tr}(\text{Cov}[\mathbf{x}])) - \ln(\|\boldsymbol{x}_* - \mathbb{E}[\mathbf{x}]\|_2)$ for no calibration, calibration via Rayleigh quotient GP regression using $\mu(n)$ as a prior mean, calibration by setting $\phi = \varepsilon^2$ and calibration using the average spectrum $\phi = \overline{\lambda}_{k+1:n}$. The average $\overline{w}$ for $10^5/n$ randomly sampled test problems is shown in Table 2.[2] Without any calibration the solver is generally overconfident. All tested calibration procedures reverse this, resulting in more cautious uncertainty estimates. We observe that Rayleigh quotient regression overcorrects for larger problems. This is due to the fact that its model correctly predicts $\boldsymbol{K}$ to be numerically singular from the dominant Rayleigh quotients, however it misses the information that the spectrum of $\boldsymbol{K}'$ is bounded from below by $\varepsilon^2$. If we know the (average) of the remaining spectrum, significantly better calibration can be achieved, but often this information is not available. Nonetheless, since in this setting the majority of eigenvalues satisfy $\lambda(\boldsymbol{K}')_i \approx \varepsilon^2$ by choosing $\phi = \psi^{-1} = \varepsilon^2$, we can get to the same degree of calibration. Therefore, we can improve the solver's uncertainty calibration at constant cost $\mathcal{O}(1)$ per iteration. For more general problems involving Gram matrices without damping we may want to rely on Rayleigh regression instead.

**Galerkin's Method for PDEs** In the spirit of applying machine learning approaches to problems in the physical sciences and vice versa [35], we use Algorithm 1 for the approximate solution of a PDE via Galerkin's method [9]. Consider the Dirichlet problem for the Poisson equation given by

$$\begin{cases} -\Delta u(x,y) = f(x,y) & (x,y) \in \text{int } \Omega \\ u(x,y) = u_{\partial\Omega}(x,y) & (x,y) \in \partial\Omega \end{cases}$$

where $\Omega$ is a connected open region with sufficiently regular boundary and $u_{\partial\Omega} : \partial\Omega \to \mathbb{R}$ defines the boundary conditions. One obtains an approximate solution by projecting the weak formulation of the PDE to a finite dimensional subspace. This results in the *Galerkin equation* $\boldsymbol{Au} = \boldsymbol{f}$, i.e. a linear system where $\boldsymbol{A}$ is the Gram matrix of the associated bilinear form. Figure 4 shows the induced uncertainty on the solution of the Dirichlet problem for $f(x,y) = 15$ and $u_{\partial\Omega}(x,y) = (x^2 - 2y)^2(1 + \sin(2\pi x))$. The mesh and corresponding Gram matrix were computed using FENICS [36]. We can exploit two properties of Algorithm 1 in this setting. First, if we need to solve multiple related problems $(\boldsymbol{A}_j, \boldsymbol{f}_j)_j$, by solving a single problem we obtain an estimate of the solution to all other problems. We can successively use the posterior over the inverse as a prior for the next problem. This approach is closely related to subspace recycling in numerical linear algebra [37, 38].

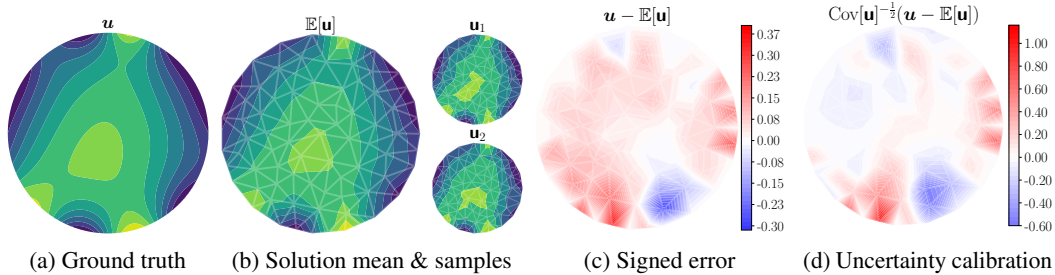

(a) Ground truth     (b) Solution mean & samples     (c) Signed error     (d) Uncertainty calibration

Figure 4: *Solving the Dirichlet problem with a probabilistic linear solver.* Figures 4a and 4b show the ground truth and mean of the solution computed with Algorithm 1 after $k = 23$ iterations along with samples from the posterior. The posterior on the coarse mesh can be used to assess uncertainty about the solution on a finer mesh. The signed error computed on the coarse mesh in Figure 4c shows that the approximation is better near the top boundary of $\Omega$. Given perfect uncertainty calibration, Figure 4d represents a sample from $\mathcal{N}(\mathbf{0}, \boldsymbol{I})$. The apparent structure in the plot and smaller than expected deviations in the upper part of $\Omega$ indicate the conservative confidence estimate of the solver.

Second, suppose we first compute a solution in a low-dimensional subspace corresponding to a coarse discretization for computational efficiency. We can then leverage the estimated solution to extrapolate to an (adaptively) refined discretization based on the posterior uncertainty. In machine learning lingo these two approaches can be viewed as forms of *transfer learning*.

## 5    Conclusion

In this work, we condensed a line of previous research on probabilistic linear algebra into agit st self-contained algorithm for the solution of linear problems in machine learning. We proposed first principles to constrain the space of possible generative models and derived a suitable covariance class. In particular, our proposed framework incorporates prior knowledge on the system matrix or its inverse and performs inference for both in a *consistent* fashion. Within our framework we identified parameter choices that recover the iterates of conjugate gradients in the mean, but add calibrated uncertainty around them in a computationally lightweight manner. To our knowledge our solver, available as part of the PROBNUM package, is the first practical implementation of this kind. In the final parts of this paper we showcased applications like kernel matrix inversion, where prior spectral information can be used for uncertainty calibration and outlined example use-cases for propagation of numerical uncertainty through computations. Naturally, there are also limitations remaining. While our theoretical framework can incorporate noisy matrix-vector product evaluations into its inference procedure via a Gaussian likelihood, practically *tractable* inference in the inverse model is more challenging. Our solver also opens up new research directions. In particular, our outlined regression model on the Rayleigh quotient may lead to a probabilistic model of the eigenspectrum. Finally, the matrix-based view of probabilistic linear solvers could inform probabilistic approaches to matrix decompositions, analogous to the way Lanczos methods are used in the classical setting.

## Broader Impact

Our research on probabilistic linear solvers is primarily aimed at members of the machine learning field working on uncertainty estimation which use linear solvers as part of their toolkit. We are convinced that numerical uncertainty induced by finite computational resources is a key missing component to be quantified in machine learning settings. By making numerical uncertainty explicit like our solver does, holistic probabilistic models incorporating all sources of uncertainty become possible. In fact, we hope that this line of work stimulates further research into numerical linear algebra for machine learning, a topic that has been largely considered solved by the community.

This is first and foremost a methods paper aiming to improve the quantification of numerical uncertainty in linear problems. While methodological papers may seem far removed from application and questions of ethical and societal impact, this is not the case. Precisely due to the general nature of the problem setting, the linear solver presented in this work is applicable to a broad range of applications,

from regression on flight data, to optimization in robotics, to the solution of PDEs in meteorology. The flip-side of this potential impact is that arguably, down the line, methodological research suffers from dual use more than any specialized field. While we cannot control the use of a probabilistic linear solver due to its general applicability, we have tried, to the best of our ability, to ensure it performs as intended.

We are hopeful that no specific population group is put at a disadvantage through this research. We are providing an open-source implementation of our method and of all experiments contained in this work. Therefore anybody with access to the internet is able to retrieve and reproduce our findings. In this manner we hope to adress the important issues of accessibility and reproducibility.

## Acknowledgments and Disclosure of Funding

The authors gratefully acknowledge financial support by the European Research Council through ERC StG Action 757275 / PANAMA; the DFG Cluster of Excellence "Machine Learning - New Perspectives for Science", EXC 2064/1, project number 390727645; the German Federal Ministry of Education and Research (BMBF) through the Tübingen AI Center (FKZ: 01IS18039A); and funds from the Ministry of Science, Research and Arts of the State of Baden-Württemberg.

JW is grateful to the International Max Planck Research School for Intelligent Systems (IMPRS-IS) for support.

We thank the reviewers for helpful comments and suggestions. JW would also like to thank Alexandra Gessner and Felix Dangel for a careful reading of an earlier version of this manuscript.

## Footnotes

[1]See Sections S2 and S3 of the supplementary material for more detail on Kronecker-type products and matrix-variate normal distributions.

[2]We decrease the number of samples with the dimension because forming *dense* kernel matrices in memory and computing their eigenvalues becomes computationally prohibitive – *not* because of the cost of our solver.

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
