[Supplementary Material]

# Supplementary Material: Probabilistic Linear Solvers for Machine Learning

**Jonathan Wenger**     **Philipp Hennig**
University of Tübingen
Max Planck Institute for Intelligent Systems
Tübingen, Germany
{jonathan.wenger, philipp.hennig}@uni-tuebingen.de

This supplement complements the paper *Probabilistic Linear Solvers for Machine Learning* and is structured as follows. Section S1 explains the approach of probabilistic numerics to model (deterministic) numerical problems probabilistically in more depth. Section S2 introduces different variants of Kronecker products used to define matrix-variate normal distributions in Section S3. Section S4 details the matrix-based inference procedure of probabilistic linear solvers based on matrix-vector product observations. It also contains some more explanation regarding prior construction and stopping criteria. Section S5 and Section S6 outline theoretical results from the paper and properties of the proposed covariance class, in particular detailed proofs. Finally, Section S7 provides some background for the application of probabilistic linear solvers to the solution of discretized partial differential equations. To provide a clear exposition to the reader in some sections we restate results from the literature. References referring to sections, equations or theorem-type environments within this document are tagged with 'S', while references to, or results from the main paper are stated as is.

**Preliminaries and Notation**  We consider the linear system $Ax_* = b$, where $A \in \mathbb{R}^{n \times n}_{\text{sym}}$ is symmetric positive definite. The random variables $\mathsf{A}$, $\mathsf{H}$ and $\mathsf{x}$ model the linear operator $A$, its inverse $H = A^{-1}$ and the solution $x_*$. Algorithm 1 chooses actions $S = [s_1, \dots, s_k] \in \mathbb{R}^{n \times k}$ given by its policy $\pi(s \mid \mathsf{A}, \mathsf{H}, \mathsf{x}, A, b)$ and computes observations $Y = [y_1, \dots, y_k] \in \mathbb{R}^{n \times k}$ given by a linear projection $y_i = As_i$ in each iteration $0 < i \le k$.

## S1  Probabilistic Modelling of Deterministic Problems

At first glance it might seem counterintuitive to frame a numerical problem in the language of probability theory. After all, when considering the exact problem $Ax_* = b$ all quantities involved $A, x_*$, and $b$ are deterministic. However, the distribution of the random variables $\mathsf{A}, \mathsf{H}$ and $\mathsf{x}$ represents *epistemic uncertainty* arising from finite computational resources. With a finite budget only a limited amount of information can be obtained about $A$ (e.g. via matrix-vector products). In particular, for a sufficiently large problem a priori the inverse $H = A^{-1}$ and the solution $x_*$, while deterministic and computable in finite time, are not known. This uncertainty about the inverse is captured by the prior distribution of $\mathsf{H}$. In the Bayesian framework the belief about the inverse $\mathsf{H}$ is then iteratively updated given new observations $y_i = As_i$.

The motivation for also estimating $A$ becomes clear if one considers the following. Usually in large-scale applications, the matrix $A$ is never actually formed in memory due to computational constraints. Instead only the matrix-vector product $v \mapsto Av$ is available. Therefore without further computation, the value of any given matrix entry $A_{ij}$ is in fact uncertain. Further, generally other properties of the matrix $A$ such as its eigenspectrum are also not readily available. The probabilistic framework provides a principled way of incorporating prior knowledge about $A$ and makes assumptions about the problem explicit. Relating the prior model $\mathsf{A}$ and $\mathsf{H}$ is important here to allow Algorithm 1 to take such prior information into account in its policy. Finally, the strongest argument for a model $\mathsf{A}$ may yet be the incorporation of noise. Suppose we only have access to $y_i = (A + E_i)s_i$ with

additive noise $\boldsymbol{E}_i$. This is a common occurrence in application, where the linear system to be solved arises from an approximation itself or if $\boldsymbol{A}$ is constructed from data. Concrete examples are batched empirical risk minimization problems or stochastic quadratic optimization. In this setting the probabilistic linear solver must estimate the true $\boldsymbol{A}$ via its observations.

The application of probabilistic inference to numerical problems goes back well into the last century [1–3] and has recently seen a resurgence in research interest in the form of *probabilistic numerics*. Overviews discussing motivations and historical perspectives can be found in Hennig et al. [4] and Oates and Sullivan [5]. Hennig [6] gives additional insight into the statistical interpretation of linear systems.

## S2 The Kronecker Product and its Variants

We will now introduce different types of Kronecker products needed for constructing covariances for matrix-variate distributions. In order to transfer results from probabilistic modelling of vector-variate random variables to the matrix-variate case, we need two types of vectorization operations, i.e. bijections between spaces of matrices and vector spaces.

Let $\text{vec} : \mathbb{R}^{m \times n} \to \mathbb{R}^{mn}$, denote the *column-wise stacking operator* [7], defined as

$$\text{vec}(\boldsymbol{X}) = (X_{11}, X_{21}, \dots, X_{m1}, X_{12}, \dots, X_{mn})^\intercal \in \mathbb{R}^{mn}.$$

Further, define $\text{svec} : \mathbb{R}^{n \times n}_{\text{sym}} \to \mathbb{R}^{\frac{1}{2}n(n+1)}$, the *column-wise symmetric stacking operator* [8] given by

$$\text{svec}(\boldsymbol{X}) = (X_{11}, \sqrt{2}X_{21}, \dots, \sqrt{2}X_{n1}, X_{22}, \sqrt{2}X_{32}, \dots, \sqrt{2}X_{n2}, \dots, X_{nn})^\intercal \in \mathbb{R}^{\frac{1}{2}n(n+1)}.$$

To translate between the two representations following Schäcke [9] we also define the matrix $\boldsymbol{Q} \in \mathbb{R}^{\frac{1}{2}n(n+1) \times n^2}$ such that for all symmetric matrices $\boldsymbol{X} \in \mathbb{R}^{n \times n}_{\text{sym}}$, we have $\boldsymbol{Q}\,\text{vec}(\boldsymbol{X}) = \text{svec}(\boldsymbol{X})$ and $\text{vec}(\boldsymbol{X}) = \boldsymbol{Q}^\intercal\,\text{svec}(\boldsymbol{X})$. Note, that $\boldsymbol{Q}$ has orthonormal rows, i.e. $\boldsymbol{Q}\boldsymbol{Q}^\intercal = \boldsymbol{I}$. For convenience we also name the inverse operations $\text{mat} := \text{vec}^{-1}$ and $\text{smat} := \text{svec}^{-1}$.

### S2.1 Kronecker Product

We make extensive use of Kronecker-type structures for covariance matrices of matrix-variate distributions in this paper. The *Kronecker product* $\boldsymbol{A} \otimes \boldsymbol{B}$ [10] of two matrices $\boldsymbol{A} \in \mathbb{R}^{m_1 \times n_1}$ and $\boldsymbol{B} \in \mathbb{R}^{m_2 \times n_2}$ is given by

$$\boldsymbol{A} \otimes \boldsymbol{B} = \begin{pmatrix} \boldsymbol{A}_{11}\boldsymbol{B} & \dots & \boldsymbol{A}_{1n_1}\boldsymbol{B} \\ \vdots & \ddots & \vdots \\ \boldsymbol{A}_{m_11}\boldsymbol{B} & \dots & \boldsymbol{A}_{m_1n_1}\boldsymbol{B} \end{pmatrix} \in \mathbb{R}^{(m_1m_2) \times (n_1n_2)}$$

The Kronecker product satisfies the characteristic property

$$(\boldsymbol{A} \otimes \boldsymbol{B})\,\text{vec}(\boldsymbol{X}) = \text{vec}(\boldsymbol{B}\boldsymbol{X}\boldsymbol{A}^\intercal), \tag{S1}$$

for $\boldsymbol{X} \in \mathbb{R}^{n_2 \times n_1}$. Characteristic properties of Kronecker-type products are useful to turn matrix equations into vector equations. We state a set of properties of the Kronecker product next without proof. More detail on Kronecker products can be found in Van Loan [10].

**Proposition S1** (Properties of the Kronecker Product [10])
*The Kronecker product satisfies the following identities:*

$$\exists \boldsymbol{A}, \boldsymbol{B} : \boldsymbol{A} \otimes \boldsymbol{B} \neq \boldsymbol{B} \otimes \boldsymbol{A} \tag{S2}$$

$$(\boldsymbol{A} \otimes \boldsymbol{B})^\intercal = \boldsymbol{A}^\intercal \otimes \boldsymbol{B}^\intercal \tag{S3}$$

$$(\boldsymbol{A} \otimes \boldsymbol{B})^{-1} = \boldsymbol{A}^{-1} \otimes \boldsymbol{B}^{-1} \tag{S4}$$

$$(\boldsymbol{A} + \boldsymbol{B}) \otimes \boldsymbol{C} = \boldsymbol{A} \otimes \boldsymbol{C} + \boldsymbol{B} \otimes \boldsymbol{C} \tag{S5}$$

$$(\boldsymbol{A} \otimes \boldsymbol{B})(\boldsymbol{C} \otimes \boldsymbol{D}) = (\boldsymbol{A}\boldsymbol{C}) \otimes (\boldsymbol{B}\boldsymbol{D}) \tag{S6}$$

$$\text{tr}(\boldsymbol{A} \otimes \boldsymbol{B}) = \text{tr}(\boldsymbol{A})\,\text{tr}(\boldsymbol{B}) \tag{S7}$$

$$\boldsymbol{A} \in \mathbb{R}^{m \times m}_{\text{sym}}, \boldsymbol{B} \in \mathbb{R}^{n \times n}_{\text{sym}} \implies \boldsymbol{A} \otimes \boldsymbol{B} \in \mathbb{R}^{mn \times mn}_{\text{sym}} \tag{S8}$$

$$\boldsymbol{A} \otimes \boldsymbol{B} = (\boldsymbol{L}_A\boldsymbol{L}_A^\intercal) \otimes (\boldsymbol{L}_B\boldsymbol{L}_B^\intercal) = (\boldsymbol{L}_A \otimes \boldsymbol{L}_B)(\boldsymbol{L}_A^\intercal \otimes \boldsymbol{L}_B^\intercal) \tag{S9}$$

$$\boldsymbol{A} \otimes \boldsymbol{B} = (\boldsymbol{U}_A\boldsymbol{\Lambda}_A\boldsymbol{U}_A^\intercal) \otimes (\boldsymbol{U}_B\boldsymbol{\Lambda}_B\boldsymbol{U}_B^\intercal) = (\boldsymbol{U}_A \otimes \boldsymbol{U}_B)(\boldsymbol{\Lambda}_A \otimes \boldsymbol{\Lambda}_B)(\boldsymbol{U}_A^\intercal \otimes \boldsymbol{U}_B^\intercal) \tag{S10}$$

## S2.2 Box Product

The *box product* $\boldsymbol{A} \boxtimes \boldsymbol{B} \in \mathbb{R}^{(m_1 m_2) \times (n_1 n_2)}$ can be defined via its characteristic property

$$(\boldsymbol{A} \boxtimes \boldsymbol{B}) \operatorname{vec}(\boldsymbol{Y}) = \operatorname{vec}(\boldsymbol{B} \boldsymbol{Y}^{\mathsf{T}} \boldsymbol{A}^{\mathsf{T}}) \tag{S11}$$

for $\boldsymbol{Y} \in \mathbb{R}^{n_1 \times n_2}$. See also Olsen et al. [11] for details.

**Proposition S2** (Properties of the Box Product [11])
*The box product satisfies the following identities:*

$$\exists \boldsymbol{A}, \boldsymbol{B} : \boldsymbol{A} \boxtimes \boldsymbol{B} \neq \boldsymbol{B} \boxtimes \boldsymbol{A} \tag{S12}$$

$$(\boldsymbol{A} \boxtimes \boldsymbol{B})^{\mathsf{T}} = \boldsymbol{B}^{\mathsf{T}} \boxtimes \boldsymbol{A}^{\mathsf{T}} \tag{S13}$$

$$(\boldsymbol{A} \boxtimes \boldsymbol{B})^{-1} = \boldsymbol{B}^{-1} \boxtimes \boldsymbol{A}^{-1} \tag{S14}$$

$$(\boldsymbol{A} + \boldsymbol{B}) \boxtimes \boldsymbol{C} = \boldsymbol{A} \boxtimes \boldsymbol{C} + \boldsymbol{B} \boxtimes \boldsymbol{C} \tag{S15}$$

$$(\boldsymbol{A} \boxtimes \boldsymbol{B})(\boldsymbol{C} \boxtimes \boldsymbol{D}) = (\boldsymbol{A} \boldsymbol{D}) \otimes (\boldsymbol{B} \boldsymbol{C}) \tag{S16}$$

$$(\boldsymbol{A} \boxtimes \boldsymbol{B})(\boldsymbol{C} \otimes \boldsymbol{D}) = (\boldsymbol{A} \boldsymbol{D}) \boxtimes (\boldsymbol{B} \boldsymbol{C}) \tag{S17}$$

$$(\boldsymbol{A} \otimes \boldsymbol{B})(\boldsymbol{C} \boxtimes \boldsymbol{D}) = (\boldsymbol{A} \boldsymbol{C}) \boxtimes (\boldsymbol{B} \boldsymbol{D}) \tag{S18}$$

$$\operatorname{tr}(\boldsymbol{A} \boxtimes \boldsymbol{B}) = \operatorname{tr}(\boldsymbol{A} \boldsymbol{B}) \tag{S19}$$

## S2.3 Symmetric Kronecker Product

The *symmetric Kronecker product* $\boldsymbol{A} \otimes \boldsymbol{B}$ of two square matrices $\boldsymbol{A}, \boldsymbol{B} \in \mathbb{R}^{n \times n}$ is defined via its characteristic property for $\boldsymbol{X} \in \mathbb{R}^{n \times n}_{\text{sym}}$ as

$$(\boldsymbol{A} \otimes \boldsymbol{B}) \operatorname{svec}(\boldsymbol{X}) = \frac{1}{2} \operatorname{svec}(\boldsymbol{B} \boldsymbol{X} \boldsymbol{A}^{\mathsf{T}} + \boldsymbol{A} \boldsymbol{X} \boldsymbol{B}^{\mathsf{T}}) \tag{S20}$$

or equivalently

$$\boldsymbol{A} \otimes \boldsymbol{B} = \frac{1}{2} \boldsymbol{Q} (\boldsymbol{A} \otimes \boldsymbol{B} + \boldsymbol{B} \otimes \boldsymbol{A}) \boldsymbol{Q}^{\mathsf{T}}.$$

**Proposition S3** (Properties of the Symmetric Kronecker Product [8, 9])
*The symmetric Kronecker product satisfies the following identities:*

$$\boldsymbol{A} \otimes \boldsymbol{B} = \boldsymbol{B} \otimes \boldsymbol{A} \tag{S21}$$

$$(\boldsymbol{A} \otimes \boldsymbol{B})^{\mathsf{T}} = \boldsymbol{A}^{\mathsf{T}} \otimes \boldsymbol{B}^{\mathsf{T}} \tag{S22}$$

$$(\boldsymbol{A} \otimes \boldsymbol{A})^{-1} = \boldsymbol{A}^{-1} \otimes \boldsymbol{A}^{-1} \tag{S23}$$

$$(\boldsymbol{A} + \boldsymbol{B}) \otimes \boldsymbol{C} = \boldsymbol{A} \otimes \boldsymbol{C} + \boldsymbol{B} \otimes \boldsymbol{C} \tag{S24}$$

$$(\boldsymbol{A} \otimes \boldsymbol{B})(\boldsymbol{C} \otimes \boldsymbol{D}) = \frac{1}{2}(\boldsymbol{A} \boldsymbol{C} \otimes \boldsymbol{B} \boldsymbol{D} + \boldsymbol{A} \boldsymbol{D} \otimes \boldsymbol{B} \boldsymbol{C}) \tag{S25}$$

$$\boldsymbol{A} \in \mathbb{R}^{n \times n}_{\text{sym}}, \boldsymbol{B} \in \mathbb{R}^{n \times n}_{\text{sym}} \implies \boldsymbol{A} \otimes \boldsymbol{B} \in \mathbb{R}^{\frac{1}{2}n(n+1) \times \frac{1}{2}n(n+1)}_{\text{sym}} \tag{S26}$$

$$\boldsymbol{A} \otimes \boldsymbol{A} = (\boldsymbol{L}_A \boldsymbol{L}_A^{\mathsf{T}}) \otimes (\boldsymbol{L}_A \boldsymbol{L}_A^{\mathsf{T}}) = (\boldsymbol{L}_A \otimes \boldsymbol{L}_A)(\boldsymbol{L}_A^{\mathsf{T}} \otimes \boldsymbol{L}_A^{\mathsf{T}}) \tag{S27}$$

$$\boldsymbol{A} \otimes \boldsymbol{A} = (\boldsymbol{U}_A \boldsymbol{\Lambda}_A \boldsymbol{U}_A^{\mathsf{T}}) \otimes (\boldsymbol{U}_A \boldsymbol{\Lambda}_A \boldsymbol{U}_A^{\mathsf{T}}) = (\boldsymbol{U}_A \otimes \boldsymbol{U}_A)(\boldsymbol{\Lambda}_A \otimes \boldsymbol{\Lambda}_A)(\boldsymbol{U}_A^{\mathsf{T}} \otimes \boldsymbol{U}_A^{\mathsf{T}}) \tag{S28}$$

*Note, that the symmetric Kronecker product represented as a $\frac{1}{2}n(n+1) \times \frac{1}{2}n(n+1)$ matrix is in general not symmetric.*

Further properties can be found in Alizadeh et al. [8] and Schäcke [9]. We prove the following technical results for mixed expressions of Kronecker-type products, which we will make use of later.

**Corollary S1** (Mixed Kronecker Product Identities)
*Let $\boldsymbol{A} \in \mathbb{R}^{n \times n}_{\text{sym}}$, $\boldsymbol{B}, \boldsymbol{C} \in \mathbb{R}^{n \times k}$ and $\boldsymbol{X} \in \mathbb{R}^{k \times k}$ such that $(\boldsymbol{C} \boldsymbol{X} \boldsymbol{B}^{\mathsf{T}})^{\mathsf{T}} = \boldsymbol{C} \boldsymbol{X} \boldsymbol{B}^{\mathsf{T}}$, then it holds that*

$$\boldsymbol{Q}^{\mathsf{T}}(\boldsymbol{A} \otimes \boldsymbol{A}) \boldsymbol{Q}(\boldsymbol{B} \otimes \boldsymbol{C}) \operatorname{vec}(\boldsymbol{X}) = \frac{1}{2}(\boldsymbol{A} \boldsymbol{B} \otimes \boldsymbol{A} \boldsymbol{C} + \boldsymbol{A} \boldsymbol{C} \boxtimes \boldsymbol{A} \boldsymbol{B}) \operatorname{vec}(\boldsymbol{X}) \tag{S29}$$

$$(\boldsymbol{B}^{\mathsf{T}} \otimes \boldsymbol{C}^{\mathsf{T}}) \boldsymbol{Q}^{\mathsf{T}}(\boldsymbol{A} \otimes \boldsymbol{A}) \boldsymbol{Q} = \frac{1}{2}(\boldsymbol{B}^{\mathsf{T}} \boldsymbol{A} \otimes \boldsymbol{C}^{\mathsf{T}} \boldsymbol{A} + \boldsymbol{B}^{\mathsf{T}} \boldsymbol{A} \boxtimes \boldsymbol{C}^{\mathsf{T}} \boldsymbol{A}). \tag{S30}$$

$$(\boldsymbol{B}^{\mathsf{T}} \otimes \boldsymbol{C}^{\mathsf{T}}) \boldsymbol{Q}^{\mathsf{T}}(\boldsymbol{A} \otimes \boldsymbol{A}) \boldsymbol{Q}(\boldsymbol{B} \otimes \boldsymbol{C}) \operatorname{vec}(\boldsymbol{X}) = \frac{1}{2}(\boldsymbol{B}^{\mathsf{T}} \boldsymbol{A} \boldsymbol{B} \otimes \boldsymbol{C}^{\mathsf{T}} \boldsymbol{A} \boldsymbol{C} + \boldsymbol{B}^{\mathsf{T}} \boldsymbol{A} \boldsymbol{C} \boxtimes \boldsymbol{C}^{\mathsf{T}} \boldsymbol{A} \boldsymbol{B}) \operatorname{vec}(\boldsymbol{X}). \tag{S31}$$

*Now, assume $\boldsymbol{A}$ to be invertible,* $\operatorname{rank}(\boldsymbol{C}) = k$ *and* $\boldsymbol{Y} \in \mathbb{R}^{k \times n}$ *such that* $(\boldsymbol{Y}\boldsymbol{C})^\mathsf{T} = \boldsymbol{Y}\boldsymbol{C}$, *then for*

$$\boldsymbol{G} = (\boldsymbol{I}_n \otimes \boldsymbol{C}^\mathsf{T})\boldsymbol{Q}^\mathsf{T}(\boldsymbol{A} \circledast \boldsymbol{A})\boldsymbol{Q}(\boldsymbol{I}_n \otimes \boldsymbol{C})$$

$$\boldsymbol{G}_{\mathrm{right}}^{-1} = (2\boldsymbol{A}^{-1} - \boldsymbol{C}(\boldsymbol{C}^\mathsf{T}\boldsymbol{A}\boldsymbol{C})^{-1}\boldsymbol{C}^\mathsf{T}) \otimes (\boldsymbol{C}^\mathsf{T}\boldsymbol{A}\boldsymbol{C})^{-1}$$

*we have* $\boldsymbol{G}\boldsymbol{G}_{\mathrm{right}}^{-1} \operatorname{vec}(\boldsymbol{Y}) = \operatorname{vec}(\boldsymbol{Y})$, *i.e.* $\boldsymbol{G}_{\mathrm{right}}^{-1}$ *is the right inverse of* $\boldsymbol{G}$. *Finally, for* $\boldsymbol{D}, \boldsymbol{E} \in \mathbb{R}^{n \times n}$ *and* $\boldsymbol{Z} \in \mathbb{R}_{\mathrm{sym}}^{n \times n}$ *such that* $(\boldsymbol{E}\boldsymbol{A}\boldsymbol{Z}\boldsymbol{A}\boldsymbol{D}^\mathsf{T})^\mathsf{T} = \boldsymbol{E}\boldsymbol{A}\boldsymbol{Z}\boldsymbol{A}\boldsymbol{D}^\mathsf{T}$, *we have*

$$(\boldsymbol{A}^\mathsf{T} \circledast \boldsymbol{A}^\mathsf{T})\boldsymbol{Q}(\boldsymbol{D} \otimes \boldsymbol{E})\boldsymbol{Q}^\mathsf{T}(\boldsymbol{A} \circledast \boldsymbol{A}) \operatorname{svec}(\boldsymbol{Z}) = (\boldsymbol{A}^\mathsf{T}\boldsymbol{D}\boldsymbol{A}) \circledast (\boldsymbol{A}^\mathsf{T}\boldsymbol{E}\boldsymbol{A}) \operatorname{svec}(\boldsymbol{Z}). \qquad \text{(S32)}$$

*Proof.* Let $\boldsymbol{X} \in \mathbb{R}^{k \times k}$ such that $(\boldsymbol{C}\boldsymbol{X}\boldsymbol{B}^\mathsf{T})^\mathsf{T} = \boldsymbol{C}\boldsymbol{X}\boldsymbol{B}^\mathsf{T}$, then

$$\begin{aligned}
\boldsymbol{Q}^\mathsf{T}(\boldsymbol{A} \circledast \boldsymbol{A})\boldsymbol{Q}(\boldsymbol{B} \otimes \boldsymbol{C}) \operatorname{vec}(\boldsymbol{X}) &= \boldsymbol{Q}^\mathsf{T}(\boldsymbol{A} \circledast \boldsymbol{A})\boldsymbol{Q} \operatorname{vec}(\boldsymbol{C}\boldsymbol{X}\boldsymbol{B}^\mathsf{T}) \\
&= \boldsymbol{Q}^\mathsf{T}(\boldsymbol{A} \circledast \boldsymbol{A}) \operatorname{svec}(\boldsymbol{C}\boldsymbol{X}\boldsymbol{B}^\mathsf{T}) \\
&= \boldsymbol{Q}^\mathsf{T} \operatorname{svec}(\boldsymbol{A}\boldsymbol{C}\boldsymbol{X}\boldsymbol{B}^\mathsf{T}\boldsymbol{A}) \\
&= \frac{1}{2} \operatorname{vec}(\boldsymbol{A}\boldsymbol{C}\boldsymbol{X}\boldsymbol{B}^\mathsf{T}\boldsymbol{A} + \boldsymbol{A}\boldsymbol{B}\boldsymbol{X}^\mathsf{T}\boldsymbol{C}^\mathsf{T}\boldsymbol{A}) \\
&= \frac{1}{2}(\boldsymbol{A}\boldsymbol{B} \otimes \boldsymbol{A}\boldsymbol{C} + \boldsymbol{A}\boldsymbol{C} \boxtimes \boldsymbol{A}\boldsymbol{B}),
\end{aligned}$$

further it holds for $\boldsymbol{W} \in \mathbb{R}_{\mathrm{sym}}^{n \times n}$

$$\begin{aligned}
(\boldsymbol{B}^\mathsf{T} \otimes \boldsymbol{C}^\mathsf{T})\boldsymbol{Q}^\mathsf{T}(\boldsymbol{A} \circledast \boldsymbol{A})\boldsymbol{Q} \operatorname{vec}(\boldsymbol{W}) &= (\boldsymbol{B}^\mathsf{T} \otimes \boldsymbol{C}^\mathsf{T})\boldsymbol{Q}^\mathsf{T} \operatorname{svec}(\boldsymbol{A}\boldsymbol{W}\boldsymbol{A}) \\
&= \operatorname{vec}(\boldsymbol{C}^\mathsf{T}\boldsymbol{A}\boldsymbol{W}\boldsymbol{A}\boldsymbol{B}) \\
&= \frac{1}{2}(\boldsymbol{C}^\mathsf{T}\boldsymbol{A}\boldsymbol{W}\boldsymbol{A}\boldsymbol{B} + \boldsymbol{C}^\mathsf{T}\boldsymbol{A}^\mathsf{T}\boldsymbol{W}^\mathsf{T}\boldsymbol{A}^\mathsf{T}\boldsymbol{B}) \\
&= \frac{1}{2}(\boldsymbol{B}^\mathsf{T}\boldsymbol{A} \otimes \boldsymbol{C}^\mathsf{T}\boldsymbol{A} + \boldsymbol{B}^\mathsf{T}\boldsymbol{A} \boxtimes \boldsymbol{C}^\mathsf{T}\boldsymbol{A}),
\end{aligned}$$

and using the properties of the Kronecker and the Box product we obtain

$$\begin{aligned}
(\boldsymbol{B}^\mathsf{T} \otimes \boldsymbol{C}^\mathsf{T})\boldsymbol{Q}^\mathsf{T}(\boldsymbol{A} \circledast \boldsymbol{A})\boldsymbol{Q}(\boldsymbol{B} \otimes \boldsymbol{C}) \operatorname{vec}(\boldsymbol{X}) &= (\boldsymbol{B}^\mathsf{T} \otimes \boldsymbol{C}^\mathsf{T})\frac{1}{2}(\boldsymbol{B}^\mathsf{T}\boldsymbol{A} \otimes \boldsymbol{C}^\mathsf{T}\boldsymbol{A} + \boldsymbol{B}^\mathsf{T}\boldsymbol{A} \boxtimes \boldsymbol{C}^\mathsf{T}\boldsymbol{A}) \operatorname{vec}(\boldsymbol{X}) \\
&= \frac{1}{2}(\boldsymbol{B}^\mathsf{T}\boldsymbol{A} \otimes \boldsymbol{C}^\mathsf{T}\boldsymbol{A} + \boldsymbol{B}^\mathsf{T}\boldsymbol{A} \boxtimes \boldsymbol{C}^\mathsf{T}\boldsymbol{A}) \operatorname{vec}(\boldsymbol{X}).
\end{aligned}$$

Now let $\boldsymbol{A}$ be invertible, let $\boldsymbol{C}$ have full rank and choose $\boldsymbol{Y} \in \mathbb{R}^{k \times n}$ arbitrarily such that $(\boldsymbol{Y}\boldsymbol{C})^\mathsf{T} = \boldsymbol{Y}\boldsymbol{C}$. Then using Proposition S1 and Proposition S2 we obtain

$$\begin{aligned}
&(\boldsymbol{I}_n \otimes \boldsymbol{C}^\mathsf{T})\boldsymbol{Q}^\mathsf{T}(\boldsymbol{A} \circledast \boldsymbol{A})\boldsymbol{Q}(\boldsymbol{I}_n \otimes \boldsymbol{C})(2\boldsymbol{A}^{-1} - \boldsymbol{C}(\boldsymbol{C}^\mathsf{T}\boldsymbol{A}\boldsymbol{C})^{-1}\boldsymbol{C}^\mathsf{T}) \otimes (\boldsymbol{C}^\mathsf{T}\boldsymbol{A}\boldsymbol{C})^{-1} \operatorname{vec}(\boldsymbol{Y}) \\
&= \frac{1}{2}(\boldsymbol{A} \otimes \boldsymbol{C}^\mathsf{T}\boldsymbol{A}\boldsymbol{C} + \boldsymbol{A}\boldsymbol{C} \boxtimes \boldsymbol{C}^\mathsf{T}\boldsymbol{A})(2\boldsymbol{A}^{-1} - \boldsymbol{C}(\boldsymbol{C}^\mathsf{T}\boldsymbol{A}\boldsymbol{C})^{-1}\boldsymbol{C}^\mathsf{T}) \otimes (\boldsymbol{C}^\mathsf{T}\boldsymbol{A}\boldsymbol{C})^{-1}) \operatorname{vec}(\boldsymbol{Y}) \\
&= (\boldsymbol{I}_n \otimes \boldsymbol{I}_k - \frac{1}{2}\boldsymbol{A}\boldsymbol{C}(\boldsymbol{C}^\mathsf{T}\boldsymbol{A}\boldsymbol{C})^{-1}\boldsymbol{C}^\mathsf{T} \otimes \boldsymbol{I}_k + \boldsymbol{A}\boldsymbol{C}(\boldsymbol{C}^\mathsf{T}\boldsymbol{A}\boldsymbol{C})^{-1} \boxtimes \boldsymbol{C}^\mathsf{T} - \frac{1}{2}\boldsymbol{A}\boldsymbol{C}(\boldsymbol{C}^\mathsf{T}\boldsymbol{A}\boldsymbol{C})^{-1} \boxtimes \boldsymbol{C}^\mathsf{T}) \operatorname{vec}(\boldsymbol{Y}) \\
&= (\boldsymbol{I}_n \otimes \boldsymbol{I}_k - \frac{1}{2}\boldsymbol{A}\boldsymbol{C}(\boldsymbol{C}^\mathsf{T}\boldsymbol{A}\boldsymbol{C})^{-1}\boldsymbol{C}^\mathsf{T} \otimes \boldsymbol{I}_k + \frac{1}{2}\boldsymbol{A}\boldsymbol{C}(\boldsymbol{C}^\mathsf{T}\boldsymbol{A}\boldsymbol{C})^{-1} \boxtimes \boldsymbol{C}^\mathsf{T}) \operatorname{vec}(\boldsymbol{Y}) \\
&= \operatorname{vec}(\boldsymbol{Y}) - \frac{1}{2}(\boldsymbol{Y}\boldsymbol{C}(\boldsymbol{C}^\mathsf{T}\boldsymbol{A}\boldsymbol{C})^{-1}\boldsymbol{C}^\mathsf{T}\boldsymbol{A} - \boldsymbol{C}^\mathsf{T}\boldsymbol{Y}^\mathsf{T}(\boldsymbol{C}^\mathsf{T}\boldsymbol{A}\boldsymbol{C})^{-1}\boldsymbol{C}^\mathsf{T}\boldsymbol{A}) \\
&= \operatorname{vec}(\boldsymbol{Y})
\end{aligned}$$

Lastly, by assumption it holds that

$$\begin{aligned}
(\boldsymbol{A}^\mathsf{T} \circledast \boldsymbol{A}^\mathsf{T})\boldsymbol{Q}(\boldsymbol{D} \otimes \boldsymbol{E})\boldsymbol{Q}^\mathsf{T}(\boldsymbol{A} \circledast \boldsymbol{A}) \operatorname{svec}(\boldsymbol{Z}) &= (\boldsymbol{A} \circledast \boldsymbol{A})\boldsymbol{Q} \operatorname{vec}(\boldsymbol{E}\boldsymbol{A}\boldsymbol{Z}\boldsymbol{A}\boldsymbol{D}^\mathsf{T}) \\
&= \operatorname{svec}(\boldsymbol{A}\boldsymbol{E}\boldsymbol{A}\boldsymbol{Z}\boldsymbol{A}\boldsymbol{D}^\mathsf{T}\boldsymbol{A}) \\
&= \frac{1}{2}(\boldsymbol{A}\boldsymbol{E}\boldsymbol{A}\boldsymbol{Z}\boldsymbol{A}\boldsymbol{D}^\mathsf{T}\boldsymbol{A} + \boldsymbol{A}\boldsymbol{D}\boldsymbol{A}\boldsymbol{Z}\boldsymbol{A}\boldsymbol{E}^\mathsf{T}\boldsymbol{A}) \\
&= (\boldsymbol{A}\boldsymbol{D}\boldsymbol{A} \circledast \boldsymbol{A}\boldsymbol{E}\boldsymbol{A}) \operatorname{svec}(\boldsymbol{Z}).
\end{aligned}$$

This concludes the proof. $\qquad\qquad \square$

## S3 The Matrix-variate Normal Distribution

In order for our probabilistic linear solvers to infer the true latent $\boldsymbol{A}$ or its inverse $\boldsymbol{H} = \boldsymbol{A}^{-1}$, we need a distribution expressing the belief of the solver over those latent quantities at any given point. A Gaussian distribution over matrices will play this role, motivated by the linear nature of the observations. This section closely follows Gupta and Nagar [12].

**Definition S1** (Matrix-variate Normal Distribution [12])
Let $\boldsymbol{X}_0 \in \mathbb{R}^{m \times n}$ and let $\boldsymbol{V} \in \mathbb{R}^m_{\text{sym}}$ and $\boldsymbol{W} \in \mathbb{R}^{n \times n}_{\text{sym}}$ be positive-definite. We say a random matrix $\mathbf{X}$ has a *matrix-variate normal distribution* with mean $\boldsymbol{X}_0$ and covariance $\boldsymbol{V} \otimes \boldsymbol{W}$, iff

$$\text{vec}(\mathbf{X}^\intercal) \sim \mathcal{N}_{mn}(\text{vec}(\boldsymbol{X}_0^\intercal), \boldsymbol{V} \otimes \boldsymbol{W}).$$

We write as a shorthand $\mathbf{X} \sim \mathcal{N}(\boldsymbol{X}_0, \boldsymbol{V} \otimes \boldsymbol{W})$.

Note, that the matrices $\boldsymbol{V}$ and $\boldsymbol{W}$ represent the covariance between rows and columns of $\mathbf{X}$, respectively. Since we model symmetric matrices in this work, we also introduce a Gaussian distribution over $\mathbb{R}^{n \times n}_{\text{sym}}$.

**Definition S2** (Symmetric Matrix-variate Normal Distribution [12])
Let $\boldsymbol{X}_0, \boldsymbol{W} \in \mathbb{R}^{n \times n}_{\text{sym}}$ such that $\boldsymbol{W}$ is positive-definite, then the random matrix $\mathbf{X}$ has a *symmetric matrix-variate normal distribution*, iff

$$\text{svec}(\mathbf{X}) \sim \mathcal{N}_{\frac{1}{2} n(n+1)}(\text{svec}(\boldsymbol{X}_0), \boldsymbol{W} \circledast \boldsymbol{W}).$$

We write $\mathbf{X} \sim \mathcal{N}(\boldsymbol{X}_0, \boldsymbol{W} \circledast \boldsymbol{W})$.

It follows immediately from the definition that realizations of a symmetric matrix-variate normal distribution are symmetric matrices. This distribution also emerges naturally by conditioning a matrix-variate normal distribution on the linear constraint $\mathbf{X} = \mathbf{X}^\intercal$.

## S4 Probabilistic Linear Solvers

Probabilistic linear solvers (PLS) [6, 13, 14] infer posterior beliefs over the matrix $\boldsymbol{A}$, its inverse $\boldsymbol{H}$ or the solution $\boldsymbol{x}_* = \boldsymbol{H}\boldsymbol{b}$ of a linear system via linear observations $\boldsymbol{Y} = \boldsymbol{A}\boldsymbol{S}$. We consider matrix-based inference [14] in this work. Assuming a prior $p(\mathbf{A})$ or $p(\mathbf{H})$, actions $\boldsymbol{S}$ and linear observations $\boldsymbol{Y}$ such methods return posterior distributions $p(\mathbf{A} \mid \boldsymbol{S}, \boldsymbol{Y})$ or $p(\mathbf{H} \mid \boldsymbol{S}, \boldsymbol{Y})$.

### S4.1 Matrix-based Inference

The generic matrix-based inference procedure of probabilistic linear solvers is a consequence of the matrix-variate version of the following standard result for Gaussian inference under linear observations.

**Theorem S1** (Linear Gaussian Inference [15])
*Let $\boldsymbol{v} \sim \mathcal{N}(\boldsymbol{\mu}, \boldsymbol{\Sigma})$, where $\boldsymbol{\mu} \in \mathbb{R}^n$ and $\boldsymbol{\Sigma} \in \mathbb{R}^{n \times n}_{\text{sym}}$ positive-definite, and assume we are given observations of the form*

$$\boldsymbol{B}\boldsymbol{v} + \boldsymbol{b} = \boldsymbol{y} \in \mathbb{R}^m,$$

*where $\boldsymbol{B} \in \mathbb{R}^{m \times n}$ and $\boldsymbol{b} \in \mathbb{R}^m$. Assuming a Gaussian likelihood*

$$p(\boldsymbol{y} \mid \boldsymbol{B}, \boldsymbol{v}, \boldsymbol{b}) = \mathcal{N}(\boldsymbol{y}; \boldsymbol{B}\boldsymbol{v} + \boldsymbol{b}, \boldsymbol{\Lambda}),$$

*for $\boldsymbol{\Lambda} \in \mathbb{R}^m_{\text{sym}}$ positive definite, results in the posterior distribution*

$$\begin{aligned} p(\boldsymbol{v} \mid \boldsymbol{y}, \boldsymbol{B}, \boldsymbol{b}) = \mathcal{N}\big(&\boldsymbol{v}; \boldsymbol{\mu} + \boldsymbol{\Sigma}\boldsymbol{B}^\intercal(\boldsymbol{B}\boldsymbol{\Sigma}\boldsymbol{B}^\intercal + \boldsymbol{\Lambda})^{-1}(\boldsymbol{y} - \boldsymbol{B}\boldsymbol{\mu} - \boldsymbol{b}), \\ &\boldsymbol{\Sigma} - \boldsymbol{\Sigma}\boldsymbol{B}^\intercal(\boldsymbol{B}\boldsymbol{\Sigma}\boldsymbol{B}^\intercal + \boldsymbol{\Lambda})^{-1}\boldsymbol{B}\boldsymbol{\Sigma}\big). \end{aligned}$$

*Further, the marginal distribution of $\boldsymbol{y}$ is given by*

$$p(\boldsymbol{y}) = \mathcal{N}(\boldsymbol{y}; \boldsymbol{B}\boldsymbol{\mu} + \boldsymbol{b}, \boldsymbol{B}\boldsymbol{\Sigma}\boldsymbol{B}^\intercal + \boldsymbol{\Lambda}).$$

### S4.1.1 Asymmetric Model

**Corollary S2** (Asymmetric matrix-based Gaussian Inference [16, 6, 14])
*Assume a prior $p(\mathbf{A}) = \mathcal{N}(\mathbf{A}; \boldsymbol{A}_0, \boldsymbol{V}_0 \otimes \boldsymbol{W}_0)$ and exact observations of the form $\boldsymbol{Y} = \boldsymbol{AS}$, corresponding to a Dirac likelihood $p(\boldsymbol{Y} \mid \mathbf{A}, \boldsymbol{S}) = \delta(\boldsymbol{Y} - \boldsymbol{AS})$, then the posterior $p(\mathbf{A} \mid \boldsymbol{S}, \boldsymbol{Y}) = \mathcal{N}(\mathbf{A}; \boldsymbol{A}_k, \boldsymbol{\Sigma}_k)$ is given by*

$$\boldsymbol{A}_k = \boldsymbol{A}_0 + \boldsymbol{\Delta}_0 \boldsymbol{U}^\mathsf{T}$$
$$\boldsymbol{\Sigma}_k = \boldsymbol{V}_0 \otimes \boldsymbol{W}_0(\boldsymbol{I}_n - \boldsymbol{SU}^\mathsf{T})$$

*where $\boldsymbol{\Delta}_0 = \boldsymbol{Y} - \boldsymbol{A}_0\boldsymbol{S}$ and $\boldsymbol{U} = \boldsymbol{W}_0\boldsymbol{S}(\boldsymbol{S}^\mathsf{T}\boldsymbol{W}_0\boldsymbol{S})^{-1}$.*

*Proof.* In vectorized form the likelihood is given by

$$p(\mathrm{vec}(\boldsymbol{Y}^\mathsf{T}) \mid \mathrm{vec}(\mathbf{A}^\mathsf{T}), \mathrm{vec}(\boldsymbol{S}^\mathsf{T})) = \delta(\mathrm{vec}(\boldsymbol{Y}^\mathsf{T}) - \mathrm{vec}(\boldsymbol{S}^\mathsf{T}\boldsymbol{A}^\mathsf{T})) = \delta(\mathrm{vec}(\boldsymbol{Y}^\mathsf{T}) - (\boldsymbol{I} \otimes \boldsymbol{S}^\mathsf{T})\,\mathrm{vec}(\boldsymbol{A}^\mathsf{T}))$$

Using the Definition S1 of the matrix-variate normal distribution, applying Theorem S1 and using property (S6) of the Kronecker product in Proposition S1 leads to

$$\begin{aligned}
\mathrm{vec}(\boldsymbol{A}_k^\mathsf{T}) &= \mathrm{vec}(\boldsymbol{A}_0^\mathsf{T}) + (\boldsymbol{V}_0 \otimes \boldsymbol{W}_0)(\boldsymbol{I} \otimes \boldsymbol{S})((\boldsymbol{I} \otimes \boldsymbol{S}^\mathsf{T})(\boldsymbol{V}_0 \otimes \boldsymbol{W}_0)(\boldsymbol{I} \otimes \boldsymbol{S}))^{-1}(\mathrm{vec}(\boldsymbol{Y}^\mathsf{T}) - (\boldsymbol{I} \otimes \boldsymbol{S}^\mathsf{T})\,\mathrm{vec}(\boldsymbol{A}_0^\mathsf{T})) \\
&= \mathrm{vec}(\boldsymbol{A}_0^\mathsf{T}) + (\boldsymbol{V}_0 \otimes \boldsymbol{W}_0\boldsymbol{S})(\boldsymbol{V}_0 \otimes \boldsymbol{S}^\mathsf{T}\boldsymbol{W}_0\boldsymbol{S})^{-1}\,\mathrm{vec}(\boldsymbol{\Delta}_0^\mathsf{T}) \\
&= \mathrm{vec}(\boldsymbol{A}_0^\mathsf{T}) + (\boldsymbol{I}_n \otimes \boldsymbol{W}_0\boldsymbol{S}(\boldsymbol{S}^\mathsf{T}\boldsymbol{W}_0\boldsymbol{S})^{-1})\,\mathrm{vec}(\boldsymbol{\Delta}_0^\mathsf{T}) \\
&= \mathrm{vec}(\boldsymbol{A}_0^\mathsf{T} + \boldsymbol{U}\boldsymbol{\Delta}_0^\mathsf{T})
\end{aligned}$$

and further analogously, additionally using bilinearity of the Kronecker product, we obtain

$$\begin{aligned}
\boldsymbol{\Sigma}_k &= \boldsymbol{V}_0 \otimes \boldsymbol{W}_0 - (\boldsymbol{V}_0 \otimes \boldsymbol{W}_0)(\boldsymbol{I} \otimes \boldsymbol{S})((\boldsymbol{I} \otimes \boldsymbol{S}^\mathsf{T})(\boldsymbol{V}_0 \otimes \boldsymbol{W}_0)(\boldsymbol{I} \otimes \boldsymbol{S}))^{-1}(\boldsymbol{I} \otimes \boldsymbol{S}^\mathsf{T})(\boldsymbol{V}_0 \otimes \boldsymbol{W}_0) \\
&= \boldsymbol{V}_0 \otimes \boldsymbol{W}_0 - (\boldsymbol{V}_0 \otimes \boldsymbol{W}_0\boldsymbol{S})(\boldsymbol{V}_0 \otimes \boldsymbol{S}^\mathsf{T}\boldsymbol{W}_0\boldsymbol{S})^{-1}(\boldsymbol{V}_0 \otimes \boldsymbol{S}^\mathsf{T}\boldsymbol{W}_0) \\
&= \boldsymbol{V}_0 \otimes \boldsymbol{W}_0 - \boldsymbol{V}_0 \otimes (\boldsymbol{W}_0\boldsymbol{S}(\boldsymbol{S}^\mathsf{T}\boldsymbol{W}_0\boldsymbol{S})^{-1}\boldsymbol{S}^\mathsf{T}\boldsymbol{W}_0) \\
&= \boldsymbol{V}_0 \otimes \boldsymbol{W}_0(\boldsymbol{I} - \boldsymbol{SU}^\mathsf{T}).
\end{aligned}$$

This concludes the proof. $\qquad\qquad\qquad\qquad\qquad\qquad\qquad\qquad\qquad\qquad\qquad\qquad\qquad\square$

### S4.1.2 Symmetric Model

**Corollary S3** (Symmetric Matrix-based Gaussian Inference [16, 6, 14])
*Assume a symmetric prior $p(\mathbf{A}) = \mathcal{N}(\mathbf{A}; \boldsymbol{A}_0, \boldsymbol{W}_0 \otimes \boldsymbol{W}_0)$ and exact observations of the form $\boldsymbol{Y} = \boldsymbol{AS}$, corresponding to a Dirac likelihood $p(\boldsymbol{Y} \mid \mathbf{A}, \boldsymbol{S}) = \delta(\boldsymbol{Y} - \boldsymbol{AS})$, then the posterior $p(\mathbf{A} \mid \boldsymbol{S}, \boldsymbol{Y}) = \mathcal{N}(\mathbf{A}; \boldsymbol{A}_k, \boldsymbol{\Sigma}_k)$ is given by*

$$\boldsymbol{A}_k = \boldsymbol{A}_0 + \boldsymbol{\Delta}_0 \boldsymbol{U}^\mathsf{T} + \boldsymbol{U}\boldsymbol{\Delta}_0^\mathsf{T} - \boldsymbol{U}\boldsymbol{S}^\mathsf{T}\boldsymbol{\Delta}_0\boldsymbol{U}^\mathsf{T} = \boldsymbol{A}_0 + \boldsymbol{U}\boldsymbol{V}^\mathsf{T} + \boldsymbol{V}\boldsymbol{U}^\mathsf{T}$$
$$\boldsymbol{\Sigma}_k = \boldsymbol{W}_0(\boldsymbol{I}_n - \boldsymbol{SU}^\mathsf{T}) \otimes \boldsymbol{W}_0(\boldsymbol{I}_n - \boldsymbol{SU}^\mathsf{T})$$

*where $\boldsymbol{\Delta}_0 = \boldsymbol{Y} - \boldsymbol{A}_0\boldsymbol{S}$, $\boldsymbol{U} = \boldsymbol{W}_0\boldsymbol{S}(\boldsymbol{S}^\mathsf{T}\boldsymbol{W}_0\boldsymbol{S})^{-1}$ and $\boldsymbol{V} = (\boldsymbol{I}_n - \frac{1}{2}\boldsymbol{U}\boldsymbol{S}^\mathsf{T})\boldsymbol{\Delta}_0$.*

*Proof.* A proof can be found in the appendix of Hennig [6]. We rederive it here in our notation. By assumption the likelihood takes the vectorized form

$$p(\mathrm{vec}(\boldsymbol{Y}^\mathsf{T}) \mid \mathrm{svec}(\mathbf{A}), \mathrm{vec}(\boldsymbol{S}^\mathsf{T})) = \delta(\mathrm{vec}(\boldsymbol{Y}^\mathsf{T}) - \mathrm{vec}(\boldsymbol{S}^\mathsf{T}\boldsymbol{A}^\mathsf{T})) = \delta(\mathrm{vec}(\boldsymbol{Y}^\mathsf{T}) - (\boldsymbol{I} \otimes \boldsymbol{S}^\mathsf{T})\boldsymbol{Q}^\mathsf{T}\,\mathrm{svec}(\boldsymbol{A}))$$

Applying Theorem S1 gives

$$\begin{aligned}
\mathrm{svec}(\boldsymbol{A}_k) &= \mathrm{svec}(\boldsymbol{A}_0) + (\boldsymbol{W}_0 \otimes \boldsymbol{W}_0)\boldsymbol{Q}(\boldsymbol{I}_n \otimes \boldsymbol{S})\boldsymbol{G}^{-1}(\mathrm{vec}(\boldsymbol{Y}^\mathsf{T}) - (\boldsymbol{I} \otimes \boldsymbol{S}^\mathsf{T})\boldsymbol{Q}^\mathsf{T}\,\mathrm{svec}(\boldsymbol{A}_0)) \\
&= \mathrm{svec}(\boldsymbol{A}_0) + (\boldsymbol{W}_0 \otimes \boldsymbol{W}_0)\boldsymbol{Q}(\boldsymbol{I}_n \otimes \boldsymbol{S})\boldsymbol{G}^{-1}\,\mathrm{vec}(\boldsymbol{\Delta}_0^\mathsf{T}) \\
\boldsymbol{\Sigma}_k &= \boldsymbol{W}_0 \otimes \boldsymbol{W}_0 - (\boldsymbol{W}_0 \otimes \boldsymbol{W}_0)\boldsymbol{Q}(\boldsymbol{I}_n \otimes \boldsymbol{S})\boldsymbol{G}^{-1}(\boldsymbol{I}_n \otimes \boldsymbol{S}^\mathsf{T})\boldsymbol{Q}^\mathsf{T}(\boldsymbol{W}_0 \otimes \boldsymbol{W}_0),
\end{aligned}$$

where $\boldsymbol{\Delta}_0 = \boldsymbol{Y} - \boldsymbol{A}_0\boldsymbol{S}$ and the Gram matrix is given by

$$\boldsymbol{G} = (\boldsymbol{I}_n \otimes \boldsymbol{S}^\mathsf{T})\boldsymbol{Q}^\mathsf{T}(\boldsymbol{W}_0 \otimes \boldsymbol{W}_0)\boldsymbol{Q}(\boldsymbol{I}_n \otimes \boldsymbol{S}) \in \mathbb{R}^{nk \times nk}.$$

Now since $(\boldsymbol{\Delta}_0^\mathsf{T}\boldsymbol{S})^\mathsf{T} = \boldsymbol{\Delta}_0^\mathsf{T}\boldsymbol{S}$, we have by Corollary S1 that the right inverse of $\boldsymbol{G}$ is given by

$$\boldsymbol{G}_{\mathrm{right}}^{-1} = (2\boldsymbol{W}_0^{-1} - \boldsymbol{S}(\boldsymbol{S}^\mathsf{T}\boldsymbol{W}_0\boldsymbol{S})^{-1}\boldsymbol{S}^\mathsf{T}) \otimes (\boldsymbol{S}^\mathsf{T}\boldsymbol{W}_0\boldsymbol{S})^{-1}$$

and therefore using (S6) and (S29) we obtain

$$\mathrm{svec}(\boldsymbol{A}_k) = \mathrm{svec}(\boldsymbol{A}_0) + (\boldsymbol{W}_0 \otimes \boldsymbol{W}_0)\boldsymbol{Q}(\boldsymbol{I}_n \otimes \boldsymbol{S})\boldsymbol{G}_{\mathrm{right}}^{-1}\mathrm{vec}(\boldsymbol{\Delta}_0^{\mathsf{T}})$$

$$= \mathrm{svec}(\boldsymbol{A}_0) + \boldsymbol{Q}\boldsymbol{Q}^{\mathsf{T}}(\boldsymbol{W}_0 \otimes \boldsymbol{W}_0)\boldsymbol{Q}(2\boldsymbol{W}_0^{-1} - \boldsymbol{S}(\boldsymbol{S}^{\mathsf{T}}\boldsymbol{W}_0\boldsymbol{S})^{-1}\boldsymbol{S}^{\mathsf{T}}) \otimes \boldsymbol{S}(\boldsymbol{S}^{\mathsf{T}}\boldsymbol{W}_0\boldsymbol{S})^{-1}\mathrm{vec}(\boldsymbol{\Delta}_0^{\mathsf{T}})$$

$$= \mathrm{svec}(\boldsymbol{A}_0) + \boldsymbol{Q}\frac{1}{2}\big((2\boldsymbol{I} - \boldsymbol{U}\boldsymbol{S}^{\mathsf{T}}) \otimes \boldsymbol{U} + \boldsymbol{U} \boxtimes (2\boldsymbol{I} - \boldsymbol{U}\boldsymbol{S}^{\mathsf{T}})\big)\mathrm{vec}(\boldsymbol{\Delta}_0^{\mathsf{T}})$$

$$= \mathrm{svec}(\boldsymbol{A}_0) + \mathrm{svec}(\boldsymbol{U}\boldsymbol{\Delta}_0^{\mathsf{T}}(\boldsymbol{I} - \frac{1}{2}\boldsymbol{U}\boldsymbol{S}^{\mathsf{T}})^{\mathsf{T}} + (\boldsymbol{I} - \frac{1}{2}\boldsymbol{U}\boldsymbol{S}^{\mathsf{T}})\boldsymbol{\Delta}_0\boldsymbol{U}^{\mathsf{T}})$$

$$= \mathrm{svec}(\boldsymbol{A}_0 + \boldsymbol{\Delta}_0\boldsymbol{U}^{\mathsf{T}} + \boldsymbol{U}\boldsymbol{\Delta}_0^{\mathsf{T}} - \boldsymbol{U}\boldsymbol{S}^{\mathsf{T}}\boldsymbol{\Delta}_0\boldsymbol{U}^{\mathsf{T}}).$$

Further by definition it holds that

$$\boldsymbol{U}\boldsymbol{V}^{\mathsf{T}} + \boldsymbol{V}\boldsymbol{U}^{\mathsf{T}} = \boldsymbol{U}\boldsymbol{\Delta}_0^{\mathsf{T}}(\boldsymbol{I}_n - \frac{1}{2}\boldsymbol{S}\boldsymbol{U}^{\mathsf{T}}) + (\boldsymbol{I}_n - \frac{1}{2}\boldsymbol{U}\boldsymbol{S}^{\mathsf{T}})\boldsymbol{\Delta}_0\boldsymbol{U}^{\mathsf{T}} = \boldsymbol{\Delta}_0\boldsymbol{U}^{\mathsf{T}} + \boldsymbol{U}\boldsymbol{\Delta}_0^{\mathsf{T}} - \boldsymbol{U}\boldsymbol{S}^{\mathsf{T}}\boldsymbol{\Delta}_0\boldsymbol{U}^{\mathsf{T}}.$$

For the covariance we obtain using the right inverse of the Gram matrix and (S32) that

$$\boldsymbol{\Sigma}_k = \boldsymbol{W}_0 \otimes \boldsymbol{W}_0 - (\boldsymbol{W}_0 \otimes \boldsymbol{W}_0)\boldsymbol{Q}(\boldsymbol{I}_n \otimes \boldsymbol{S})\boldsymbol{G}^{-1}(\boldsymbol{I}_n \otimes \boldsymbol{S}^{\mathsf{T}})\boldsymbol{Q}^{\mathsf{T}}(\boldsymbol{W}_0 \otimes \boldsymbol{W}_0)$$

$$= \boldsymbol{W}_0 \otimes \boldsymbol{W}_0 - (2\boldsymbol{W}_0 - \boldsymbol{W}_0\boldsymbol{S}(\boldsymbol{S}^{\mathsf{T}}\boldsymbol{W}_0\boldsymbol{S})^{-1}\boldsymbol{S}^{\mathsf{T}}\boldsymbol{W}_0) \otimes (\boldsymbol{W}_0\boldsymbol{S}(\boldsymbol{S}^{\mathsf{T}}\boldsymbol{W}_0\boldsymbol{S})^{-1}\boldsymbol{S}^{\mathsf{T}}\boldsymbol{W}_0)$$

$$= (\boldsymbol{W}_0 - \boldsymbol{W}_0\boldsymbol{S}(\boldsymbol{S}^{\mathsf{T}}\boldsymbol{W}_0\boldsymbol{S})^{-1}\boldsymbol{S}^{\mathsf{T}}\boldsymbol{W}_0) \otimes (\boldsymbol{W}_0 - \boldsymbol{W}_0\boldsymbol{S}(\boldsymbol{S}^{\mathsf{T}}\boldsymbol{W}_0\boldsymbol{S})^{-1}\boldsymbol{S}^{\mathsf{T}}\boldsymbol{W}_0)$$

$$= \boldsymbol{W}_0(\boldsymbol{I}_n - \boldsymbol{S}\boldsymbol{U}^{\mathsf{T}}) \otimes \boldsymbol{W}_0(\boldsymbol{I}_n - \boldsymbol{S}\boldsymbol{U}^{\mathsf{T}}).$$

$$\square$$

## S4.2  Matrix-variate Prior Construction

From a practical point of view it is important to be able to construct a prior for **A** and **H** from an initial guess $\boldsymbol{x}_0$ for the solution. This reduces down to finding $\boldsymbol{A}_0$ and $\boldsymbol{H}_0$ symmetric positive definite, such that $\boldsymbol{A}_0 = \boldsymbol{H}_0^{-1}$ and $\boldsymbol{x}_0 = \boldsymbol{H}_0\boldsymbol{b}$ for the covariance class derived in Section 3. We provide a computationally efficient construction of such a prior here.

**Proposition S4**
*Let $\boldsymbol{x}_0 \in \mathbb{R}^n$ and $\boldsymbol{b} \in \mathbb{R}^n \setminus \{0\}$. Assume $\boldsymbol{x}_0^{\mathsf{T}}\boldsymbol{b} > 0$, then for $\alpha < \frac{\boldsymbol{b}^{\mathsf{T}}\boldsymbol{x}_0}{\boldsymbol{b}^{\mathsf{T}}\boldsymbol{b}}$,*

$$\boldsymbol{H}_0 = \alpha\boldsymbol{I} + \frac{1}{(\boldsymbol{x}_0 - \alpha\boldsymbol{b})^{\mathsf{T}}\boldsymbol{b}}(\boldsymbol{x}_0 - \alpha\boldsymbol{b})(\boldsymbol{x}_0 - \alpha\boldsymbol{b})^{\mathsf{T}}$$

*is symmetric positive definite and $\boldsymbol{H}_0\boldsymbol{b} = \boldsymbol{x}_0$. Further it holds that*

$$\boldsymbol{A}_0 = \boldsymbol{H}_0^{-1} = \alpha^{-1}\boldsymbol{I} - \frac{\alpha^{-1}}{(\boldsymbol{x}_0 - \alpha\boldsymbol{b})^{\mathsf{T}}\boldsymbol{x}_0}(\boldsymbol{x}_0 - \alpha\boldsymbol{b})(\boldsymbol{x}_0 - \alpha\boldsymbol{b})^{\mathsf{T}}.$$

*If $\boldsymbol{x}_0^{\mathsf{T}}\boldsymbol{b} < 0$ or $\boldsymbol{x}_0^{\mathsf{T}}\boldsymbol{b} = 0$, then for $\boldsymbol{x}_1 = -\boldsymbol{x}_0$ or $\boldsymbol{x}_1 = \frac{\boldsymbol{b}^{\mathsf{T}}\boldsymbol{b}}{\boldsymbol{b}^{\mathsf{T}}\boldsymbol{A}\boldsymbol{b}}\boldsymbol{b}$ respectively, it holds that $\|\boldsymbol{x}_1 - \boldsymbol{x}_*\|_{\boldsymbol{A}}^2 < \|\boldsymbol{x}_0 - \boldsymbol{x}_*\|_{\boldsymbol{A}}^2$, i.e. $\boldsymbol{x}_1$ is a strictly better initialization than $\boldsymbol{x}_0$.*

*Proof.* Let $\boldsymbol{H}_0$ as above. Then $\boldsymbol{H}_0\boldsymbol{b} = \alpha\boldsymbol{b} + \boldsymbol{x}_0 - \alpha\boldsymbol{b} = \boldsymbol{x}_0$. The second term of the sum in the form of $\boldsymbol{H}_0$ is of rank 1. Its non-zero eigenvalue is given by

$$\lambda = \frac{1}{(\boldsymbol{x}_0 - \alpha\boldsymbol{b})^{\mathsf{T}}\boldsymbol{b}}(\boldsymbol{x}_0 - \alpha\boldsymbol{b})^{\mathsf{T}}(\boldsymbol{x}_0 - \alpha\boldsymbol{b}) = \frac{1}{\boldsymbol{x}_0^{\mathsf{T}}\boldsymbol{b} - \alpha\boldsymbol{b}^{\mathsf{T}}\boldsymbol{b}}\|\boldsymbol{x}_0 - \alpha\boldsymbol{b}\|_2^2 \geq 0$$

since by assumption $\boldsymbol{x}_0^{\mathsf{T}}\boldsymbol{b} > 0$ and $\alpha < \frac{\boldsymbol{b}^{\mathsf{T}}\boldsymbol{x}_0}{\boldsymbol{b}^{\mathsf{T}}\boldsymbol{b}}$. Now by Weyl's theorem it holds that $\lambda_{\min}(\boldsymbol{A}) + \lambda_{\min}(\boldsymbol{E}) \leq \lambda_{\min}(\boldsymbol{A} + \boldsymbol{E})$ and therefore $\boldsymbol{H}_0$ is positive definite. By the matrix inversion lemma we have for $\gamma = \frac{\alpha^{-1}}{(\boldsymbol{x}_0 - \alpha\boldsymbol{b})^{\mathsf{T}}\boldsymbol{b}}$ that

$$\boldsymbol{A}_0 = \boldsymbol{H}_0^{-1} = \alpha^{-1}(\boldsymbol{I} - \frac{\gamma}{1 + \gamma\|\boldsymbol{x}_0 - \alpha\boldsymbol{b}\|_2^2}(\boldsymbol{x}_0 - \alpha\boldsymbol{b})(\boldsymbol{x}_0 - \alpha\boldsymbol{b})^{\mathsf{T}})$$

$$= \alpha^{-1}\boldsymbol{I} - \frac{\alpha^{-2}}{(\boldsymbol{x}_0 - \alpha\boldsymbol{b})^{\mathsf{T}}\boldsymbol{b} + \alpha^{-1}\|\boldsymbol{x}_0 - \alpha\boldsymbol{b}\|_2^2}(\boldsymbol{x}_0 - \alpha\boldsymbol{b})(\boldsymbol{x}_0 - \alpha\boldsymbol{b})^{\mathsf{T}}$$

$$= \alpha^{-1}\boldsymbol{I} - \frac{\alpha^{-1}}{(\boldsymbol{x}_0 - \alpha\boldsymbol{b})^{\mathsf{T}}\boldsymbol{x}_0}(\boldsymbol{x}_0 - \alpha\boldsymbol{b})(\boldsymbol{x}_0 - \alpha\boldsymbol{b})^{\mathsf{T}}.$$

Finally, we obtain

$$\|\boldsymbol{x}_0 - \boldsymbol{x}_*\|_{\boldsymbol{A}}^2 = (\boldsymbol{x}_0 - \boldsymbol{A}^{-1}\boldsymbol{b})^{\mathsf{T}}\boldsymbol{A}(\boldsymbol{x}_0 - \boldsymbol{A}^{-1}\boldsymbol{b}) = \boldsymbol{x}_0^{\mathsf{T}}\boldsymbol{A}\boldsymbol{x}_0 + \boldsymbol{b}^{\mathsf{T}}\boldsymbol{A}^{-1}\boldsymbol{b} - 2\boldsymbol{b}^{\mathsf{T}}\boldsymbol{x}_0.$$

Therefore if either $\boldsymbol{x}_0^{\mathsf{T}}\boldsymbol{b} < 0$ or $\boldsymbol{x}_0^{\mathsf{T}}\boldsymbol{b} = 0$, then $\boldsymbol{x}_1 = -\boldsymbol{x}_0$ or $\boldsymbol{x}_1 = \frac{\boldsymbol{b}^{\mathsf{T}}\boldsymbol{b}}{\boldsymbol{b}^{\mathsf{T}}\boldsymbol{A}\boldsymbol{b}}\boldsymbol{b}$, respectively are closer to $\boldsymbol{x}_*$ in $\boldsymbol{A}$ norm by positive definiteness of $\boldsymbol{A}$. This concludes the proof. ☐

### S4.3 Stopping Criteria

In addition to the classic stopping criteria $\|\boldsymbol{A}\boldsymbol{x}_k - \boldsymbol{b}\|_2 \le \max(\delta_{\mathrm{rtol}}\|\boldsymbol{b}\|_2, \delta_{\mathrm{atol}})$ it is natural from a probabilistic viewpoint to use the induced posterior covariance of $\mathsf{x}$. Let $\boldsymbol{M} \in \mathbb{R}_{\mathrm{sym}}^{n \times n}$ be a positive-definite matrix, then by linearity and the cyclic property of the trace it holds that

$$
\begin{aligned}
\mathbb{E}_{\boldsymbol{x}_*}[\|\boldsymbol{x}_* - \mathbb{E}[\mathsf{x}]\|_{\boldsymbol{M}}^2] &= \mathbb{E}_{\boldsymbol{x}_*}[(\boldsymbol{x}_* - \mathbb{E}[\mathsf{x}])^{\mathsf{T}}\boldsymbol{M}(\boldsymbol{x}_* - \mathbb{E}[\mathsf{x}])] \\
&= \mathrm{tr}(\mathbb{E}_{\boldsymbol{x}_*}[(\boldsymbol{x}_* - \mathbb{E}[\mathsf{x}])^{\mathsf{T}}\boldsymbol{M}(\boldsymbol{x}_* - \mathbb{E}[\mathsf{x}])]) \\
&= \mathbb{E}_{\boldsymbol{x}_*}[\mathrm{tr}((\boldsymbol{x}_* - \mathbb{E}[\mathsf{x}])^{\mathsf{T}}\boldsymbol{M}(\boldsymbol{x}_* - \mathbb{E}[\mathsf{x}]))] \\
&= \mathbb{E}_{\boldsymbol{x}_*}[\boldsymbol{M}\,\mathrm{tr}((\boldsymbol{x}_* - \mathbb{E}[\mathsf{x}])(\boldsymbol{x}_* - \mathbb{E}[\mathsf{x}])^{\mathsf{T}})] \\
&= \mathrm{tr}(\boldsymbol{M}\mathbb{E}_{\boldsymbol{x}_*}[(\boldsymbol{x}_* - \mathbb{E}[\mathsf{x}])(\boldsymbol{x}_* - \mathbb{E}[\mathsf{x}])^{\mathsf{T}}]) \\
&= \mathrm{tr}(\boldsymbol{M}(\mathrm{Cov}[\boldsymbol{x}_* - \mathbb{E}[\mathsf{x}]] + (\mathbb{E}_{\boldsymbol{x}_*}[\boldsymbol{x}_*] - \mathbb{E}[\mathsf{x}])^{\mathsf{T}}(\mathbb{E}_{\boldsymbol{x}_*}[\boldsymbol{x}_*] - \mathbb{E}[\mathsf{x}]))) \\
&= \mathrm{tr}(\boldsymbol{M}\,\mathrm{Cov}[\boldsymbol{x}_*]) + \|\mathbb{E}_{\boldsymbol{x}_*}[\boldsymbol{x}_*] - \mathbb{E}[\mathsf{x}]\|_{\boldsymbol{M}}^2.
\end{aligned}
$$

Assuming calibration holds, i.e. $\boldsymbol{x}_* \sim \mathcal{N}(\mathbb{E}[\mathsf{x}], \mathrm{Cov}[\mathsf{x}])$, we can bound the (relative) error by terminating when $\mathrm{tr}(\boldsymbol{M}\,\mathrm{Cov}[\mathsf{x}]) \le \max(\delta_{\mathrm{rtol}}\|\boldsymbol{b}\|, \delta_{\mathrm{atol}})$ either in $l_2$-norm for $\boldsymbol{M} = \boldsymbol{I}$ or in $\boldsymbol{A}$-norm for $\boldsymbol{M} = \boldsymbol{A}$.

We can efficiently evaluate the required $\mathrm{tr}(\boldsymbol{M}\,\mathrm{Cov}[\mathsf{x}])$ without ever forming $\mathrm{Cov}[\mathsf{x}]$ in memory from already computed quantities. At iteration $k$ we have $\mathrm{Cov}[\mathsf{x}] = \mathrm{Cov}[\mathsf{H}\boldsymbol{b}] = \frac{1}{2}(\boldsymbol{W}_k^{\mathsf{H}}(\boldsymbol{b}^{\mathsf{T}}\boldsymbol{W}_k^{\mathsf{H}}\boldsymbol{b}) + (\boldsymbol{W}_k^{\mathsf{H}}\boldsymbol{b})(\boldsymbol{W}_k^{\mathsf{H}}\boldsymbol{b})^{\mathsf{T}})$ and therefore

$$\mathrm{tr}(\boldsymbol{M}\,\mathrm{Cov}[\mathsf{x}]) = \frac{1}{2}\big((\boldsymbol{b}^{\mathsf{T}}\boldsymbol{W}_k^{\mathsf{H}}\boldsymbol{b})\,\mathrm{tr}(\boldsymbol{M}\boldsymbol{W}_k^{\mathsf{H}}) + (\boldsymbol{W}_k^{\mathsf{H}}\boldsymbol{b})^{\mathsf{T}}\boldsymbol{M}(\boldsymbol{W}_k^{\mathsf{H}}\boldsymbol{b})\big).$$

Given the update for the covariance of the inverse view, we obtain the following recursion for its trace

$$\mathrm{tr}(\boldsymbol{M}\boldsymbol{W}_k^{\mathsf{H}}) = \mathrm{tr}(\boldsymbol{M}\boldsymbol{W}_{k-1}^{\mathsf{H}}) - \frac{1}{\boldsymbol{y}_k^{\mathsf{T}}\boldsymbol{W}_{k-1}^{\mathsf{H}}\boldsymbol{y}_k}\,\mathrm{tr}\big((\boldsymbol{W}_{k-1}^{\mathsf{H}}\boldsymbol{y}_k)^{\mathsf{T}}\boldsymbol{M}(\boldsymbol{W}_{k-1}^{\mathsf{H}}\boldsymbol{y}_k)\big).$$

Computing the trace in this iterative fashion adds at most three matrix-vector products and three inner products for arbitrary $\boldsymbol{M}$ all other quantities are computed for the covariance update anyhow.

For our proposed covariance class (3) we obtain for $\boldsymbol{M} = \boldsymbol{I}$ and $\boldsymbol{\Psi} = \psi\boldsymbol{I}$ that

$$
\begin{aligned}
\mathrm{tr}(\boldsymbol{W}_0^{\mathsf{H}}) &= \mathrm{tr}\big(\boldsymbol{A}_0^{-1}\boldsymbol{Y}(\boldsymbol{Y}^{\mathsf{T}}\boldsymbol{A}_0^{-1}\boldsymbol{Y})^{-1}\boldsymbol{Y}^{\mathsf{T}}\boldsymbol{A}_0^{-1} + (\boldsymbol{I} - \boldsymbol{Y}(\boldsymbol{Y}^{\mathsf{T}}\boldsymbol{Y})^{-1}\boldsymbol{Y}^{\mathsf{T}})\boldsymbol{\Psi}(\boldsymbol{I} - \boldsymbol{Y}(\boldsymbol{Y}^{\mathsf{T}}\boldsymbol{Y})^{-1}\boldsymbol{Y}^{\mathsf{T}})\big) \\
&= \mathrm{tr}\big((\boldsymbol{Y}^{\mathsf{T}}\boldsymbol{A}_0^{-1}\boldsymbol{Y})^{-1}\boldsymbol{Y}^{\mathsf{T}}\boldsymbol{A}_0^{-1}\boldsymbol{A}_0^{-1}\boldsymbol{Y}\big) + \psi\,\mathrm{tr}\big((\boldsymbol{I} - \boldsymbol{Y}(\boldsymbol{Y}^{\mathsf{T}}\boldsymbol{Y})^{-1}\boldsymbol{Y}^{\mathsf{T}})(\boldsymbol{I} - \boldsymbol{Y}(\boldsymbol{Y}^{\mathsf{T}}\boldsymbol{Y})^{-1}\boldsymbol{Y}^{\mathsf{T}})\big) \\
&= \mathrm{tr}\big((\boldsymbol{Y}^{\mathsf{T}}\boldsymbol{A}_0^{-1}\boldsymbol{Y})^{-1}\boldsymbol{Y}^{\mathsf{T}}\boldsymbol{A}_0^{-1}\boldsymbol{A}_0^{-1}\boldsymbol{Y}\big) + \psi\,\mathrm{tr}\big(\boldsymbol{I} - \boldsymbol{Y}(\boldsymbol{Y}^{\mathsf{T}}\boldsymbol{Y})^{-1}\boldsymbol{Y}^{\mathsf{T}}\big) \\
&= \mathrm{tr}\big((\boldsymbol{Y}^{\mathsf{T}}\boldsymbol{A}_0^{-1}\boldsymbol{Y})^{-1}\boldsymbol{Y}^{\mathsf{T}}\boldsymbol{A}_0^{-1}\boldsymbol{A}_0^{-1}\boldsymbol{Y}\big) + \psi(n - k),
\end{aligned}
$$

which for a scalar prior mean $\boldsymbol{A}_0 = \alpha\boldsymbol{I}$ reduces to $\mathrm{tr}(\boldsymbol{W}_0^{\mathsf{H}}) = \alpha^{-1}k + \psi(n - k)$.

### S4.4 Implementation

In order to maintain numerical stability when performing low rank updates to symmetric positive definite matrices, as is the case in Algorithm 1 for the mean and covariance estimates, it is advantageous use a representation based on the Cholesky decomposition. One can perform the rank-2 update for the mean estimate and the rank-1 downdate for the covariance in Corollary S3 in each iteration of the algorithm for their respective Cholesky factors instead (see also Seeger [17]). The rank-2 update can be seen as a combination of a rank-1 up- and downdate by recognizing that

$$\boldsymbol{u}\boldsymbol{v}^{\mathsf{T}} + \boldsymbol{v}\boldsymbol{u}^{\mathsf{T}} = \frac{1}{2}((\boldsymbol{u} + \boldsymbol{v})(\boldsymbol{u} + \boldsymbol{v})^{\mathsf{T}} - (\boldsymbol{u} - \boldsymbol{v})(\boldsymbol{u} - \boldsymbol{v})^{\mathsf{T}}).$$

Similar updates arise in Quasi-Newton methods for the approximate (inverse) Hessian [18]. Having Cholesky factors of the mean and covariance available has the additional advantage that downstream sampling or the evaluation of the probability density function is computationally cheap.

## S5    Theoretical Properties: Proofs for Section 2.3

In this section we provide detailed proofs for the theoretical results on convergence and the connection of Algorithm 1 to the method of conjugate gradients. We restate each theorem here as a reference to the reader. We begin by proving an intermediate result giving an interpretation to the posterior mean of $\mathbf{A}$ and $\mathbf{H}$ at each step of the method.

**Proposition S5** (Subspace Equivalency)
*Let $\boldsymbol{A}_k$ and $\boldsymbol{H}_k$ be the posterior means defined as in Section 2.1 and assume $\boldsymbol{A}_0$ and $\boldsymbol{H}_0$ are symmetric. Then for $1 \leq k \leq n$ it holds that*

$$\boldsymbol{A}_k \boldsymbol{S} = \boldsymbol{Y} \quad \text{and} \quad \boldsymbol{H}_k \boldsymbol{Y} = \boldsymbol{S}, \tag{S33}$$

*i.e. $\boldsymbol{A}_k$ and $\boldsymbol{H}_k$ act like $\boldsymbol{A}$ and $\boldsymbol{A}^{-1}$ on the spaces spanned by the actions $\boldsymbol{S}$, respectively the observations $\boldsymbol{Y}$.*

*Proof.* Since $\boldsymbol{A}_0$ and $\boldsymbol{H}_0$ are symmetric so are the expressions $\boldsymbol{\Delta_A} \boldsymbol{S}$ and $\boldsymbol{\Delta_H^\mathsf{T}} \boldsymbol{Y}$. We have that

$$\begin{aligned}
\boldsymbol{A}_k \boldsymbol{S} &= (\boldsymbol{A}_0 + \boldsymbol{\Delta_A} \boldsymbol{U_A^\mathsf{T}} + \boldsymbol{U_A} \boldsymbol{\Delta_A^\mathsf{T}} - \boldsymbol{U_A} \boldsymbol{S}^\mathsf{T} \boldsymbol{\Delta_A} \boldsymbol{U_A^\mathsf{T}}) \boldsymbol{S} \\
&= \boldsymbol{A}_0 \boldsymbol{S} + \boldsymbol{\Delta_A} \boldsymbol{I} + \boldsymbol{U_A} \boldsymbol{\Delta_A^\mathsf{T}} \boldsymbol{S} - \boldsymbol{U_A} \boldsymbol{S}^\mathsf{T} \boldsymbol{\Delta_A} \boldsymbol{I} \\
&= \boldsymbol{A}_0 \boldsymbol{S} + \boldsymbol{Y} - \boldsymbol{A}_0 \boldsymbol{S} \\
&= \boldsymbol{Y}.
\end{aligned}$$

In the case of the inverse model we obtain

$$\begin{aligned}
\boldsymbol{H}_k \boldsymbol{Y} &= (\boldsymbol{H}_0 + \boldsymbol{\Delta_H} \boldsymbol{U_H^\mathsf{T}} + \boldsymbol{U_H} \boldsymbol{\Delta_H^\mathsf{T}} - \boldsymbol{U_H} \boldsymbol{Y}^\mathsf{T} \boldsymbol{\Delta_H} \boldsymbol{U_H^\mathsf{T}}) \boldsymbol{Y} \\
&= \boldsymbol{H}_0 \boldsymbol{Y} + \boldsymbol{\Delta_H} \boldsymbol{I} + \boldsymbol{U_H} \boldsymbol{\Delta_H^\mathsf{T}} \boldsymbol{Y} - \boldsymbol{U_H} \boldsymbol{Y}^\mathsf{T} \boldsymbol{\Delta_H} \boldsymbol{I} \\
&= \boldsymbol{H}_0 \boldsymbol{Y} + \boldsymbol{S} - \boldsymbol{H}_0 \boldsymbol{Y} \\
&= \boldsymbol{S}
\end{aligned}$$

$\square$

### S5.1    Conjugate Directions Method

**Theorem 1** (Conjugate Directions Method)
*Given a prior $p(\mathbf{H}) = \mathcal{N}(\mathbf{H}; \boldsymbol{H}_0, \boldsymbol{W}_0^{\mathbf{H}} \otimes \boldsymbol{W}_0^{\mathbf{H}})$ such that $\boldsymbol{H}_0, \boldsymbol{W}_0^{\mathbf{H}} \in \mathbb{R}_{\text{sym}}^{n \times n}$ positive definite, then actions $\boldsymbol{s}_i$ of Algorithm 1 are $\boldsymbol{A}$-conjugate, i.e. for $0 \leq i, j \leq k$ with $i \neq j$ it holds that $\boldsymbol{s}_i^\mathsf{T} \boldsymbol{A} \boldsymbol{s}_j = 0$.*

*Proof.* Since $\boldsymbol{H}_0$ is assumed to be symmetric, the form of the posterior mean in Section 2.1 implies that $\boldsymbol{H}_k$ is symmetric for all $1 \leq k \leq n$. Now conjugacy is shown by induction. To that end, first consider the base case $k = 2$. We have

$$\begin{aligned}
\boldsymbol{s}_2^\mathsf{T} \boldsymbol{A} \boldsymbol{s}_1 &= -\boldsymbol{r}_1^\mathsf{T} \boldsymbol{H}_1 \boldsymbol{A} \boldsymbol{s}_1 = -(\boldsymbol{r}_0^\mathsf{T} + \alpha_1 \boldsymbol{y}_1^\mathsf{T}) \boldsymbol{H}_1 \boldsymbol{A} \boldsymbol{s}_1 = -\left( \boldsymbol{r}_0^\mathsf{T} \boldsymbol{H}_1 - \frac{\boldsymbol{s}_1^\mathsf{T} \boldsymbol{r}_0}{\boldsymbol{s}_1^\mathsf{T} \boldsymbol{y}_1} \boldsymbol{y}_1^\mathsf{T} \boldsymbol{H}_1 \right) \boldsymbol{y}_1 \\
&= -\boldsymbol{r}_0^\mathsf{T} \boldsymbol{s}_1 + \boldsymbol{s}_1^\mathsf{T} \boldsymbol{r}_0 = 0
\end{aligned}$$

where we used (S33) and the definition of $\alpha_i$ in Algorithm 1. Now for the induction step, assume that $\boldsymbol{s}_i^\mathsf{T} \boldsymbol{A} \boldsymbol{s}_j = 0$ for all $i \neq j$ such that $1 \leq i, j \leq k$. We obtain for $1 \leq j \leq k$ that

$$\begin{aligned}
\boldsymbol{s}_{k+1}^\mathsf{T} \boldsymbol{A} \boldsymbol{s}_j &= -\boldsymbol{r}_k^\mathsf{T} \boldsymbol{H}_k \boldsymbol{A} \boldsymbol{s}_j = -\left( \sum_{1 \leq l \leq k} \alpha_l \boldsymbol{y}_l + \boldsymbol{r}_0 \right)^\mathsf{T} \boldsymbol{H}_k \boldsymbol{y}_j = -\sum_{1 \leq l \leq k} \alpha_l \boldsymbol{y}_l^\mathsf{T} \boldsymbol{s}_j - \boldsymbol{r}_0^\mathsf{T} \boldsymbol{s}_j \\
&= -\alpha_j \boldsymbol{y}_j^\mathsf{T} \boldsymbol{s}_j - \boldsymbol{r}_0^\mathsf{T} \boldsymbol{s}_j = \boldsymbol{s}_j^\mathsf{T} \boldsymbol{r}_{j-1} - \boldsymbol{r}_0^\mathsf{T} \boldsymbol{s}_j = \boldsymbol{s}_j^\mathsf{T} \left( \sum_{1 \leq l < j} \alpha_l \boldsymbol{y}_l + \boldsymbol{r}_0 \right) - \boldsymbol{r}_0^\mathsf{T} \boldsymbol{s}_j \\
&= \boldsymbol{s}_j^\mathsf{T} \boldsymbol{r}_0 - \boldsymbol{r}_0^\mathsf{T} \boldsymbol{s}_j = 0
\end{aligned}$$

where we used the update equation of the residual $\boldsymbol{r}_i$ in Algorithm 1, the definition of $\alpha_i$, the induction hypothesis and (S33). This proves the statement. $\square$

## S5.2 Relationship to the Conjugate Gradient Method

**Theorem 2** (Connection to the Conjugate Gradient Method)
*Given a scalar prior mean $\boldsymbol{A}_0 = \boldsymbol{H}_0^{-1} = \alpha\boldsymbol{I}$ with $\alpha > 0$, assume (1) and (2) hold, then the iterates $\boldsymbol{x}_i$ of Algorithm 1 are identical to the ones produced by the conjugate gradient method.*

*Proof.* The proof outlined here is closely related to the proofs connecting Quasi-Newton methods to the conjugate gradient method [19, 6], but makes different assumptions on the prior distribution.

We begin by recognizing that the choice of step length $\alpha_i$ in Algorithm 1 is identical to the one in the conjugate gradient method [18]. Hence, it suffices to show that $\boldsymbol{s}_i \propto \boldsymbol{s}_i^{\text{CG}}$. Theorem 1 established that Algorithm 1 is a conjugate directions method. Now by assumption $\boldsymbol{A}_0 = \alpha\boldsymbol{I}$ and $\boldsymbol{H}_0 = \boldsymbol{A}_0^{-1}$, therefore $\boldsymbol{s}_1 = -\alpha\boldsymbol{I}\boldsymbol{r}_0 \propto -\boldsymbol{r}_0 = \boldsymbol{s}_1^{\text{CG}}$. It suffices show that $\boldsymbol{s}_i$ lies in the Krylov space $\mathcal{K}_i(\boldsymbol{A}, \boldsymbol{r}_0) = \{\boldsymbol{r}_0, \boldsymbol{A}\boldsymbol{r}_0, \ldots, \boldsymbol{A}^{i-1}\boldsymbol{r}_0\}$ for all $0 < i \le n$. This completes the argument, since $\mathcal{K}_i(\boldsymbol{A}, \boldsymbol{r}_0)$ is an $i$-dimensional subspace of $\mathbb{R}^n$ and thus $\boldsymbol{A}$-conjugacy uniquely determines the search directions up to scaling, as $\boldsymbol{A}$ is positive definite.

To complete the proof we proceed as follows. The posterior mean of the inverse model $\boldsymbol{H}_{i-1}$ at step $i-1$ maps an arbitrary vector $\boldsymbol{v} \in \mathbb{R}^n$ to $\operatorname{span}(\boldsymbol{H}_0\boldsymbol{v}, \boldsymbol{H}_0\boldsymbol{Y}_{1:i-1}, \boldsymbol{S}_{1:i-1}, \boldsymbol{W}_0^{\mathsf{H}}\boldsymbol{Y}_{1:i-1})$. This follows directly from its form in given in Section 2.1. By assumption $\boldsymbol{H}_0 = \boldsymbol{A}_0^{-1} = \alpha^{-1}\boldsymbol{I}$, therefore using (1) and (2) we have $\operatorname{span}(\boldsymbol{W}_0^{\mathsf{H}}\boldsymbol{Y}_{1:i-1}) = \operatorname{span}(\boldsymbol{Y}_{1:i-1})$. This implies $\boldsymbol{H}_{i-1}$ maps to $\operatorname{span}(\boldsymbol{v}, \boldsymbol{S}_{1:i-1}, \boldsymbol{Y}_{1:i-1})$ and thus $\boldsymbol{s}_i \in \operatorname{span}(\boldsymbol{r}_{i-1}, \boldsymbol{S}_{1:i-1}, \boldsymbol{Y}_{1:i-1})$. We will now show that $\operatorname{span}(\boldsymbol{r}_{i-1}, \boldsymbol{S}_{1:i-1}, \boldsymbol{Y}_{1:i-1}) \subset \mathcal{K}_i(\boldsymbol{A}, \boldsymbol{r}_0)$ by induction, completing the argument.

We begin with the base case. Since $\boldsymbol{H}_0$ is assumed to be scalar, we have $\boldsymbol{s}_1 \propto \boldsymbol{r}_0 \in \mathcal{K}_0(\boldsymbol{A}, \boldsymbol{r}_0)$ and therefore $\boldsymbol{y}_1 = \boldsymbol{A}\boldsymbol{s}_1$ and $\boldsymbol{r}_1 = \boldsymbol{r}_0 + \alpha_1\boldsymbol{y}_1$ are in $\mathcal{K}_1(\boldsymbol{A}, \boldsymbol{r}_0)$. For the induction step assume $\operatorname{span}(\boldsymbol{r}_{i-1}, \boldsymbol{S}_{1:i-1}, \boldsymbol{Y}_{1:i-1}) \subset \mathcal{K}_i(\boldsymbol{A}, \boldsymbol{r}_0)$. The definition of the policy of Algorithm 1 gives

$$\boldsymbol{s}_i = -\mathbb{E}[\mathsf{H}]\boldsymbol{r}_{i-1} \propto \boldsymbol{H}_{i-1}\boldsymbol{r}_{i-1} \in \operatorname{span}(\boldsymbol{r}_{i-1}, \boldsymbol{S}_{1:i-1}, \boldsymbol{Y}_{1:i-1}) \subset \mathcal{K}_i(\boldsymbol{A}, \boldsymbol{r}_0),$$

where we used the induction hypothesis. This implies that $\boldsymbol{y}_i = \boldsymbol{A}\boldsymbol{s}_i \in \mathcal{K}_{i+1}(\boldsymbol{A}, \boldsymbol{r}_0)$ and $\boldsymbol{r}_i = \boldsymbol{r}_{i-1} + \alpha_i\boldsymbol{y}_i \in \mathcal{K}_{i+1}(\boldsymbol{A}, \boldsymbol{r}_0)$ by the definition of the Krylov space. Therefore, $\operatorname{span}(\boldsymbol{r}_i, \boldsymbol{S}_{1:i}, \boldsymbol{Y}_{1:i}) \subset \mathcal{K}_{i+1}(\boldsymbol{A}, \boldsymbol{r}_0)$. This completes the proof. $\square$

# S6 Prior Covariance Class: Proofs for Section 3

## S6.1 Hereditary Positive-Definiteness

**Proposition 1** (Hereditary Positive Definiteness [20, 16])
*Let $\boldsymbol{A}_0 \in \mathbb{R}^{n\times n}_{\text{sym}}$ be positive definite. Assume the actions $\boldsymbol{S}$ are $\boldsymbol{A}$-conjugate and $\boldsymbol{W}_0^{\mathsf{A}}\boldsymbol{S} = \boldsymbol{Y}$, then for $i \in \{0, \ldots, k-1\}$ it holds that $\boldsymbol{A}_{i+1}$ is symmetric positive definite.*

*Proof.* This is shown in Hennig and Kiefel [16]. We give an identical proof in our notation as a reference to the reader. By Theorem 7.5 in Dennis and Moré [20] it holds that if $\boldsymbol{A}_i$ is positive definite and $\boldsymbol{s}_{i+1}^{\mathsf{T}}\boldsymbol{W}_i^{\mathsf{A}}\boldsymbol{s}_{i+1} \neq 0$, then $\boldsymbol{A}_{i+1}$ is positive definite if and only if $\det(\boldsymbol{A}_{i+1}) > 0$. By the matrix determinant lemma and the recursive formulation of the posterior we have

$$\det(\boldsymbol{A}_{i+1}) = \det(\boldsymbol{A}_i)\bigg(\frac{1}{(\boldsymbol{s}_{i+1}^{\mathsf{T}}\boldsymbol{W}_i^{\mathsf{A}}\boldsymbol{s}_{i+1})^2}\big((\boldsymbol{y}_{i+1}^{\mathsf{T}}\boldsymbol{A}_i^{-1}\boldsymbol{W}_i^{\mathsf{A}}\boldsymbol{s}_{i+1})^2$$

$$- (\boldsymbol{y}_{i+1}^{\mathsf{T}}\boldsymbol{A}_i^{-1}\boldsymbol{y}_{i+1})(\boldsymbol{s}_{i+1}^{\mathsf{T}}\boldsymbol{W}_i^{\mathsf{A}}\boldsymbol{A}_i^{-1}\boldsymbol{W}_i^{\mathsf{A}}\boldsymbol{s}_{i+1}) + (\boldsymbol{s}_{i+1}^{\mathsf{T}}\boldsymbol{W}_i^{\mathsf{A}}\boldsymbol{A}_i^{-1}\boldsymbol{W}_i^{\mathsf{A}}\boldsymbol{s}_{i+1})(\boldsymbol{y}_{i+1}^{\mathsf{T}}\boldsymbol{s}_{i+1})\big)\bigg)$$

Hence it suffices to show that

$$0 < (\boldsymbol{y}_{i+1}^{\mathsf{T}}\boldsymbol{A}_i^{-1}\boldsymbol{W}_i^{\mathsf{A}}\boldsymbol{s}_{i+1})^2 - (\boldsymbol{y}_{i+1}^{\mathsf{T}}\boldsymbol{A}_i^{-1}\boldsymbol{y}_{i+1})(\boldsymbol{s}_{i+1}^{\mathsf{T}}\boldsymbol{W}_i^{\mathsf{A}}\boldsymbol{A}_i^{-1}\boldsymbol{W}_i^{\mathsf{A}}\boldsymbol{s}_{i+1})$$
$$+ (\boldsymbol{s}_{i+1}^{\mathsf{T}}\boldsymbol{W}_i^{\mathsf{A}}\boldsymbol{A}_i^{-1}\boldsymbol{W}_i^{\mathsf{A}}\boldsymbol{s}_{i+1})(\boldsymbol{y}_{i+1}^{\mathsf{T}}\boldsymbol{s}_{i+1}),$$

which simplifies to

$$\boldsymbol{y}_{i+1}^{\mathsf{T}}\boldsymbol{A}_i^{-1}\boldsymbol{y}_{i+1} - \frac{(\boldsymbol{y}_{i+1}^{\mathsf{T}}\boldsymbol{A}_i^{-1}\boldsymbol{W}_i^{\mathsf{A}}\boldsymbol{s}_{i+1})^2}{\boldsymbol{s}_{i+1}^{\mathsf{T}}\boldsymbol{W}_i^{\mathsf{A}}\boldsymbol{A}_i^{-1}\boldsymbol{W}_i^{\mathsf{A}}\boldsymbol{s}_{i+1}} < \boldsymbol{y}_{i+1}^{\mathsf{T}}\boldsymbol{s}_{i+1}$$

Now by $W_0^{\mathsf{A}} S = Y$, we have $W_i^{\mathsf{A}} s_{i+1} = W_0^{\mathsf{A}} s_{i+1} = y_{i+1}$ and the above reduces to

$$0 < s_{i+1}^{\mathsf{T}} A s_{i+1},$$

which is fulfilled by the assumption that $A$ is positive definite. Thus $A_{i+1}$ is positive definite. Symmetry follows immediately from the form of the posterior mean. $\square$

## S6.2 Posterior Correspondence

**Definition 1**
Let $A_i$ and $H_i$ be the means of $\mathsf{A}$ and $\mathsf{H}$ at step $i$. We say a prior induces *posterior correspondence* if

$$A_i^{-1} = H_i \tag{S34}$$

for all steps $0 \leq i \leq k$ of the solver. If only

$$A_i^{-1} Y = H_i Y, \tag{S35}$$

we say that *weak posterior correspondence* holds.

### S6.2.1 Matrix-variate Normal Prior

We begin by establishing posterior correspondence in the case of general matrix-variate normal priors, i.e. the inference setting detailed in Corollary S2. We begin by proving a general non-constructive condition and close with a sufficient condition for correspondence with limits the possible choices of covariance factors to a specific class.

**Lemma S1** (General Correspondence)
*Let $1 \leq k \leq n$, $W_0^{\mathsf{A}}, W_0^{\mathsf{H}}$ symmetric positive-definite and assume $A_0^{-1} = H_0$, then* (S34) *holds if and only if*

$$0 = (AS - A_0 S)\left[ (S^{\mathsf{T}} W_0^{\mathsf{A}} A_0^{-1} AS)^{-1} S^{\mathsf{T}} W_0^{\mathsf{A}} A_0^{-1} - (S^{\mathsf{T}} A^{\mathsf{T}} W_0^{\mathsf{H}} AS)^{-1} S^{\mathsf{T}} A^{\mathsf{T}} W_0^{\mathsf{H}} \right]. \tag{S36}$$

*Proof.* By the matrix inversion lemma we have

$$
\begin{aligned}
0 &= A_k^{-1} - H_k \\
&= \left( A_0 + (Y - A_0 S)(S^{\mathsf{T}} W_0^{\mathsf{A}} S)^{-1} S^{\mathsf{T}} W_0^{\mathsf{A}} \right)^{-1} - H_0 - (S - H_0 Y)(Y^{\mathsf{T}} W_0^{\mathsf{H}} Y)^{-1} Y^{\mathsf{T}} W_0^{\mathsf{H}} \\
&= A_0^{-1} - A_0^{-1}(Y - A_0 S)(S^{\mathsf{T}} W_0^{\mathsf{A}} S + S^{\mathsf{T}} W_0^{\mathsf{A}} A_0^{-1}(Y - A_0 S))^{-1} S^{\mathsf{T}} W_0^{\mathsf{A}} A_0^{-1} \\
&\quad - A_0^{-1} - A_0^{-1}(A_0 S - Y)(Y^{\mathsf{T}} W_0^{\mathsf{H}} Y)^{-1} Y^{\mathsf{T}} W_0^{\mathsf{H}} \\
&= -A_0^{-1}(Y - A_0 S)\left[ (S^{\mathsf{T}} W_0^{\mathsf{A}} A_0^{-1} Y)^{-1} S^{\mathsf{T}} W_0^{\mathsf{A}} A_0^{-1} - (Y^{\mathsf{T}} W_0^{\mathsf{H}} Y)^{-1} Y^{\mathsf{T}} W_0^{\mathsf{H}} \right],
\end{aligned}
$$

where we used the assumption $H_0 = A_0^{-1}$. Left-multiplying with $-A_0$ and using $Y = AS$ completes the proof. $\square$

**Corollary S4** (Correspondence at Convergence)
*Let $k = n$, $H_0 = A_0^{-1}$ and assume $S$ has full rank, i.e. the linear solver has performed $n$ linearly independent actions, then* (S34) *holds for any symmetric positive-definite choice of $W_0^{\mathsf{A}}$ and $W_0^{\mathsf{H}}$.*

*Proof.* By assumption, $S^{\mathsf{T}} W_0^{\mathsf{A}} A_0^{-1}$ and $S^{\mathsf{T}} A^{\mathsf{T}} W_0^{\mathsf{H}}$ are invertible. Then by Lemma S1 the correspondence condition (S34) holds. $\square$

**Theorem S2** (Sufficient Condition for Correspondence)
*Let $1 \leq k \leq n$ arbitrary and assume $H_0 = A_0^{-1}$. Assume $W_0^A, A_0, W_0^H$ satisfy*

$$0 = S^{\mathsf{T}}(W_0^{\mathsf{A}} A_0^{-1} - A^{\mathsf{T}} W_0^{\mathsf{H}}) \tag{S37}$$

*or equivalently let $B_{\langle S \rangle^{\perp}} \in \mathbb{R}^{n \times k}$ be a basis of the orthogonal space $\langle S \rangle^{\perp}$ spanned by the actions. For $\Phi \in \mathbb{R}^{(n-k) \times n}$ arbitrary, if*

$$W_0^{\mathsf{H}} = A^{-\mathsf{T}}(W_0^{\mathsf{A}} A_0^{-1} - B_{\langle S \rangle^{\perp}} \Phi) \tag{S38}$$

*and the commutation relations*

$$[\boldsymbol{A}_0, \boldsymbol{A}] = \boldsymbol{0} \tag{S39}$$

$$[\boldsymbol{W}_0^{\mathsf{A}}, \boldsymbol{A}] = \boldsymbol{0} \tag{S40}$$

$$[\boldsymbol{B}_{\langle \boldsymbol{S} \rangle^\perp} \boldsymbol{\Phi}, \boldsymbol{A}] = \boldsymbol{0} \tag{S41}$$

*are fulfilled, then* $\boldsymbol{W}_0^{\mathsf{H}}$ *is symmetric and* (S34) *holds.*

*Proof.* By assumption $\boldsymbol{W}_0^{\mathsf{A}}$ is symmetric positive-definite and (S37) is equivalent to $\boldsymbol{S}^{\mathsf{T}} \boldsymbol{W}_0^{\mathsf{A}} \boldsymbol{A}_0^{-1} = \boldsymbol{S}^{\mathsf{T}} \boldsymbol{A}^{\mathsf{T}} \boldsymbol{W}_0^{\mathsf{H}}$, which implies (S36). Now, assumption (S37) is equivalent to columns of the difference $\boldsymbol{W}_0^{\mathsf{A}} \boldsymbol{A}_0^{-1} - \boldsymbol{A}^{\mathsf{T}} \boldsymbol{W}_0^{\mathsf{H}}$ lying in $L$, i.e. we can choose a basis $\boldsymbol{B}_{\langle \boldsymbol{S} \rangle^\perp}$ and coefficient matrix $\boldsymbol{\Phi}$ such that

$$\boldsymbol{W}_0^{\mathsf{A}} \boldsymbol{A}_0^{-1} - \boldsymbol{A}^{\mathsf{T}} \boldsymbol{W}_0^{\mathsf{H}} = \boldsymbol{B}_{\langle \boldsymbol{S} \rangle^\perp} \boldsymbol{\Phi}.$$

Rearranging the above gives (S38). With the commutation relations and

$$[\boldsymbol{A}, \boldsymbol{B}] = \boldsymbol{0} \iff [\boldsymbol{A}^{-1}, \boldsymbol{B}] = \boldsymbol{0} \iff [\boldsymbol{A}, \boldsymbol{B}^{-1}] = \boldsymbol{0} \iff [\boldsymbol{A}^{-1}, \boldsymbol{B}^{-1}] = \boldsymbol{0}$$

it holds that

$$(\boldsymbol{W}_0^{\mathsf{H}})^{\mathsf{T}} = \boldsymbol{W}_0^{\mathsf{A}} \boldsymbol{A}_0^{-1} \boldsymbol{A}^{-1} - \boldsymbol{B}_{\langle \boldsymbol{S} \rangle^\perp} \boldsymbol{\Phi} \boldsymbol{A}^{-1} = \boldsymbol{A}^{-\mathsf{T}} \boldsymbol{W}_0^{\mathsf{A}} \boldsymbol{A}_0^{-1} - \boldsymbol{A}^{-\mathsf{T}} \boldsymbol{B}_{\langle \boldsymbol{S} \rangle^\perp} \boldsymbol{\Phi} = \boldsymbol{W}_0^{\mathsf{H}}$$

hence $\boldsymbol{W}_0^{\mathsf{H}}$ is symmetric. Finally, by Lemma S1 posterior mean correspondence (S34) holds. $\square$

If we want to ensure correspondence for all iterations, (S41) is trivially satisfied. The question now becomes what form can $\boldsymbol{A}_0$ and $\boldsymbol{W}_0^{\mathsf{A}}$ take in order to ensure symmetric $\boldsymbol{W}_0^{\mathsf{H}}$. This comes down to finding matrices which commute with $\boldsymbol{A}$.

**Lemma S2** (Commuting Matrices of a Symmetric Matrix)
*Let* $r \in \mathbb{N}$, $\boldsymbol{M} \in \mathbb{R}^{n \times n}$ *and* $\boldsymbol{A} \in \mathbb{R}^{n \times n}$ *symmetric. Assume* $\boldsymbol{M}$ *has the form*

$$\boldsymbol{M} = \mathfrak{p}_r(\boldsymbol{A}) = \sum_{i=0}^{r} c_i \boldsymbol{A}^i$$

*for a set of coefficients* $c_i \in \mathbb{R}$, *then* $\boldsymbol{M}$ *and* $\boldsymbol{A}$ *commute. If* $\boldsymbol{A}$ *has* $n$ *distinct eigenvalues,* $\boldsymbol{M}$ *is diagonalizable and* $[\boldsymbol{M}, \boldsymbol{A}] = \boldsymbol{0}$, *then*

$$\boldsymbol{M} = \mathfrak{p}_{n-1}(\boldsymbol{A}),$$

*i.e.* $\boldsymbol{M}$ *is a polynomial in* $\boldsymbol{A}$ *of degree at most* $n - 1$.

*Proof.* The first result follows immediately since

$$\boldsymbol{W}_0^{\mathsf{A}} \boldsymbol{A} = \mathfrak{p}_r(\boldsymbol{A}) \boldsymbol{A} = \sum_{i=0}^{r} c_i \boldsymbol{A}^{i+1} = \boldsymbol{A} \mathfrak{p}_r(\boldsymbol{A}) = \boldsymbol{A} \boldsymbol{W}_0^{\mathsf{A}}.$$

Assume now that $\boldsymbol{A}$ has $n$ distinct eigenvalues $\lambda_0, \ldots, \lambda_{n-1}$, $\boldsymbol{M}$ is diagonalizable and $\boldsymbol{M}$ and $\boldsymbol{A}$ commute. Now, if and only if $[\boldsymbol{A}, \boldsymbol{M}] = \boldsymbol{0}$, then $\boldsymbol{A}$ and $\boldsymbol{M}$ are simultaneously diagonalizable by Theorem 5.2 in Conrad [21], i.e. we can find a common basis in which both $\boldsymbol{A}$ and $\boldsymbol{M}$ are represented by diagonal matrices. Hence, the set of matrices commuting with $\boldsymbol{A}$ forms an $n$-dimensional subspace $\mathcal{U}_n \subset \mathbb{R}^{n \times n}$. Now, by the first part of this proof $\{\boldsymbol{I}, \boldsymbol{A}, \ldots, \boldsymbol{A}^{n-1}\} \subset \mathcal{U}_n$. It remains to be shown, that this set forms a basis of $\mathcal{U}_n$. By isomorphism of finite dimensional vector spaces this is equivalent to proving that

$$\{\boldsymbol{b}_0, \boldsymbol{b}_1, \ldots, \boldsymbol{b}_{n-1}\} := \left\{ \begin{pmatrix} 1 \\ \vdots \\ 1 \end{pmatrix}, \begin{pmatrix} \lambda_0 \\ \vdots \\ \lambda_{n-1} \end{pmatrix}, \ldots, \begin{pmatrix} \lambda_0^{n-1} \\ \vdots \\ \lambda_{n-1}^{n-1} \end{pmatrix} \right\}$$

forms a basis of $\mathbb{R}^n$. It suffices to show that all $\boldsymbol{b}_i$ are independent. Assume the contrary, then $\sum_{i=0}^{n-1} \alpha_i \boldsymbol{b}_i = \boldsymbol{0}$ for some $\alpha_0, \ldots, \alpha_{n-1} \in \mathbb{R}$, such that not all $\alpha_i = 0$. This implies that the polynomial $\sum_{i=0}^{n-1} \alpha_i x^i$ has $n$ zeros $\lambda_0, \ldots, \lambda_{n-1}$. This contradicts the fundamental theorem of algebra, concluding the proof. $\square$

The above suggests that tractable choices of $A_0$ and $W_0^{\mathsf{A}}$ for the non-symmetric matrix-variate prior, which imply symmetric $W_0^{\mathsf{H}}$, are of polynomial form in $A$.

**Example S1** (Posterior Correspondence Covariance Class)
Tractable choices of the prior parameters in the $\mathbf{A}$ view, which satisfy posterior correspondence and the commutation relations are for example

$$A_0 = c_0 I \qquad \text{and} \qquad W_0^{\mathsf{A}} = \sum_{i=1}^{n-1} c_i A^i,$$

where $H_0 = A_0^{-1}$ with $c_i \in \mathbb{R}$. Motivated by $\operatorname{tr}(A) \overset{!}{=} \operatorname{tr}(A_0)$ an initial choice could be $c_0 = n^{-1}\operatorname{tr}(A)$.

Finally, note that in practice we do not actually require $W_0^{\mathsf{A}}$. We only ever need access to $W_0^{\mathsf{A}} S$.

### S6.2.2 Symmetric Matrix-variate Normal Prior

We now turn to the symmetric model, which we assumed throughout the paper, given in Corollary S3. We prove Theorem 3, the main result of this section demonstrating *weak posterior correspondence* for the symmetric Kronecker covariance, by employing the matrix inversion lemma for the posterior mean $A_k$. We begin by establishing a set of technical lemmata first, which mainly expand terms appearing during matrix block inversion.

**Lemma S3** (Symmetric Posterior Inverse)
*Under the assumptions of Corollary S3, the inverse of the posterior mean is given by*

$$A_k^{-1} = A_0^{-1} - A_0^{-1} \begin{bmatrix} U_{\mathsf{A}} & V_{\mathsf{A}} \end{bmatrix} \begin{bmatrix} U_{\mathsf{A}}^{\mathsf{T}} A_0^{-1} U_{\mathsf{A}} & I + U_{\mathsf{A}}^{\mathsf{T}} A_0^{-1} V_{\mathsf{A}} \\ I + V_{\mathsf{A}}^{\mathsf{T}} A_0^{-1} U_{\mathsf{A}} & V_{\mathsf{A}}^{\mathsf{T}} A_0^{-1} V_{\mathsf{A}} \end{bmatrix}^{-1} \begin{bmatrix} U_{\mathsf{A}}^{\mathsf{T}} \\ V_{\mathsf{A}}^{\mathsf{T}} \end{bmatrix} A_0^{-1}$$

*where*

$$U_{\mathsf{A}} := W_0^{\mathsf{A}} S (S^{\mathsf{T}} W_0^{\mathsf{A}} S)^{-1} \in \mathbb{R}^{n \times k},$$
$$V_{\mathsf{A}} := (I - \frac{1}{2} U_{\mathsf{A}} S^{\mathsf{T}})(Y - A_0 S) = (I - \frac{1}{2} U_{\mathsf{A}} S^{\mathsf{T}}) \Delta_{\mathsf{A}} \in \mathbb{R}^{n \times k}.$$

*Proof.* We rewrite the rank-2 update in Section 2.1 as follows

$$A_k = A_0 + U_{\mathsf{A}} V_{\mathsf{A}}^{\mathsf{T}} + V_{\mathsf{A}} U_{\mathsf{A}}^{\mathsf{T}} = A_0 + \begin{bmatrix} U_{\mathsf{A}} & V_{\mathsf{A}} \end{bmatrix} \begin{bmatrix} 0 & I \\ I & 0 \end{bmatrix} \begin{bmatrix} U_{\mathsf{A}}^{\mathsf{T}} \\ V_{\mathsf{A}}^{\mathsf{T}} \end{bmatrix}.$$

Then the statement follows directly from the matrix inversion lemma. $\qquad\square$

Next, we expand the terms inside the blocks of the matrix to be inverted in Lemma S3. This leads to the following lemma.

**Lemma S4**
*Given the assumptions of Corollary S3, let $W_0^{\mathsf{A}}$ and $A_0$ be symmetric and assume (2) and (1) hold. Define*

$$\Lambda = S^{\mathsf{T}} W_0^{\mathsf{A}} S$$
$$\Pi = S^{\mathsf{T}} W_0^{\mathsf{A}} A_0^{-1} \Delta_{\mathsf{A}},$$

*then $\Lambda \in \mathbb{R}^{m \times m}$ and $\Lambda + \Pi \in \mathbb{R}^{m \times m}$ are symmetric and invertible and we obtain*

$$\Lambda + \Pi = S^{\mathsf{T}} W_0^{\mathsf{A}} A_0^{-1} A S = S^{\mathsf{T}} A A_0^{-1} A S = S^{\mathsf{T}} A W_0^{\mathsf{H}} A S \tag{S42}$$

$$\Pi = \Delta_{\mathsf{A}}^{\mathsf{T}} A_0^{-1} A S \tag{S43}$$

$$U_{\mathsf{A}}^{\mathsf{T}} A_0^{-1} \Delta_{\mathsf{A}} = \Lambda^{-1} \Pi \tag{S44}$$

$$\Delta_{\mathsf{A}}^{\mathsf{T}} S = S^{\mathsf{T}} \Delta_{\mathsf{A}} \tag{S45}$$

$$U_{\mathsf{A}} = A S \Lambda^{-1} \tag{S46}$$

$$U_{\mathsf{A}}^{\mathsf{T}} A_0^{-1} U_{\mathsf{A}} = \Lambda^{-1}(\Lambda + \Pi)\Lambda^{-1} \tag{S47}$$

$$I + U_{\mathsf{A}}^{\mathsf{T}} A_0^{-1} V_{\mathsf{A}} = \Lambda^{-1}(\Lambda + \Pi)(I - \frac{1}{2}\Lambda^{-1}S^{\mathsf{T}}\Delta_{\mathsf{A}}) \tag{S48}$$

$$I + V_{\mathsf{A}}^{\mathsf{T}} A_0^{-1} U_{\mathsf{A}} = (I - \frac{1}{2}\Delta_{\mathsf{A}}^{\mathsf{T}}S\Lambda^{-1})(\Lambda + \Pi)\Lambda^{-1} \tag{S49}$$

$$V_{\mathsf{A}}^{\mathsf{T}} A_0^{-1} V_{\mathsf{A}} = \Pi - \frac{1}{2}\left((\Lambda + \Pi)\Lambda^{-1}S^{\mathsf{T}}\Delta_{\mathsf{A}} + \Delta_{\mathsf{A}}^{\mathsf{T}}S\Lambda^{-1}(\Lambda + \Pi)\right) \tag{S50}$$

$$+ \frac{1}{4}\Delta_{\mathsf{A}}^{\mathsf{T}}S\Lambda^{-1}(\Lambda + \Pi)\Lambda^{-1}S^{\mathsf{T}}\Delta_{\mathsf{A}} \tag{S51}$$

*Proof.* We begin by proving that $\Lambda$ and $\Lambda + \Pi$ are symmetric and invertible. We have by Sylvester's rank inequality that $\Lambda$ is invertible. For symmetric $W_0^{\mathsf{A}}$, $\Lambda$ is symmetric by definition. We have that

$$\Lambda + \Pi = S^{\mathsf{T}} W_0^{\mathsf{A}} S + S^{\mathsf{T}} W_0^{\mathsf{A}} A_0^{-1}(AS - A_0 S) = S^{\mathsf{T}} W_0^{\mathsf{A}} A_0^{-1} AS = S^{\mathsf{T}} A A_0^{-1} AS$$

$$= S^{\mathsf{T}} W_0^{\mathsf{A}} A_0^{-1} AS = S^{\mathsf{T}} A W_0^{\mathsf{H}} AS$$

Thus, by Sylvester's rank inequality $\Lambda + \Pi$ is invertible. Given symmetric $A_0$, it is symmetric. Further, it holds that

$$\Pi = \Lambda + \Pi - \Lambda = S^{\mathsf{T}} A A_0^{-1} AS - S^{\mathsf{T}} AS = \Delta_{\mathsf{A}}^{\mathsf{T}} A_0^{-1} AS$$

$$U_{\mathsf{A}}^{\mathsf{T}} A_0^{-1} \Delta_{\mathsf{A}} = (S^{\mathsf{T}} W_0^{\mathsf{A}} S)^{-1} S^{\mathsf{T}} W_0^{\mathsf{A}} A_0^{-1} \Delta_{\mathsf{A}} = \Lambda^{-1}\Pi$$

$$\Delta_{\mathsf{A}}^{\mathsf{T}} S = (AS - A_0 S)^{\mathsf{T}} S = S^{\mathsf{T}} AS - S^{\mathsf{T}} A_0 S$$

$$U_{\mathsf{A}} = W_0^{\mathsf{A}} S (S^{\mathsf{T}} W_0^{\mathsf{A}} S)^{-1} = AS\Lambda^{-1}$$

$$U_{\mathsf{A}}^{\mathsf{T}} A_0^{-1} U_{\mathsf{A}} = \Lambda^{-1}S^{\mathsf{T}} A A_0^{-1} AS\Lambda^{-1} = \Lambda^{-1}(\Lambda + \Pi)\Lambda^{-1}$$

$$I + U_{\mathsf{A}}^{\mathsf{T}} A_0^{-1} V_{\mathsf{A}} = I + \Lambda^{-1}S^{\mathsf{T}} A A_0^{-1}(I - \frac{1}{2}U_{\mathsf{A}}S^{\mathsf{T}})\Delta_{\mathsf{A}} = I + \Lambda^{-1}S^{\mathsf{T}} A A_0^{-1}(I - \frac{1}{2}AS\Lambda^{-1}S^{\mathsf{T}})\Delta_{\mathsf{A}}$$

$$= I + \Lambda^{-1}S^{\mathsf{T}} A A_0^{-1}(AS - A_0 S) - \frac{1}{2}\Lambda^{-1}S^{\mathsf{T}} A A_0^{-1} AS\Lambda^{-1}S^{\mathsf{T}}\Delta_{\mathsf{A}}$$

$$= \Lambda^{-1}(\Lambda + \Pi) - \frac{1}{2}\Lambda^{-1}(\Lambda + \Pi)\Lambda^{-1}S^{\mathsf{T}}\Delta_{\mathsf{A}} = \Lambda^{-1}(\Lambda + \Pi)(I - \frac{1}{2}\Lambda^{-1}S^{\mathsf{T}}\Delta_{\mathsf{A}})$$

$$I + V_{\mathsf{A}}^{\mathsf{T}} A_0^{-1} U_{\mathsf{A}} = (I + U_{\mathsf{A}}^{\mathsf{T}} A_0^{-1} V_{\mathsf{A}})^{\mathsf{T}} = (\Lambda^{-1}(\Lambda + \Pi)(I - \frac{1}{2}\Lambda^{-1}S^{\mathsf{T}}\Delta_{\mathsf{A}}))^{\mathsf{T}}$$

$$= (I - \frac{1}{2}\Delta_{\mathsf{A}}^{\mathsf{T}}S\Lambda^{-1})(\Lambda + \Pi)\Lambda^{-1},$$

where we used that $\Lambda$ and $\Lambda + \Pi$ are symmetric. Finally, we have that

$$V_{\mathsf{A}}^{\mathsf{T}} A_0^{-1} V_{\mathsf{A}} = \Delta_{\mathsf{A}}^{\mathsf{T}}(I - \frac{1}{2}SU_{\mathsf{A}}^{\mathsf{T}})A_0^{-1}(I - \frac{1}{2}U_{\mathsf{A}}S^{\mathsf{T}})\Delta_{\mathsf{A}}$$

$$= \Delta_{\mathsf{A}}^{\mathsf{T}}(I - \frac{1}{2}S\Lambda^{-1}S^{\mathsf{T}}A)A_0^{-1}(I - \frac{1}{2}AS\Lambda^{-1}S^{\mathsf{T}})\Delta_{\mathsf{A}}$$

$$= \Delta_{\mathsf{A}}^{\mathsf{T}} A_0^{-1}(I - \frac{1}{2}AS\Lambda^{-1}S^{\mathsf{T}})\Delta_{\mathsf{A}} - \frac{1}{2}\Delta_{\mathsf{A}}^{\mathsf{T}}S\Lambda^{-1}S^{\mathsf{T}} A A_0^{-1}(I - \frac{1}{2}AS\Lambda^{-1}S^{\mathsf{T}})\Delta_{\mathsf{A}}$$

$$= (S^{\mathsf{T}} A A_0^{-1} - S^{\mathsf{T}})(I - \frac{1}{2}AS\Lambda^{-1}S^{\mathsf{T}})\Delta_{\mathsf{A}} - \frac{1}{2}\Delta_{\mathsf{A}}^{\mathsf{T}}S\Lambda^{-1}S^{\mathsf{T}} A A_0^{-1}\Delta_{\mathsf{A}}$$

$$+ \frac{1}{4}\Delta_{\mathsf{A}}^{\mathsf{T}}S\Lambda^{-1}S^{\mathsf{T}} A A_0^{-1} AS\Lambda^{-1}S^{\mathsf{T}}\Delta_{\mathsf{A}}$$

$$= S^{\mathsf{T}} A A_0^{-1}\Delta_{\mathsf{A}} - S^{\mathsf{T}}\Delta_{\mathsf{A}} - \frac{1}{2}S^{\mathsf{T}} A A_0^{-1} AS\Lambda^{-1}S^{\mathsf{T}}\Delta_{\mathsf{A}} + \frac{1}{2}S^{\mathsf{T}} AS\Lambda^{-1}S^{\mathsf{T}}\Delta_{\mathsf{A}}$$

$$- \frac{1}{2}\Delta_{\mathsf{A}}^{\mathsf{T}}S\Lambda^{-1}S^{\mathsf{T}} A A_0^{-1}\Delta_{\mathsf{A}} + \frac{1}{4}\Delta_{\mathsf{A}}^{\mathsf{T}}S\Lambda^{-1}S^{\mathsf{T}} A A_0^{-1} AS\Lambda^{-1}S^{\mathsf{T}}\Delta_{\mathsf{A}}$$

$$= S^{\mathsf{T}} A A_0^{-1} AS - S^{\mathsf{T}} AS - S^{\mathsf{T}}\Delta_{\mathsf{A}} - \frac{1}{2}(\Lambda + \Pi)\Lambda^{-1}S^{\mathsf{T}}\Delta_{\mathsf{A}} + \frac{1}{2}S^{\mathsf{T}}\Delta_{\mathsf{A}}$$

$$- \frac{1}{2}\Delta_{\mathsf{A}}^{\mathsf{T}}S\Lambda^{-1}S^{\mathsf{T}} A A_0^{-1}\Delta_{\mathsf{A}} + \frac{1}{4}\Delta_{\mathsf{A}}^{\mathsf{T}}S\Lambda^{-1}S^{\mathsf{T}} A A_0^{-1} AS\Lambda^{-1}S^{\mathsf{T}}\Delta_{\mathsf{A}}$$

$$= \Pi - \frac{1}{2}S^{\mathsf{T}}\Delta_{\mathsf{A}} - \frac{1}{2}(\Lambda + \Pi)\Lambda^{-1}S^{\mathsf{T}}\Delta_{\mathsf{A}} - \frac{1}{2}\Delta_{\mathsf{A}}^{\mathsf{T}}S\Lambda^{-1}S^{\mathsf{T}} A A_0^{-1}(AS - A_0 S)$$

$$+ \frac{1}{4}\boldsymbol{\Delta}_\mathsf{A}^\mathsf{T} \boldsymbol{S}\boldsymbol{\Lambda}^{-1}\boldsymbol{S}^\mathsf{T}\boldsymbol{A}\boldsymbol{A}_0^{-1}\boldsymbol{A}\boldsymbol{S}\boldsymbol{\Lambda}^{-1}\boldsymbol{S}^\mathsf{T}\boldsymbol{\Delta}_\mathsf{A}$$

$$= \boldsymbol{\Pi} - \frac{1}{2}\boldsymbol{S}^\mathsf{T}\boldsymbol{\Delta}_\mathsf{A} - \frac{1}{2}(\boldsymbol{\Lambda} + \boldsymbol{\Pi})\boldsymbol{\Lambda}^{-1}\boldsymbol{S}^\mathsf{T}\boldsymbol{\Delta}_\mathsf{A} - \frac{1}{2}\boldsymbol{\Delta}_\mathsf{A}^\mathsf{T}\boldsymbol{S}\boldsymbol{\Lambda}^{-1}(\boldsymbol{\Lambda} + \boldsymbol{\Pi}) + \frac{1}{2}\boldsymbol{\Delta}_\mathsf{A}^\mathsf{T}\boldsymbol{S}\boldsymbol{\Lambda}^{-1}\boldsymbol{\Lambda}$$

$$+ \frac{1}{4}\boldsymbol{\Delta}_\mathsf{A}^\mathsf{T} \boldsymbol{S}\boldsymbol{\Lambda}^{-1}\boldsymbol{S}^\mathsf{T}\boldsymbol{A}\boldsymbol{A}_0^{-1}\boldsymbol{A}\boldsymbol{S}\boldsymbol{\Lambda}^{-1}\boldsymbol{S}^\mathsf{T}\boldsymbol{\Delta}_\mathsf{A}$$

$$= \boldsymbol{\Pi} - \frac{1}{2}\big((\boldsymbol{\Lambda} + \boldsymbol{\Pi})\boldsymbol{\Lambda}^{-1}\boldsymbol{S}^\mathsf{T}\boldsymbol{\Delta}_\mathsf{A} + \boldsymbol{\Delta}_\mathsf{A}^\mathsf{T}\boldsymbol{S}\boldsymbol{\Lambda}^{-1}(\boldsymbol{\Lambda} + \boldsymbol{\Pi})\big) + \frac{1}{4}\boldsymbol{\Delta}_\mathsf{A}^\mathsf{T}\boldsymbol{S}\boldsymbol{\Lambda}^{-1}(\boldsymbol{\Lambda} + \boldsymbol{\Pi})\boldsymbol{\Lambda}^{-1}\boldsymbol{S}^\mathsf{T}\boldsymbol{\Delta}_\mathsf{A},$$

where we dropped some of the terms temporarily for clarity of exposition. $\qquad\square$

We will now use these intermediate results to perform block inversion on the $2k \times 2k$ matrix to be inverted in Lemma S3.

**Lemma S5**

*Given the assumptions of Corollary S3, additionally assume* (1) *and* (2) *hold. Let*

$$T = \begin{bmatrix} T_{11} & T_{12} \\ T_{21} & T_{22} \end{bmatrix} = \begin{bmatrix} \boldsymbol{U}_\mathsf{A}^\mathsf{T}\boldsymbol{A}_0^{-1}\boldsymbol{U}_\mathsf{A} & \boldsymbol{I} + \boldsymbol{U}_\mathsf{A}^\mathsf{T}\boldsymbol{A}_0^{-1}\boldsymbol{V}_\mathsf{A} \\ \boldsymbol{I} + \boldsymbol{V}_\mathsf{A}^\mathsf{T}\boldsymbol{A}_0^{-1}\boldsymbol{U}_\mathsf{A} & \boldsymbol{V}_\mathsf{A}^\mathsf{T}\boldsymbol{A}_0^{-1}\boldsymbol{V}_\mathsf{A} \end{bmatrix}^{-1},$$

*then the block matrices $\boldsymbol{T}_{ij} \in \mathbb{R}^{m\times m}$ are given by*

$$T_{11} = \boldsymbol{\Lambda}(\boldsymbol{\Lambda} + \boldsymbol{\Pi})^{-1}\boldsymbol{\Lambda} - (\boldsymbol{I} - \frac{1}{2}\boldsymbol{S}^\mathsf{T}\boldsymbol{\Delta}_\mathsf{A}\boldsymbol{\Lambda}^{-1})(\boldsymbol{I} - \frac{1}{2}\boldsymbol{\Lambda}^{-1}\boldsymbol{\Delta}_\mathsf{A}^\mathsf{T}\boldsymbol{S})$$

$$T_{12} = (\boldsymbol{I} - \frac{1}{2}\boldsymbol{S}^\mathsf{T}\boldsymbol{\Delta}_\mathsf{A}\boldsymbol{\Lambda}^{-1})$$

$$T_{21} = T_{12}^\mathsf{T} = (\boldsymbol{I} - \frac{1}{2}\boldsymbol{\Lambda}^{-1}\boldsymbol{\Delta}_\mathsf{A}^\mathsf{T}\boldsymbol{S})$$

$$T_{22} = -\boldsymbol{\Lambda}^{-1}.$$

*Proof.* Let

$$K = T^{-1} = \begin{bmatrix} \boldsymbol{U}_\mathsf{A}^\mathsf{T}\boldsymbol{A}_0^{-1}\boldsymbol{U}_\mathsf{A} & \boldsymbol{I} + \boldsymbol{U}_\mathsf{A}^\mathsf{T}\boldsymbol{A}_0^{-1}\boldsymbol{V}_\mathsf{A} \\ \boldsymbol{I} + \boldsymbol{V}_\mathsf{A}^\mathsf{T}\boldsymbol{A}_0^{-1}\boldsymbol{U}_\mathsf{A} & \boldsymbol{V}_\mathsf{A}^\mathsf{T}\boldsymbol{A}_0^{-1}\boldsymbol{V}_\mathsf{A} \end{bmatrix},$$

then the inverse of the Schur complement $D = K/(\boldsymbol{U}_\mathsf{A}^\mathsf{T}\boldsymbol{A}_0^{-1}\boldsymbol{U}_\mathsf{A})$ is given by

$$\begin{aligned}
D^{-1} &= (\boldsymbol{K}_{22} - \boldsymbol{K}_{21}\boldsymbol{K}_{11}^{-1}\boldsymbol{K}_{12})^{-1} \\
&= \big(\boldsymbol{V}_\mathsf{A}^\mathsf{T}\boldsymbol{A}_0^{-1}\boldsymbol{V}_\mathsf{A} - (\boldsymbol{I} + \boldsymbol{V}_\mathsf{A}^\mathsf{T}\boldsymbol{A}_0^{-1}\boldsymbol{U}_\mathsf{A})(\boldsymbol{U}_\mathsf{A}^\mathsf{T}\boldsymbol{A}_0^{-1}\boldsymbol{U}_\mathsf{A})^{-1}(\boldsymbol{I} + \boldsymbol{U}_\mathsf{A}^\mathsf{T}\boldsymbol{A}_0^{-1}\boldsymbol{V}_\mathsf{A})\big)^{-1} \\
&= \big(\boldsymbol{V}_\mathsf{A}^\mathsf{T}\boldsymbol{A}_0^{-1}\boldsymbol{V}_\mathsf{A} - (\boldsymbol{I} - \frac{1}{2}\boldsymbol{\Delta}_\mathsf{A}^\mathsf{T}\boldsymbol{S}\boldsymbol{\Lambda}^{-1})(\boldsymbol{\Lambda} + \boldsymbol{\Pi})(\boldsymbol{I} - \frac{1}{2}\boldsymbol{\Lambda}^{-1}\boldsymbol{S}^\mathsf{T}\boldsymbol{\Delta}_\mathsf{A})\big)^{-1} \\
&= \big(\boldsymbol{V}_\mathsf{A}^\mathsf{T}\boldsymbol{A}_0^{-1}\boldsymbol{V}_\mathsf{A} - (\boldsymbol{\Lambda} - \frac{1}{2}\boldsymbol{\Delta}_\mathsf{A}^\mathsf{T}\boldsymbol{S})\boldsymbol{\Lambda}^{-1}(\boldsymbol{\Lambda} + \boldsymbol{\Pi})\boldsymbol{\Lambda}^{-1}(\boldsymbol{\Lambda} - \frac{1}{2}\boldsymbol{S}^\mathsf{T}\boldsymbol{\Delta}_\mathsf{A})\big)^{-1} \\
&= \big(\boldsymbol{V}_\mathsf{A}^\mathsf{T}\boldsymbol{A}_0^{-1}\boldsymbol{V}_\mathsf{A} - (\boldsymbol{\Lambda} + \boldsymbol{\Pi}) + \frac{1}{2}\big(\boldsymbol{\Delta}_\mathsf{A}^\mathsf{T}\boldsymbol{S}\boldsymbol{\Lambda}^{-1}(\boldsymbol{\Lambda} + \boldsymbol{\Pi}) + (\boldsymbol{\Lambda} + \boldsymbol{\Pi})\boldsymbol{\Lambda}^{-1}\boldsymbol{S}^\mathsf{T}\boldsymbol{\Delta}_\mathsf{A}\big) \\
&\qquad - \frac{1}{4}\boldsymbol{\Delta}_\mathsf{A}^\mathsf{T}\boldsymbol{S}\boldsymbol{\Lambda}^{-1}(\boldsymbol{\Lambda} + \boldsymbol{\Pi})\boldsymbol{\Lambda}^{-1}\boldsymbol{S}^\mathsf{T}\boldsymbol{\Delta}_\mathsf{A}\big)^{-1} \\
&= (\boldsymbol{\Pi} - \boldsymbol{\Lambda} - \boldsymbol{\Pi})^{-1} \\
&= -\boldsymbol{\Lambda}^{-1},
\end{aligned}$$

where we used Lemma S4. By block matrix inversion and again with Lemma S4 we obtain

$$\begin{aligned}
T_{11} &= (\boldsymbol{U}_\mathsf{A}^\mathsf{T}\boldsymbol{A}_0^{-1}\boldsymbol{U}_\mathsf{A})^{-1} + (\boldsymbol{U}_\mathsf{A}^\mathsf{T}\boldsymbol{A}_0^{-1}\boldsymbol{U}_\mathsf{A})^{-1}(\boldsymbol{I} + \boldsymbol{U}_\mathsf{A}^\mathsf{T}\boldsymbol{A}_0^{-1}\boldsymbol{V}_\mathsf{A})D^{-1}(\boldsymbol{I} + \boldsymbol{V}_\mathsf{A}^\mathsf{T}\boldsymbol{A}_0^{-1}\boldsymbol{U}_\mathsf{A})(\boldsymbol{U}_\mathsf{A}^\mathsf{T}\boldsymbol{A}_0^{-1}\boldsymbol{U}_\mathsf{A})^{-1} \\
&= \boldsymbol{\Lambda}(\boldsymbol{\Lambda} + \boldsymbol{\Pi})^{-1}\boldsymbol{\Lambda} + \boldsymbol{\Lambda}(\boldsymbol{I} - \frac{1}{2}\boldsymbol{\Lambda}^{-1}\boldsymbol{S}^\mathsf{T}\boldsymbol{\Delta}_\mathsf{A})D^{-1}(\boldsymbol{I} - \frac{1}{2}\boldsymbol{\Delta}_\mathsf{A}^\mathsf{T}\boldsymbol{S}\boldsymbol{\Lambda}^{-1})\boldsymbol{\Lambda} \\
&= \boldsymbol{\Lambda}(\boldsymbol{\Lambda} + \boldsymbol{\Pi})^{-1}\boldsymbol{\Lambda} + (\boldsymbol{\Lambda} - \frac{1}{2}\boldsymbol{S}^\mathsf{T}\boldsymbol{\Delta}_\mathsf{A})D^{-1}(\boldsymbol{\Lambda} - \frac{1}{2}\boldsymbol{\Delta}_\mathsf{A}^\mathsf{T}\boldsymbol{S})
\end{aligned}$$

as well as

$$T_{12} = -(U_A^\mathsf{T} A_0^{-1} U_A)^{-1}(I + U_A^\mathsf{T} A_0^{-1} V_A)D^{-1}$$

$$= -\Lambda(\Lambda + \Pi)^{-1}\Lambda\Lambda^{-1}(\Lambda + \Pi)(I - \frac{1}{2}\Lambda^{-1}S^\mathsf{T}\Delta_A)D^{-1}$$

$$= -(\Lambda - \frac{1}{2}S^\mathsf{T}\Delta_A)D^{-1}$$

$$T_{21} = T_{12}^\mathsf{T} = -D^{-\mathsf{T}}(\Lambda - \frac{1}{2}\Delta_A^\mathsf{T}S)$$

and finally $T_{22} = D^{-1} = -\Lambda^{-1}$. $\qquad\qquad\square$

**Lemma S6**

*Given the assumptions of Corollary S3, additionally assume (1) and (2) hold. Let*

$$F = A_0^{-1}\begin{bmatrix}U_A & V_A\end{bmatrix}\begin{bmatrix}T_{11} & T_{12}\\T_{21} & T_{22}\end{bmatrix}\begin{bmatrix}U_A^\mathsf{T}\\V_A^\mathsf{T}\end{bmatrix}A_0^{-1},$$

*where $T$ is chosen as in Lemma S5, then if $S^\mathsf{T} A S = I$, we have*

$$F = A_0^{-1}AS(I + \Pi)^{-1}S^\mathsf{T} AA_0^{-1} - SS^\mathsf{T}.$$

*Proof.* By expanding the quadratic and using Lemma S5, we obtain the terms

$$F_{11} := A_0^{-1}U_A T_{11}U_A^\mathsf{T} A_0^{-1}$$

$$= A_0^{-1}U_A\Lambda(\Lambda + \Pi)^{-1}\Lambda U_A^\mathsf{T} A_0^{-1} - A_0^{-1}U_A(I - \frac{1}{2}S^\mathsf{T}\Delta_A\Lambda^{-1})(I - \frac{1}{2}\Lambda^{-1}\Delta_A^\mathsf{T}S)U_A^\mathsf{T} A_0^{-1}$$

$$= A_0^{-1}AS(\Lambda + \Pi)^{-1}S^\mathsf{T} AA_0^{-1} - A_0^{-1}AS\Lambda^{-1}(I - \frac{1}{2}S^\mathsf{T}\Delta_A\Lambda^{-1})(I - \frac{1}{2}\Lambda^{-1}\Delta_A^\mathsf{T}S)\Lambda^{-1}S^\mathsf{T} AA_0^{-1}$$

$$= A_0^{-1}AS(\Lambda + \Pi)^{-1}S^\mathsf{T} AA_0^{-1} - A_0^{-1}AS\Lambda^{-2}S^\mathsf{T} AA_0^{-1}$$

$$\quad + \frac{1}{2}A_0^{-1}AS\Lambda^{-1}(S^\mathsf{T}\Delta_A\Lambda^{-1} + \Lambda^{-1}\Delta_A^\mathsf{T}S)\Lambda^{-1}S^\mathsf{T} AA_0^{-1}$$

$$\quad - \frac{1}{4}A_0^{-1}AS\Lambda^{-1}S^\mathsf{T}\Delta_A\Lambda^{-2}\Delta_A^\mathsf{T}S\Lambda^{-1}S^\mathsf{T} AA_0^{-1}$$

$$F_{12} := A_0^{-1}U_A T_{12}V_A^\mathsf{T} A_0^{-1}$$

$$= A_0^{-1}U_A(I - \frac{1}{2}S^\mathsf{T}\Delta_A\Lambda^{-1})V_A^\mathsf{T} A_0^{-1}$$

$$= A_0^{-1}AS\Lambda^{-1}(I - \frac{1}{2}S^\mathsf{T}\Delta_A\Lambda^{-1})\Delta_A^\mathsf{T}(I - \frac{1}{2}SU_A^\mathsf{T})A_0^{-1}$$

$$= A_0^{-1}AS\Lambda^{-1}(I - \frac{1}{2}S^\mathsf{T}\Delta_A\Lambda^{-1})\Delta_A^\mathsf{T}(I - \frac{1}{2}S\Lambda^{-1}S^\mathsf{T} A)A_0^{-1}$$

$$= A_0^{-1}AS\Lambda^{-1}\Delta_A^\mathsf{T} A_0^{-1} - \frac{1}{2}A_0^{-1}AS\Lambda^{-1}(S^\mathsf{T}\Delta_A\Lambda^{-1}\Delta_A^\mathsf{T} + \Delta_A^\mathsf{T}S\Lambda^{-1}S^\mathsf{T} A)A_0^{-1}$$

$$\quad + \frac{1}{4}A_0^{-1}AS\Lambda^{-1}S^\mathsf{T}\Delta_A\Lambda^{-1}\Delta_A^\mathsf{T}S\Lambda^{-1}S^\mathsf{T} AA_0^{-1}$$

$$F_{21} := F_{12}^\mathsf{T} = A_0^{-1}(I - \frac{1}{2}AS\Lambda^{-1}S^\mathsf{T})\Delta_A(I - \frac{1}{2}\Lambda^{-1}\Delta_A^\mathsf{T}S)\Lambda^{-1}SAA_0^{-1}$$

$$= A_0^{-1}\Delta_A\Lambda^{-1}S^\mathsf{T} AA_0^{-1} - \frac{1}{2}A_0^{-1}(\Delta_A\Lambda^{-1}\Delta_A^\mathsf{T}S + AS\Lambda^{-1}S^\mathsf{T}\Delta_A)\Lambda^{-1}S^\mathsf{T} AA_0^{-1}$$

$$\quad + \frac{1}{4}A_0^{-1}AS\Lambda^{-1}S^\mathsf{T}\Delta_A\Lambda^{-1}\Delta_A^\mathsf{T}S\Lambda^{-1}S^\mathsf{T} AA_0^{-1}$$

$$F_{22} := A_0^{-1}V_A T_{22}V_A^\mathsf{T} A_0^{-1}$$

$$= -A_0^{-1}(I - \frac{1}{2}U_A S^\mathsf{T})\Delta_A\Lambda^{-1}\Delta_A^\mathsf{T}(I - \frac{1}{2}SU_A^\mathsf{T})A_0^{-1}$$

$$= -A_0^{-1}(I - \frac{1}{2}AS\Lambda^{-1}S^\mathsf{T})\Delta_A\Lambda^{-1}\Delta_A^\mathsf{T}(I - \frac{1}{2}S\Lambda^{-1}S^\mathsf{T} A)A_0^{-1}$$

$$= -A_0^{-1}\Delta_\mathsf{A}\Lambda^{-1}\Delta_\mathsf{A}^\mathsf{T}A_0^{-1} + \frac{1}{2}A_0^{-1}(AS\Lambda^{-1}S^\mathsf{T}\Delta_\mathsf{A}\Lambda^{-1}\Delta_\mathsf{A}^\mathsf{T} + \Delta_\mathsf{A}\Lambda^{-1}\Delta_\mathsf{A}^\mathsf{T}S\Lambda^{-1}S^\mathsf{T}A)A_0^{-1}$$

$$- \frac{1}{4}A_0^{-1}AS\Lambda^{-1}S^\mathsf{T}\Delta_\mathsf{A}\Lambda^{-1}\Delta_\mathsf{A}^\mathsf{T}S\Lambda^{-1}S^\mathsf{T}AA_0^{-1}$$

Assuming $S^\mathsf{T}AS = I$, it holds that

$$F_{11} = A_0^{-1}AS(I + \Pi)^{-1}S^\mathsf{T}AA_0^{-1} - A_0^{-1}ASS^\mathsf{T}AA_0^{-1} + \frac{1}{2}A_0^{-1}AS(S^\mathsf{T}\Delta_\mathsf{A} + \Delta_\mathsf{A}^\mathsf{T}S)S^\mathsf{T}AA_0^{-1}$$

$$- \frac{1}{4}A_0^{-1}ASS^\mathsf{T}\Delta_\mathsf{A}\Delta_\mathsf{A}^\mathsf{T}SS^\mathsf{T}AA_0^{-1}$$

$$F_{12} = A_0^{-1}ASS^\mathsf{T}AA_0^{-1} - A_0^{-1}ASS^\mathsf{T} - \frac{1}{2}A_0^{-1}AS(S^\mathsf{T}\Delta_\mathsf{A}\Delta_\mathsf{A}^\mathsf{T} + \Delta_\mathsf{A}^\mathsf{T}SS^\mathsf{T}A)A_0^{-1}$$

$$+ \frac{1}{4}A_0^{-1}ASS^\mathsf{T}\Delta_\mathsf{A}\Delta_\mathsf{A}^\mathsf{T}SS^\mathsf{T}AA_0^{-1}$$

$$F_{21} = A_0^{-1}ASS^\mathsf{T}AA_0^{-1} - SS^\mathsf{T}AA_0^{-1} - \frac{1}{2}A_0^{-1}(\Delta_\mathsf{A}\Delta_\mathsf{A}^\mathsf{T}S + ASS^\mathsf{T}\Delta_\mathsf{A})S^\mathsf{T}AA_0^{-1}$$

$$+ \frac{1}{4}A_0^{-1}ASS^\mathsf{T}\Delta_\mathsf{A}\Delta_\mathsf{A}^\mathsf{T}SS^\mathsf{T}AA_0^{-1}$$

$$F_{22} = A_0^{-1}\Delta_\mathsf{A}S^\mathsf{T} - A_0^{-1}\Delta_\mathsf{A}S^\mathsf{T}AA_0^{-1} + \frac{1}{2}(ASS^\mathsf{T}\Delta_\mathsf{A}\Delta_\mathsf{A}^\mathsf{T} + \Delta_\mathsf{A}\Delta_\mathsf{A}^\mathsf{T}SS^\mathsf{T}A)A_0^{-1}$$

$$- \frac{1}{4}A_0^{-1}ASS^\mathsf{T}\Delta_\mathsf{A}\Delta_\mathsf{A}^\mathsf{T}SS^\mathsf{T}AA_0^{-1},$$

which leads to

$$F_{11} + F_{12} = A_0^{-1}AS(I + \Pi)^{-1}S^\mathsf{T}AA_0^{-1} - A_0^{-1}ASS^\mathsf{T} + \frac{1}{2}A_0^{-1}AS(S^\mathsf{T}\Delta_\mathsf{A}S^\mathsf{T}A - S^\mathsf{T}\Delta_\mathsf{A}\Delta_\mathsf{A}^\mathsf{T})A_0^{-1}$$

$$F_{21} + F_{22} = A_0^{-1}\Delta_\mathsf{A}S^\mathsf{T} + \frac{1}{2}A_0^{-1}AS(S^\mathsf{T}\Delta_\mathsf{A}\Delta_\mathsf{A}^\mathsf{T} - S^\mathsf{T}\Delta_\mathsf{A}S^\mathsf{T}A)A_0^{-1}$$

$$= A_0^{-1}ASS^\mathsf{T} - SS^\mathsf{T} + \frac{1}{2}A_0^{-1}AS(S^\mathsf{T}\Delta_\mathsf{A}\Delta_\mathsf{A}^\mathsf{T} - S^\mathsf{T}\Delta_\mathsf{A}S^\mathsf{T}A)A_0^{-1}.$$

Finally, adding up the individual terms we obtain

$$F = F_{11} + F_{12} + F_{21} + F_{22} = A_0^{-1}AS(I + \Pi)^{-1}S^\mathsf{T}AA_0^{-1} - SS^\mathsf{T}.$$

$\square$

**Theorem 2** (Weak Posterior Correspondence)
*Let $W_0^\mathsf{H} \in \mathbb{R}_{\mathrm{sym}}^{n \times n}$ be positive definite. Assume $H_0 = A_0^{-1}$, and that $W_0^\mathsf{A}, A_0, W_0^\mathsf{H}$ satisfy (1) and (2), then weak posterior correspondence holds for the symmetric Kronecker covariance.*

*Proof.* First note that without loss of generality $S^\mathsf{T}AS = I$, i.e. only the direction of the action matters in Algorithm 1 not its magnitude. This can be seen from the forms of $A_k$ and $H_k$ in Section 2.1. Any positive factor $\alpha > 0$ of $s_k$ cancels in the update expressions. Expanding the right hand side we have using (S33), that $H_k Y = S$. Then by Lemma S3, Lemma S6 and $S^\mathsf{T}AS = I$, the left hand side evaluates to

$$A_k^{-1}Y = (A_0^{-1} - F)Y$$
$$= (A_0^{-1} - A_0^{-1}AS(I + \Pi)^{-1}S^\mathsf{T}AA_0^{-1} + SS^\mathsf{T})AS$$
$$= A_0^{-1}AS - A_0^{-1}AS + S$$
$$= S$$
$$= H_k Y.$$

This concludes the proof. $\square$

This theorem shows that for a certain choice of symmetric matrix-variate normal prior the estimated inverse of the matrix $H_k$ corresponds to the inverse of the estimated matrix $A_k^{-1}$. It also shows that both act like $A^{-1}$ on the space spanned by $Y$, consistent with the interpretation of the two being the best guess for the inverse $A^{-1}$.

## S7 Galerkin's Method for PDEs

In the spirit of applying machine learning in the sciences [22], we briefly outlined an application of Algorithm 1 to the solution of partial differential equations in Section 4. As an example we considered the Dirichlet problem for the Poisson equation given by

$$\begin{cases} -\Delta u(x,y) = f(x,y) & (x,y) \in \text{int}\,\Omega \\ u(x,y) = u_{\partial\Omega}(x,y) & (x,y) \in \partial\Omega \end{cases} \tag{S52}$$

where $\Omega$ is a connected open region with sufficiently regular boundary and $u_{\partial\Omega} : \partial\Omega \to \mathbb{R}$ defines the boundary conditions. The corresponding weak solution of (S52) is given by $u \in V$ such that for all test functions $v \in V$

$$a(u,v) := \int_\Omega \nabla u \cdot \nabla v \, dx = \int_\Omega fv \, dx =: f(v), \tag{S53}$$

where $a(\cdot,\cdot)$ is a bilinear form. Next, one derives the *Galerkin equation* by choosing a finite-dimensional subspace $V_\square \subset V$ and corresponding basis $e_1^\square, \ldots, e_n^\square$. Then (S53) reduces to finding $u \in V_\square$ such that for all $i \in \{1, \ldots, n\}$ it holds that $a(u, e_i^\square) = \sum_{j=1}^n u_j a(e_j^\square, e_i^\square) = f(e_i^\square)$ which is a linear system $\boldsymbol{A}\boldsymbol{u} = \boldsymbol{f}$ with the entries of the Gram matrix given by $\boldsymbol{A}_{ij} = a(e_j^\square, e_i^\square)$ and $f_i = f(e_i^\square)$.

### S7.1 Operator View

The operator view provides another motivation for placing a distribution over the matrix $\boldsymbol{A}$ of a linear system. When approximating the solution to a PDE, as we do here, then solution-based inference for linear systems [13, 14] can be viewed as placing a Gaussian process prior over the solution $u : \Omega \to \mathbb{R}$ [23]. The matrix-based approach [6] instead can be interpreted as placing a Gaussian measure [24] on the infinite-dimensional space of the differential operator instead. This induces a Gaussian distribution on the Gram matrix $\mathbf{A}$ modelling the uncertainty about the actions of the (discretized) differential operator.

**Definition S3** (Infinite-dimensional Gaussian Measures [24])
Let $W$ be a topological vector space with Borel probability measure $\mu$, then $\mu$ is Gaussian, iff for each continuous linear functional $f \in W^*$, the pushforward $\mu \circ f^{-1}$ is a Gaussian measure on $\mathbb{R}$, i.e. $f$ is a Gaussian random variable on $(W, \mathcal{B}_W, \mu)$.

This definition and further detail on Gaussian measures in infinite-dimensional spaces can be found in the book by Bogachev [24]. We now model the differential operator as a random variable on the space of bounded linear operators and show that this induces a distribution on the Gram matrix arising from discretization via Galerkin's method.

**Theorem S3** (Gaussian Measures on the Space of Bounded Linear Operators)
*Let $V$ be a Hilbert space and let $W = B(V,V)$ be the space of bounded linear operators from $V$ to $V$ with Borel probability measure $\mu$ and let $\mathsf{A}$ be a Gaussian random variable on $(W, \mathcal{B}_W, \mu)$. Consider the operator equation*

$$\mathsf{A}u = \mathsf{f}$$

*and let $a : V \times V \to \mathbb{R}, (u,v) \mapsto \langle \mathsf{A}u, v \rangle_V = \langle \mathsf{f}, v \rangle_V$ be its corresponding bilinear form. Let $V_\square$ be an $n$-dimensional subspace of $V$, then the resulting Gram matrix $\mathbf{A} \in \mathbb{R}^{n \times n}$ is matrix-variate Gaussian.*

*Proof.* Since $V$ is Banach, so is $W$. Define the functional $a_W : W \to \mathbb{R}$ given by $a_W(\mathsf{A}, u, v) = a(u,v)$ for fixed $u, v \in V$. The map $a_W(\cdot, u, v)$ is linear by linearity of the inner product and bounded since using the Cauchy-Schwarz inequality, it holds that

$$|a_W(\mathsf{A}, u, v)| = |\langle \mathsf{A}u, v \rangle_V| \le \|\mathsf{A}u\|_V \|v\|_V \le \|\mathsf{A}\|_W \|u\|_V \|v\|_V = C\|\mathsf{A}\|_W.$$

Therefore $a_W(\cdot, u, v) \in W^*$ for all $u, v \in V$. By Definition S3 of a Gaussian measure the push forward $\mu \circ a_W^{-1}$ is a Gaussian measure on $\mathbb{R}$ for all $u, v \in V$, in particular also for a basis $\{v_i\}_{i=1}^n$ of $V_\square$. Therefore the Gram matrix $\mathbf{A}$ given by $\mathbf{A}_{ij} = a(v_i, v_j) = a_W(\mathsf{A}, v_i, v_j)$ is matrix-variate Gaussian since its components are Gaussian. $\qquad\square$

**Remark S1**
The Laplacian $\Delta : H^2(\Omega) \to L^2(\Omega)$ is a bounded linear operator on the Sobolev space $H^2(\Omega)$. Note, that in general differential operators are in fact *not bounded*. Hence, the simple argument above does not generalize to arbitrary differential operators.

**Remark S2**
If the bilinear form $a$ in addition to being continuous is also weakly coercive, then by the Lax-Milgram theorem the operator equation has a unique solution. A symmetric and weakly coercive operator implies a symmetric positive-definite Gram matrix.

## S7.2 Discretization Refinement

The linear system $\boldsymbol{Au} = \boldsymbol{f}$ arises from discretizing (S52) using Galerkin's method on a given mesh $\square$ defined via a finite-dimensional subspace $V_\square \subset V$ such that $\boldsymbol{u} \in V_\square$. By solving this problem using a probabilistic linear solver we obtain a posterior distribution over the inverse $\boldsymbol{H}$ of the discretized differential operator $\boldsymbol{A}$. Our goal is to leverage the obtained information about the solution on the coarse mesh to extrapolate to a refined discretization, similar in spirit to multi-grid methods [25]. This approach can be seen as an instance of transfer learning and could be used for adaptive probabilistic mesh refinement strategies based on the uncertainty about the solution in a certain region of the mesh.

Consider a fine mesh $\boxplus$ given by $V_\boxplus$, where $n_\boxplus = \dim(V_\boxplus) > \dim(V_\square) = n_\square$ such that $V_\square \subset V_\boxplus \subset V$. We would like to transfer information from solving the problem on the coarse mesh $V_\square$ to the solution of the discretized PDE on the fine mesh $V_\boxplus$. To do so we compute the predictive distribution on the fine mesh, given the belief over the inverse differential operator on the coarse mesh, i.e.

$$p(\mathbf{H}_\boxplus) = \int p(\mathbf{H}_\boxplus \mid \mathbf{H}_\square) p(\mathbf{H}_\square)\, d\mathbf{H}_\square.$$

Define the *prolongation operator* $\boldsymbol{P} : \mathbb{R}^{n_\square} \to \mathbb{R}^{n_\boxplus}$ given by $\boldsymbol{P}_{ij} = \langle e_i^\boxplus, e_j^\square \rangle$ satisfying $\boldsymbol{P}^\mathsf{T}\boldsymbol{P} = \boldsymbol{I} \in \mathbb{R}^{n_\square \times n_\square}$, implying it is injective. The distribution over the inverse operator on the fine mesh given the inverse operator on the coarse mesh is given by

$$p(\mathbf{H}_\boxplus \mid \mathbf{H}_\square) = \mathcal{N}(\mathbf{H}_\boxplus; \boldsymbol{P}\mathbf{H}_\square\boldsymbol{P}^\mathsf{T}, \boldsymbol{\Lambda}) \tag{S54}$$

where $\boldsymbol{\Lambda} \in \mathbb{R}^{n_\boxplus \times n_\boxplus}_{\text{sym}}$ positive definite models the numerical uncertainty induced by the coarser discretization. This corresponds to the assumption that solving the problem on a coarser grid approximates the solution on a fine grid projected to the coarse grid.

Now assume we have a posterior distribution over the inverse differential operator on the coarse grid from a solve of the coarse problem using Algorithm 1, given by

$$p(\mathbf{H}_\square) = \mathcal{N}(\mathbf{H}_\square; \boldsymbol{H}_\square^k, \boldsymbol{W}_\square^k \otimes \boldsymbol{W}_\square^k).$$

The projection in (S54) is a linear map, since by the characteristic property of the Kronecker product (S1) we have

$$\operatorname{svec}(\boldsymbol{P}\mathbf{H}_\square\boldsymbol{P}^\mathsf{T}) = \boldsymbol{Q}(\boldsymbol{P} \otimes \boldsymbol{P})\boldsymbol{Q}^\mathsf{T}\operatorname{svec}(\mathbf{H}_\square).$$

Therefore by Theorem S1 the predictive distribution is also closed-form and Gaussian.

**Proposition S6** (Predictive Distribution on Fine Mesh)
*Let $p(\mathbf{H}_\square) = \mathcal{N}(\mathbf{H}_\square; \boldsymbol{H}_\square^k, \boldsymbol{W}_\square^k \otimes \boldsymbol{W}_\square^k)$ be a prior on $\mathbf{H}_\square$ and assume a likelihood of the form* (S54). *Then the predictive distribution is given by $p(\mathbf{H}_\boxplus) = \mathcal{N}(\mathbf{H}_\boxplus; \boldsymbol{H}_\boxplus^0, \boldsymbol{\Sigma}_\boxplus^0)$, where*

$$\boldsymbol{H}_\boxplus^0 = \boldsymbol{P}\boldsymbol{H}_\square^k\boldsymbol{P}^\mathsf{T},$$
$$\boldsymbol{\Sigma}_\boxplus^0 = \boldsymbol{P}\boldsymbol{W}_\square^k\boldsymbol{P}^\mathsf{T} \otimes \boldsymbol{P}\boldsymbol{W}_\square^k\boldsymbol{P}^\mathsf{T} + \boldsymbol{\Lambda}.$$

*Proof.* By Theorem S1 we obtain for the mean and covariance of the predictive distribution

$$\boldsymbol{H}_\boxplus^0 = \boldsymbol{P}\boldsymbol{H}_\square^k\boldsymbol{P}^\mathsf{T}$$
$$\boldsymbol{\Sigma}_\boxplus^0 = \boldsymbol{Q}(\boldsymbol{P} \otimes \boldsymbol{P})\boldsymbol{Q}^\mathsf{T}(\boldsymbol{W}_\square^k \otimes \boldsymbol{W}_\square^k)\boldsymbol{Q}(\boldsymbol{P}^\mathsf{T} \otimes \boldsymbol{P}^\mathsf{T})\boldsymbol{Q}^\mathsf{T} + \boldsymbol{\Lambda}$$
$$= \frac{1}{2}\boldsymbol{Q}(\boldsymbol{P}\boldsymbol{W}_\square^k\boldsymbol{P}^\mathsf{T} \otimes \boldsymbol{P}\boldsymbol{W}_\square^k\boldsymbol{P}^\mathsf{T} + \boldsymbol{P}\boldsymbol{W}_\square^k\boldsymbol{P}^\mathsf{T} \boxtimes \boldsymbol{P}\boldsymbol{W}_\square^k\boldsymbol{P}^\mathsf{T})\boldsymbol{Q}^\mathsf{T} + \boldsymbol{\Lambda}$$
$$= \boldsymbol{P}\boldsymbol{W}_\square^k\boldsymbol{P}^\mathsf{T} \otimes \boldsymbol{P}\boldsymbol{W}_\square^k\boldsymbol{P}^\mathsf{T} + \boldsymbol{\Lambda}$$

where we used (S31) and the symmetry of $\boldsymbol{W}_\square^k$. $\qquad\square$

For general $\boldsymbol{\Lambda}$ the covariance of the predictive distribution does not have symmetric Kronecker form, making its use as a prior for a new solve on the fine mesh challenging. We aim to exploit structural assumptions on $\boldsymbol{\Lambda}$ and results on nearest Kronecker products to a sum of Kronecker products to remedy this shortcoming in the future.