[Reviews · NeurIPS 2020]

Review 1

Summary and Contributions: This paper introduces an algorithm for solving linear systems Ax = b, which has a probabilistic Bayesian interpretation. Thus, it has posterior distributions for A, A^{-1} and x. The algorithm is proven to collapse the posterior to the true value of these quantities in n iterations if A \in R^{n x n}, and for some choices of prior is equivalent to the conjugate gradients algorithm. The uncertainty in the posterior of these quantities can then be transmitted downstream to the algorithm that uses a linear solver.

Strengths: EDIT after rebuttal: Thank you authors for clarifying the following: - GP regression on log(Rayleigh_i): satisfactory reply, this algorithm takes into account uncertainty about eigenvalues beyond t+1. - Transfer learning: reusing the posterior covariance as a prior makes the method converge faster than if just the mean is reused. I'm still confused about this, but a little bit less: - Empirical Bayes: is indeed common, and in many applications the prior is updated as more data comes in. For example, in Bayesian optimization, after acquiring an extra point the GP hyperparameters are re-optimized. However, the weird thing here, which the authors clarified in the rebuttal, is that the prior used at each time step *contains future observations in it*. Does this imply that the posterior covariance is impossible to calculate in the middle of the algorithm, before it is terminated and thus we have the full S matrix? This is weird enough that I think it should be at least pointed out. Or am I still misunderstanding things? ----- The paper is very relevant to the NeurIPS community, since many of our algorithms use linear solvers (GPs, kernel methods, for example). Ali Rahimi, in his 2017 NeurIPS test of time award presentation (with Ben Recht), made a call for "approximate, fast linear algebra methods", which this paper is answering. Its philosophy of quantifying numerical uncertainty probabilistically also follows that of Michael Osborne (see his PhD thesis) and others working on "probabilistic numerics". The algorithm is provably sound, as fast and accurate as the current state of the art, and it seems to be calibrated in its error estimation. It also provides an estimate of A^{-1} (the inverse matrix) in addition to x = A^{-1}b (the solution vector), which conjugate gradients does not. However, it is possible to obtain an approximation to the eigendecomposition given the output of conjugate gradients, which is as good as A^{-1} for most purposes. Very good assessment of the broader impact!

Weaknesses: I think this work is good, and addresses an excellent research question. I can see one weakness in the Bayesian interpretation of this algorithm, which may just be a misunderstanding. Naively, the view of how the proposed method works is as follows: a priori, we specify a Matrix-variate normal distribution for the matrix A and its inverse. During the course of the algorithm, these two distributions get iteratively updated using Bayesian conditioning, with the observations y_i = A s_i. A calibration correction is applied using \phi and \psi. However, in equation (3), the prior matrix-variances are specified using the observations Y and S. Does this mean that they get updated at every iteration? If so, a reasonable interpretation of this is that a new prior is specified at every iteration, and then its corresponding posterior from Section 2.1 is calculated. This is, in a way, Bayesian, since the output of the algorithm is a probability distribution describing its belief. However, it does not behave as we would expect it to when specifying a prior for A and A^{-1}. To have a sound Bayesian interpretation of this algorithm, a new prior and likelihood should be devised, that removes the need for updating the prior at every iteration. Additionally, the authors state that the uncertainty in the posterior over x (obtained, I believe, from the posterior over A^{-1}, and the known b) is miscalibrated, and has to be calibrated using another algorithm. For example, GP regression on the Rayleigh coefficient. My question is: what is the advantage of this algorithm over the following procedure?: Run conjugate gradients, and at iteration t, estimate x as a Gaussian distribution with mean \hat{x}, its CG estimate, and variance span(S)^\perp * (output of the GP at t+1) ? Relatedly: in terms of transfer learning for Galerkin's method, why is it not as good to reuse the previous estimate of x as the initial guess for your CG ? I'm fairly confident the authors will satisfactorily rebut these last two paragraphs, but also I believe it would make the paper better to include the answer to it. I'm looking forward to reading this rebuttal, and wholeheartedly giving this paper a "7: Good paper, accept".

Correctness: As far as I can tell, yes. I have not checked the proofs in detail.

Clarity: It is written rather well, but needs some clarifications: - Line 93: why is the action s_i chosen? The paper should redirect readers to S4.4, or include a 1-sentence explanation. - Line 89: desidarata -> desiderata - In footnote 3, page 7: do you mean that calculating the ground truth that you compare your solver to, is too expensive? If so, it would be good to be explicit - If the prior isn't updated at every iteration (eq. 3), then this was not clear to me.

Relation to Prior Work: Yes, though I do not know the literature well.

Reproducibility: Yes

Additional Feedback: -Line 134: note that the GP regression in Section 3 can be performed in O(1) per iteration by approximating it with a Kalman filter. The approximation is exact for the Matérn family of kernels [SHT, NeurIPS 2018]. The approximation is possible because the input, which is the iteration number t, is 1-dimensional. [SHT 2018] http://papers.nips.cc/paper/7608-infinite-horizon-gaussian-processes


Review 2

Summary and Contributions: This paper develops a new probabilistic linear solver to address many of the gaps in the literature on such solvers. To accomplish this, the authors begin by compiling a list of desired properties of such solvers before introducing their proposed solver which is designed to fulfil as many of these properties as possible. The proposed solver builds on existing probabilistic linear solvers by following a matrix-based perspective as introduced in [11], but updates the earlier method in several key ways. First, the solver is designed to infer both A, A^{-1} and x* (the solution to a linear system Ax = b) simultaneously in a consistent fashion. Second, the authors propose a method for calibrating the uncertainty of the output distribution to ensure that it is a useful representation of the uncertainty which remains in the unknown solution x*. The authors evaluate the new method on two test problems from machine learning / applied mathematics. The first is the computation of the posterior mean and covariance for a Gaussian process, and the second is the solution of Poisson's equation after discretisation with the finite element method.

Strengths: The work is novel and represents a clear contribution to the literature on probabilistic linear solvers. The statement of desiderata in Table 1 is in itself novel. Providing an algorithm for inference of both A and H = A^{-1} with guarantees on correspondence between the two in a particular, well-defined sense is also novel, as is the work on uncertainty calibration. While separate inference algorithms have been provided for A and H in the past, the algorithm presented in Algorithm 1 covers calibration, consistent inference and termination criteria in a self-consistent fashion. The paper reads as a comprehensive resource for probabilistic numerical methods for linear systems in the matrix-based view. The theoretical results look sound and I didn't spot any errors in the proofs. The experimental evaluation covers two challenging problems. While a more detailed evaluation would be nice to see, given the limit on page length I believe the evaluation strikes a good balance between length and detail. The PDE example is particularly interesting as matrix-based methods have not been applied to such problems before, to my knowledge. The NeurIPS community has shown an interest in similar probabilistic numerical methods in the past, and given how important linear solvers are in machine learning and applied mathematics, work on such solvers is highly relevant.

Weaknesses: The authors suggest in Section 4 some advantages of the approach that are not explored in the numerical experiments, specifically subspace recycling and an application to adaptive mesh refinement. Unless I have missed this application, I would suggest that they add a comment to the effect that this is likely to be a subject of further work. Related to this, the algorithm is dependent on a specific choice of right-hand-side b, to generate the directions (S, Y). The impact of this is explored implicitly in Section 4, in that when the posterior on A^{-1} is used to compute the covariance of the GP regressor it will be applied to vectors other than the data y. However since the posteriors and the uncertainty calibration each depend upon this right hand side, I feel that a comment on the impact of this should be added in regard to the potential application to subspace recycling.

Correctness: I disagree with the authors over the "tick" next to point (5) in Table 1. While the priors over A and H do correspond in a weak sense that the authors define concretely later in the paper, since this has not yet been introduced I would argue that "corresponding" here is easily misinterpreted to mean that p(H) is induced from p(A). I would suggest that this "tick" be replaced with a ~ for clarity.

Clarity: The paper is extremely well written.

Relation to Prior Work: The novelty of the contribution is clearly stated in the introduction and previous work is well cited in Section 2.4.

Reproducibility: Yes

Additional Feedback:


Review 3

Summary and Contributions: Given a symmetric positive definite linear system A x_* = b, the paper lays out a framework for inference about x_* and H = A^(-1), given that linear measurements Y = A S are done, where the s_j are sequentially chosen. The resulting algorithm class mimics iterative linear solvers (like P-CG and Lanczos), but also provides certain posterior covariances at the end, which could be useful as additional output. The algorithm can recover the CG sequence. Its cost is more like Lanczos with re-orthogonalization, but the authors claim that their algo also maintains conjugacy. For some reason, the algorithm maintains posteriors over A and H, and they provide some choice of prior covariance under which these posteriors are linked in a reasonable way. They also provide a means of uncertainty calibration. Finally, for two relevant linear systems problems, they show they can obtain uncertainties about the solution. I read the author feedback. They acknowledge their proposed prior changes with the data, but point out this is normal for empirical Bayesian procedures. Also, they point out a practical value for calibration (termination criterion). These points should be in the main text.

Strengths: This is a clean and methodological paper. It contains seriously difficult and interesting concepts from linear algebra. It seems the culmination of a long period (mostly work with Hennig being involved). Certainly, the probabilistic perspective on linear solvers is very inspiring. The authors also provide code for their methods, which is likely open sourced. The paper comes with a long and carefully written Appendix (supplemental material), where many gaps in the paper are filled. This paper scores high for originality and methodological strength.

Weaknesses: First, while the writing is methodological, it is not always clear, and partly hard to understand. The authors hardly give intuitions or try to explain things that arise. Here are some: - Why even maintain p(A), when you maintain p(H), and only use E[H] in your algo? You'd then not have to worry about them remaining related in some sense - No intuition is given about the "uncertainties" coming out of this, and in particular how they relate to the choice of prior covariances. Given these are n-by-n, they have many more DoF's than all the measurements. What are reasonable choices? - Your algorithm is as expensive as Lanczos with re-orthog, both in time and memory. With Lanczos, I can obtain eigenvalues and eigenvectors. Can I get them here as well? - The choice of prior cov in (3) is very confusing. This needs all of S and Y, which only become available during the algo, and they themselves depend on W_0^H due to MVM with E[H]. This is circular, unless maybe S, Y are growing, in which case this is not a prior, because it changes all the time. - This "uncertainty calibration" sounds really interesting, and I think it is important, but the authors do not explain for what. Does this help to maintain conjugacy (as in Lanczos)? Or is this just about the final posterior covariances, and does not affect the x_j sequence? What is the real problem posed by miscalibration? What happens if I do not do it? Most important, I am not convinced about the applications. Given that papers on probabilistic linear solvers go back quite a while, and the algos are more expensive and complex, what is the real gain in practice? I am sure there is some, but it should be stated in the paper. For the GP regression example, I did not understand why we need this type of uncertainty. Posterior variances can also be estimated by Lanczos, see GPyTorch. Is what you are doing better, or cheaper?

Correctness: The details are too involved to fully check, but given the supplemental material, I do have a high trust in most of this being correct. But the choice of prior in (3) does not make sense to me (see above), while it may be formally correct, it remains not realizable (unless I missed something).

Clarity: The writing is very methodological, and mostly the writing is very clear. The authors clearly put a large amount of effort into this. The figures are very nice, and I especially like the list of desiderata. Having said that, the paper sometimes glosses over obvious questions (see above). They could improve it by being a little more sympathetic to the reader not so into the details.

Relation to Prior Work: The paper cites a lot of prior work, this is very good. In most of these [18, 11, 28, 13], Hennig is involved. The present paper seems a bit like bringing this all together in one place. The authors do not clearly say what is really novel here over this work.

Reproducibility: Yes

Additional Feedback: - It would be nice to explain where re-orthogonalization happens. My suspicion is when you multiply with E[H] - Please make some effort to in particular explain this calibration better. I sense it is important, and I'd be very interested in how you make use of known kernel eigenvalue decays to somehow gain an advantage in your algorithm


Review 4

Summary and Contributions: The paper proposes a new approach for a probabilistic linear solver. The key contribution is to construct a prior that simultaneously renders both the covariance and precision posteriors tractable. The algorithm recovers CG as a special case, and can incorporate prior information via a matrix of free parameters. As the range of prior information that can be incorporated is quite limited (necessary to retain a tractable algorithm), the posterior is not perfectly calibrated. Several calibration methods are investigated and discussed.

Strengths: The idea is a good one and it adds a new tool to one of the most fundamental problems in ML. The prior has a full N^2 set of parameters, potentially allowing the incorporation of quite specific prior information to the solver. The method recovers a the classical algorithm so can be seen as a strict improvement over existing methods. The method is relevant for many aspects of machine learning. The GP approximation community might be particularly interested in light it might shed on truncated CG algorithms.

Weaknesses: The construction requires a prior that does not reflect actual prior beliefs, e.g. positive definiteness is not encodable. There are many competing methods that make approximations for convenience, e.g. low rank or local methods, that may work well in some settings but not others. These methods are not that much discussed. In the GP kernel inversion problem there is an extensive literature on reducing the computation through approximation. For example, VI with inducing points has the same complexity as the presented algorithm, and the approximation error is faithfully propagated to the predictive distribution.

Correctness: Paper seems correct. The last experiment is very hastily described so it is not clear what has been done, however.

Clarity: While the quality of the writing is generally high, I have concerns about the positioning of the method and its application. My concern can be illustrated with the claim in line 20 that linear systems solvers are ‘subject to noise’. I find this quite confusing. There might be circumstances when the arithmetic itself is significantly noisy, but this isn’t common in the usual computing paradigm. If it meant to refer to floating point error or random bit flips then that is a particular imprecision of computation that has an extensive literature that is not discussed and does not feature in the experiments. If the noise is part of the model then the problem is not one of PN but just one of inference in a probabilistic model. In line 59 (‘noise corrupted’) and then in 100 (’noisy setting’) there further mentions of noise, but without it being clear where this noise is coming from. As the method does not actually deal with noisy observations (at least, not tractably) is seems unnecessary to keep referring to it. Later in the paper the method is motivated to quantify uncertainty ‘induced by finite computation, solve multiple consecutive linear systems, and propagate information between problems’. Only the first two of these are actually demonstrated in the experiments, and the the second is touched on in so briefly that it is unclear exactly what is done.

Relation to Prior Work: While prior work in PN is appropriately referenced, the specific contributions of the paper are not very clearly delineated and related methods are under-discussed. A small point: it is confusing that the paper claims in the introduction to offer a unifying view, but cites a paper with the title “Probabilistic linear solvers: A unifying view” without making it clear what the previous unifying view missed. Also I note that previous work (Hennig [2015] and Cockayne et al. [2018]) has also recovered the CG algorithm, but this fact is not made clear.

Reproducibility: Yes

Additional Feedback: I think this method has the potential to make a very strong paper, but the current work suffers from lack of focus on what the practical use is intended to be, and lack of comparison with competing approaches. To be clear, I don’t mean that the paper necessarily needs many columns of results in the experiments section, but that it needs to be clear exactly which problem the method is attempting to solve and why it might offer an advantage over existing work, rather than the vague catch-alls given at various points in the paper. From looking at the experiments it seems that the emphasis is in on settings when the number of iterations required for the exact solution is considered prohibitive. I.e. with computation complexity O(N^2M), where M<<N. In the context of the Gram matrix inversion for a GP there is a wide literature of alternative methods of the same scaling (e.g. inducing points with VI, as mentioned above) that also quantify the approximation error. (As a side note, the presented method does not compute logdet terms in the marginal likelihood, so might be of limited use for practitioners. A note in favour of the method, however, is that it does not necessarily require computing all elements of the NxN Gram matrix, which in practice is the bottleneck for these large-scale applications (e.g Exact Gaussian Processes on a Million Data Points, Wang et al 2019)). Despite my complaint about the motivation and focus of the paper, I share the belief of PN that the predominant paradigm of treating numeric routines as black boxes is suboptimal. I do believe that resources would be better allocated by treating finite computation as part of the model, and that computation should only be expended to reduced uncertainty as required for a downstream task. This presented paper does not affirm this view very strongly, however, and a reader less convinced in the vision of PN might not see the value in what has been done. Small points: Must /Phi and /Psi be symmetric? ‘Previous works generally committed to either one view or the other’ I would be better to have these spelled out. ‘more sophisticated schemes become computationally feasible since their cost amortizes’. This could do with some more detail. The same goes for the last experiment: it isn’t clear to me exactly what has been done. EDIT: raised a point after rebuttal and discussion with other reviewers.

[Author Response · NeurIPS 2020]

We thank the reviewers for their insightful feedback! We are encouraged they recognize the importance of probabilistic linear solvers (PLS) for ML (R1, R2, R4) and the need to quantify uncertainty arising from finite computation (R1, R4).

**Contributions and Novelty** We are pleased the reviewers appreciate the originality of this work (R2, R3), its methodological contributions (R1, R2, R3), the stipulated desiderata marking a roadmap for PLS research (R2, R3), the value of the proposed prior class (R2, R4), and the importance of the novel uncertainty calibration procedure (R2, R3). It remedies a primary shortcoming of PLS, enables probabilistic stopping criteria and estimates $\log(\det(\boldsymbol{A}))$ (see below). We also provide the first practical open-source implementation of a PLS returning distributions over $\boldsymbol{A}$, $\boldsymbol{A}^{-1}$ and $\boldsymbol{x}$. This is an important step towards a framework which targets computational resources "to reduce uncertainty as required for a downstream task" (R4). The returned posterior means by the PLS are low-rank approximations to $\boldsymbol{A}$ and $\boldsymbol{A}^{-1}$, which among others find use in kernel methods. The returned covariances provide a bound to the error and their structure can be exploited for novel use cases, e.g. as proposed for probabilistic mesh refinement in Galerkin's method.

**Bayesian Interpretation and Prior Class (R1, R3)** The generic inference procedure in Section 2 for a given prior covariance $\boldsymbol{W}_0^{\mathsf{A}} \otimes \boldsymbol{W}_0^{\mathsf{A}}$ is Bayesian since it relies on Bayes' theorem. Algorithm 1 performs sequential Bayesian updates for single action - observation pairs $(\boldsymbol{s}_i, \boldsymbol{y}_i)$. This can be seen by recognizing that the posterior (see Section 2.1) is of the same form as the prior (for any $1 \leq k \leq n$). Guided by the desiderata in Table 1, we restrict the $n \times n$ DoFs in the prior. This results in the proposed prior class in eq. (3). This prior and our calibration procedure depend on the entire collected 'data' during a run of Algorithm 1. When using the proposed prior class our method is thus not strictly Bayesian in the philosophical sense, but empirical Bayesian (i.e. it uses data to fit hyperparameters of the prior). As this approach is standard in GP regression (where kernel parameters are set by type-II maximum likelihood), we neglected to make this distinction. We will clarify this in the final version. This leaves the question how the algorithm is realizable for the proposed prior (3) given its dependence on future data. The posterior mean in Section 2.1 only depends on $\boldsymbol{W}_0^{\mathsf{A}} \boldsymbol{S} = \boldsymbol{Y}$ *not* on $\boldsymbol{W}_0^{\mathsf{A}}$ alone. By eq. (3), this product is given by the previously made observations $\boldsymbol{Y}$. Similar reasoning applies for the inverse. Now, the posterior covariances do depend on $\boldsymbol{W}_0^{\mathsf{A}}$, resp. $\boldsymbol{W}_0^{\mathsf{H}}$ alone, but during a run of Algorithm 1, we only require $\mathrm{tr}(\mathrm{Cov}[\mathbf{x}])$ for the stopping criterion. We show in Section S4.5 under the assumptions of Theorem 2 how to compute this at any iteration $i$ without access to future actions and observations.

**Calibration Procedure (R1, R3)** Calibration ensures that the uncertainty returned by the solver has the right scale, i.e. it bounds the expected (relative) error (see Sections 2.2 and S4.5). Since the policy $\pi$ only depends on the posterior mean $\mathbb{E}[\mathbf{H}]$ and not the covariance, the hyperparameters $\boldsymbol{\Phi}, \boldsymbol{\Psi}$ and thus calibration do *not* change the solution estimate $\boldsymbol{x}_i$ only its associated covariance at iteration $i$. While structure in the uncertainty is preserved, miscalibration negatively impacts the probabilistic termination criterion. In our experiments in Table 2 the solver without calibration terminates early, since it is overconfident and thus has larger error than with calibration. *Why not return $\mathcal{N}(\hat{x}^{\mathrm{CG}}, \mathrm{span}(\boldsymbol{S})^{\perp}(GP\ output\ at\ t+1))$?* (R1): This is similar to Algorithm 1's output assuming Theorem 2 holds. However, it omits prior knowledge about the space $\mathrm{span}(\boldsymbol{S})$ explored by the algorithm (e.g. information about the dominant eigenspectrum). Further, only using the GP prediction at $t + 1$ implies that the algorithm's uncertainty about the action of $\boldsymbol{A}$ in $\boldsymbol{S}^{\perp}$ is of the same order as the next eigenvalue. This ignores any information about eigenvalues $\lambda_{t+2}, \ldots, \lambda_n$ contained in Rayleigh quotients and a priori known decay patterns for specific matrix classes.

**Importance of Noise (R4)** When referring to noise, we consider matrix-vector products of the form $\boldsymbol{v} \mapsto (\boldsymbol{A} + \mathbf{E}_i)\boldsymbol{v}$, where $\mathbf{E}_i \in \mathbb{R}_{\mathrm{sym}}^{n \times n}$ is Gaussian with zero mean. CG fails to converge in such a setting. While this approach can model floating-point arithmetic, typically in ML settings noise from subsampling dominates. An important example is large-scale empirical risk minimization. Due to memory constraints data needs to be batched and thus only approximate Hessian-vector products $\boldsymbol{H}_{\mathrm{batch}}\boldsymbol{v} = (\boldsymbol{H} + \mathbf{E}_{\mathrm{batch}})\boldsymbol{v}$ are available. One could use a PLS for Hessian-free optimization in this setting. This results in a trade-off between computing an accurate Hessian by sampling new batches in each iteration of the solver and taking more optimization steps in parameter space. This approach results in an optimizer which interpolates between SGD and Newton's method, depending on the batch size and number of PLS iterations $k$.

**Applications** *Transfer Learning* (R1): Using a posterior from a related problem as a prior on a new problem has the advantage over only setting $\boldsymbol{x}_0^{\mathrm{new}} = \boldsymbol{x}_k^{\mathrm{prev}}$, that uncertainty in already explored directions $\boldsymbol{S}$ is low. Hence, if the new problem $(\boldsymbol{A}^{\mathrm{new}}, \boldsymbol{b}^{\mathrm{new}})$ is similar, the covariance will contract faster. In turn also convergence will be faster (as in subspace recycling). *Kernel matrix inversion* (R4): We recognize the variety of methods available for Gram matrix inversion in the Gaussian process setting. While a comparison for different priors adapted to the kernel choice vs. a set of inducing point methods is an interesting experiment, this would have exceeded the scope of this paper. *Log-Determinant Estimation* (R3, R4): The PLS can estimate the log-determinant in $\mathcal{O}(n)$ using the proposed ln-Rayleigh regression model for uncertainty calibration via $\ln(\det(\boldsymbol{A})) = -\sum_{i=1}^{n} \ln R(\boldsymbol{A}, \boldsymbol{s}_i) \approx -(\sum_{i=1}^{k} \ln R(\boldsymbol{A}, \boldsymbol{s}_i) + \sum_{i=k+1}^{n} \mathbb{E}[\ln R_i \mid \boldsymbol{A}, \boldsymbol{S}])$. *Galerkin's Method* (R2, R4): When using a PLS as part of Galerkin's method the posterior (predictive) on a refined mesh can be derived analytically (see Proposition S6). We leave comparisons to multi-grid methods for future work.

**Other (R3, R4)** *Reorthogonalization* (R3) is a consequence of the policy choice $\boldsymbol{s}_i = -\mathbb{E}[\mathbf{H}]\boldsymbol{r}_i$, where $\mathbb{E}[\mathbf{H}]$ depends on all previous search directions as opposed to just $\boldsymbol{s}_{i-1}$ for (naive) CG. *Unification of PLS theory* (R4): Bartels et al. [13] demonstrate that the matrix-based view generalizes the solution-based view. We focus on presenting a unified matrix-based framework, which among others, connects the inference perspectives for $\boldsymbol{A}$ and $\boldsymbol{A}^{-1}$ in a rigorous way.

[Meta-Review · NeurIPS 2020]

The paper proposes a new probabilistic solver for linear systems and shows that it can improve uncertainty quantification for linear solvers which is fundamental building block that's widely used in machine learning. This is a well-written paper, I particularly enjoyed Table 1 which helps situate the work in the wider literature. The author rebuttal addresses most of the major concerns and all reviewers lean towards accept in the final discussion. I recommend accept.